

# OMEN-SED 0.9: A novel, numerically efficient organic matter sediment diagenesis module for coupling to Earth system models

Dominik Hülse[1,2], Sandra Arndt[1,3], Stuart Daines[4], Pierre Regnier[3], and Andy Ridgwell[1,2]

[1]BRIDGE, School of Geographical Sciences, University of Bristol, Clifton, Bristol BS8 1SS, UK
[2]Department of Earth Sciences, University of California, Riverside, CA 92521, USA
[3]BGeosys, Department Geoscience, Environment & Society (DGES), Université Libre de Bruxelles, Brussels, Belgium
[4]Earth System Science, University of Exeter, North Park Road, Exeter EX4 4QE, UK

*Correspondence to:* Dominik Hülse (dominik.huelse@ucr.edu)

**Abstract.** We present the first version of OMEN-SED (Organic Matter ENabled SEDiment model), a new, one-dimensional analytical early diagenetic model resolving organic matter cycling and associated biogeochemical dynamics in marine sediments designed to be coupled to Earth system models. OMEN-SED explicitly describes organic matter (OM) cycling as well as associated dynamics of the most important terminal electron acceptors (i.e. $O_2$, $NO_3$, $SO_4$) and methane ($CH_4$), related

reduced substances ($NH_4$, $H_2S$), macronutrients ($PO_4$) and associated pore water quantities (ALK, DIC). Its reaction network accounts for the most important primary and secondary redox reactions, equilibrium reactions, mineral dissolution and precipitation, as well as adsorption and desorption processes associated with OM dynamics that affect the dissolved and solid species explicitly resolved in the model. To represent a redox-dependent sedimentary P cycle we also include a representation of the formation and burial of Fe-bound P and authigenic Ca-P minerals. Thus, OMEN-SED is able to capture the main features of

diagenetic dynamics in marine sediments and, therefore, offers similar predictive abilities than a complex, numerical diagenetic model. Yet, its computational efficiency allows its coupling to global Earth system models and therefore the investigation of coupled global biogeochemical dynamics over a wide range of climate relevant timescales. This paper provides a detailed description of the new sediment model, an extensive sensitivity analysis, as well as an evaluation of OMEN-SED's performance through comprehensive comparisons with observations and results from a more complex numerical model. We find solid phase

and dissolved pore water profiles for different ocean depths are reproduced with good accuracy and simulated terminal electron acceptor fluxes fall well within the range of globally observed fluxes. Finally, we illustrate its application in an Earth system model framework by coupling OMEN-SED to the Earth system model cGENIE and tune the OM degradation rate constants to optimise the fit of simulated benthic OM contents to global observations. We find simulated sediment characteristics of the coupled model framework, such as OM degradation rates, oxygen penetration depths and sediment-water interface fluxes

are generally in good agreement with observations and in line with what one would expect on a global scale. Coupled to an Earth system model, OMEN-SED is thus a powerful tool that will not only help elucidate the role of benthic-pelagic exchange processes in the evolution and, in particular, the termination of a wide range of climate events, but will also allow a direct comparison of model output with the sedimentary record - the most important climate archive on Earth.



# 1 Introduction

Marine surface sediments are key components in the Earth system. They host the largest carbon reservoir within the surficial Earth system, provide the primary long term sink for atmospheric $CO_2$, recycle nutrients, and represent the most important geochemical archive used for deciphering past changes of biogeochemical cycles and climate (e.g. Berner, 1991; Archer and Maier-Reimer, 1994; Ridgwell and Zeebe, 2005; Arndt et al., 2013). Physical and chemical processes in sediments (i.e. diagenetic processes) depend on the water column and vice versa: Diagenesis is controlled by the external supply of solid material (e.g. organic matter, calcium carbonate, opal) from the water column and is affected by overlying bottom water concentrations of solutes. At the same time, sediments impact the water column directly either by short- and long-term storage of deposited material or diagenetic processing of deposited material and transport of terminal electron acceptors (e.g. $O_2$, $SO_4$) into the sediments, as well as metabolic products (e.g. nutrients, DIC) to the overlying bottom waters. This so-called benthic-pelagic coupling is essential for understanding global biogeochemical cycles and climate (e.g. Archer and Maier-Reimer, 1994; Archer et al., 2000; Soetaert et al., 2000; Mackenzie, 2005).

Biological primary production of organic matter (OM, generally represented in its simple form $CH_2O$ in equation R1) and the reverse process of degradation can be written in a greatly simplified reaction as:

$$CO_2 + H_2O \rightleftharpoons CH_2O + O_2. \tag{R1}$$

On geological timescales, production of OM is generally greater than degradation, which results in some organic matter being buried in marine sediments and oxygen accumulating in the atmosphere. Thus, burial of OM deep into the sediment leads to net oxygen input to, and $CO_2$ removal from the atmosphere (Berner, 2004). On shorter timescales, the upper few meters of the sediments, where early diagenesis occurs are specifically important, as this zone controls whether a substance is recycled to the water column or buried for a longer period of time in the deeper sediments (Hensen et al., 2006). Most biogeochemical cycles and reactions in this part of marine sediments can be related either directly or indirectly to the degradation of organic matter (Middelburg et al., 1993; Arndt et al., 2013). Oxygen and nitrate for instance, the highest energy yielding electron acceptors, are preferentially consumed in the course of the degradation of organic matter, resulting in the release of ammonium and phosphorus to the pore water. As such, degradation of OM in the sediments can profoundly affect the oxygen and nutrient inventory of the ocean and thus primary productivity (Van Cappellen and Ingall, 1994; Lenton and Watson, 2000). Furthermore, organic matter degradation releases metabolic $CO_2$ to the pore water, causing it to have a lower pH and carbonate ion concentration, thus provoking the dissolution of calcium carbonate $CaCO_3$ (Emerson and Bender, 1981).

Benthic nutrient recycling from marine sediments has been suggested to play a key role for climate and ocean biogeochemistry throughout Earth history. For example, feedbacks between phosphorus storage and erosion from shelf sediments and marine productivity have been hypothesised to play an important role for glacial/interglacial atmospheric $CO_2$ changes (Broecker, 1982; Ruttenberg, 1993). Furthermore, benthic nutrient recycling from anoxic sediments has been invoked to explain the occurrence of more extreme events in Earth history, for instance Oceanic Anoxic Events (OAEs, e.g. Van Cappellen and Ingall, 1994; Mort et al., 2007; Tsandev and Slomp, 2009). OAEs represent severe disturbances of the global carbon, oxygen and nutrient cycles of the ocean and are usually characterized by widespread bottom water anoxia and photic zone eu-




xinia (Jenkyns, 2010). One way to explain the genesis and persistence of OAEs is increased oxygen demand due to enhanced primary productivity. Increased nutrient inputs to fuel primary productivity may in turn have come from marine sediments as the burial efficiency of phosphorus declines when bottom waters become anoxic (Ingall and Jahnke, 1994; Van Cappellen and Ingall, 1994). The recovery from OAE like conditions is thought to involve the permanent removal of excess $CO_2$ from the

atmosphere and ocean by burying carbon in the form of organic matter in marine sediments (e.g. Arthur et al., 1988; Jarvis et al., 2011), which is consistent with the geological record of widespread black shale formation (Stein et al., 1986). Models capable of simulating not only the expansion and intensification of oxygen minimum zones, but also of predicting how the underlying sediments interact are hence needed.

Quantifications of diagenetic processes in the sediments are possible through the application of idealised mathematical

representations, or so-called diagenetic models (see e.g. Berner, 1980; Boudreau, 1997). A plethora of different approaches have been developed, mainly following two distinct directions (see Arndt et al., 2013, for an overview). First, state-of-the art vertically resolved numerical models simulating the entire suite of essential coupled redox and equilibrium reactions within marine sediments (e.g. BRNS, Aguilera et al., 2005; CANDI, Boudreau, 1996; MEDIA, Meysman et al., 2003; MUDS, Archer et al., 2002; STEADYSED, Van Cappellen and Wang, 1996). These "complete", multi-component steady-state or non-steady-

state models, thus resolve the resulting characteristic redox-zonation of marine sediments through explicitly accounting for oxic OM degradation, denitrification, oxidation by manganese and iron (hydr)oxides, sulfate reduction and methanogenesis as well as the reoxidation of reduced byproducts (i.e. $NH_4$, $Mn^{2+}$, $Fe^{2+}$, $H_2S$, $CH_4$, see e.g. Regnier et al., 2011). Furthermore, they incorporate various mineral dissolution and precipitation reactions, as well as fast equilibrium sorption processes for example of $NH_4$, $PO_4$ and metal ions (i.e. $Mn^{2+}$, $Fe^{2+}$ and $Mg^{2+}$, compare Van Cappellen and Wang, 1996; Meysman et al.,

2003). Modelled, depth-dependent, transport processes usually comprise advection, diffusion, bioturbation and bio-irrigation. This group of diagenetic models generally describes OM degradation via a so-called multi-G approach (Jørgensen, 1978; Berner, 1980), thus dividing the bulk organic matter pool into a number of compound classes that are characterised by different degradabilities $k_i$. Alternative approaches, so-called continuum models (Middelburg, 1989; Boudreau and Ruddick, 1991), assume a continuous distribution of reactive types but, although conceptually superior, are much less popular (Arndt et al.,

2013). These complex, multi-component models have a great potential for quantifying diagenetic dynamics at sites where comprehensive observational data sets are available to constrain its model parameters (see e.g. Boudreau et al., 1998; Wang and Van Cappellen, 1996; Thullner et al., 2009, for applications). However, due to the high degree of coupled processes and depth-varying parameters, the diagenetic equation needs to be solved numerically, thus resulting in a very high computational demand and consequently rendering their application in an Earth system model (ESM) framework with a large number of grid

points prohibitive. Additionally, their global applicability is seriously compromised by the restricted transferability of model parameters from one site to the global scale (Arndt et al., 2013).

The second group of diagenetic models emerged during the early days of diagenetic modelling when computing power was severely restricted (e.g. Berner, 1964). These models solve the diagenetic equation analytically, thus providing an alternative and computationally more efficient approach. Finding an analytical solution, especially when complex reaction networks are

to be considered, is not straightforward and analytical models are thus usually less sophisticated and comprehensive than nu-



merical models and generally require the assumption of steady state conditions. It has been shown that the complexity of the reaction network can be reduced by dividing the sediment column into distinct zones and accounting for the most pertinent biogeochemical processes within each zone, thus increasing the likelihood of finding an analytical solution without oversimplifying the problem. Analytical approaches with distinct biogeochemical zones were implemented and used in the seventies and eighties to describe observed pore water profiles (e.g. Vanderborght and Billen, 1975; Vanderborght et al., 1977; Billen, 1982; Goloway and Bender, 1982; Boudreau and Westrich, 1984) and later for inclusion into multi-box ecosystem models (e.g. Ruardij and Van Raaphorst, 1995; Gypens et al., 2008) and global Earth system models (Tromp et al., 1995). However, in addition to the oxic zone these models generally only describe one anoxic zone explicitly, either a denitrification (Vanderborght and Billen, 1975; Billen, 1982; Goloway and Bender, 1982; Ruardij and Van Raaphorst, 1995; Gypens et al., 2008) or a sulfate reduction zone (Boudreau and Westrich, 1984; Tromp et al., 1995). Furthermore, the approaches of Vanderborght and Billen (1975), Goloway and Bender (1982) and Tromp et al. (1995) do not explicitly account for reduced species (i.e. $NH_4$ and $H_2S$, respectively).

In most current ESMs sediment-water dynamics are either neglected or treated in a very simplistic way (Soetaert et al., 2000; Hülse et al., 2017). Most Earth system Models of Intermediate Complexity (EMICs) and also some of the higher resolution Earth system/climate models represent the sediment-water interface either as a reflective or a conservative/semi-reflective boundary (Hülse et al., 2017). Thus, all particulate material deposited on the seafloor is either instantaneously consumed (reflective boundary), or a fixed fraction is buried in the sediments (conservative/semi-reflective boundary). Both highly simplified approaches furthermore completely neglect the exchange of solute species through the sediment-water interface and, therefore, cannot resolve the complex benthic-pelagic coupling. However, due to their computational efficiency, both representations are often used in global biogeochemical models (e.g. Najjar et al., 2007; Ridgwell et al., 2007; Goosse et al., 2010). Analytical diagenetic models represent the most complex description of diagenetic dynamics in Earth system models. Examples of global ESMs employing a vertically resolved diagenetic model are NorESM (Tjiputra et al., 2013) and HAMOCC (Palastanga et al., 2011; Ilyina et al., 2013), both using a version of Heinze et al. (1999). None of the EMICs reviewed by Hülse et al. (2017) use such a sediment representation. DCESS (Shaffer et al., 2008) and MBM (Munhoven, 2007) are box models employing a vertically resolved diagenetic model. These analytic models account for the most important transport processes (i.e advection, bioturbation and molecular diffusion) through basic parametrizations and include fewer biogeochemical reactions which are generally restricted to the upper, bioturbated 10 cm of the sediments. Pore water species explicitly represented in DCESS (Shaffer et al., 2008) and the HAMOCC model of Heinze et al. (1999) and Palastanga et al. (2011) are restricted to DIC, TA, $PO_4$ and $O_2$. The MEDUSA model (Munhoven, 2007) considers $CO_2$, $HCO_3^-$, $CO_3^{2-}$ and $O_2$. Other species produced or consumed during OM degradation are neglected. Thus, with oxygen being the only TEA explicitly modelled, the influence of reduced species is only implicitly included in the boundary conditions for $O_2$. A newer version of the HAMOCC model is a notable exception, as Ilyina et al. (2013) include $NO_3$ and denitrification explicitly. Furthermore, the version of Palastanga et al. (2011) represents an redox-dependent explicit sedimentary phosphorus cycle. Yet, reoxidation of reduced byproducts, so-called secondary redox reactions (e.g. oxidation of $NH_4$, $H_2S$ or $CH_4$), or sorption processes are not included in any of the discussed models. Furthermore, these global models assume that the sedimentary organic matter pool is composed of just



a single compound class which is either degraded with a globally invariant degradation rate constant (Munhoven, 2007) or a fixed rate constant depending on local oxygen concentrations (Shaffer et al., 2008; Palastanga et al., 2011).

Obviously, such a simplification of the OM pool can neither account for the observed vast structural complexity in natural organic matter and its resulting different degradation rates nor for the rapid decrease in OM degradability in the uppermost
centimetres of the sediments (Arndt et al., 2013). It has been suggested that at least a 3G approach is necessary to accurately represent organic matter dynamics in this part of the sediments where most OM is degraded (e.g. Soetaert et al., 1996). Even more restrictive is the use of $O_2$ as the only TEA and the complete absence of reduced substances and related secondary redox reactions. For the majority of the modern sediments (i.e. in the deep-ocean) $O_2$ is the primary electron acceptor, however, recent model and data studies have reported that sulfate reduction is the dominant degradation pathway on a global average
(with contributions of 55-76% Canfield et al., 2005; Jørgensen and Kasten, 2006; Thullner et al., 2009). Oxygen becomes progressively less important as TEA with decreasing seafloor depth and sulfate reduction has been shown to account for 83% of OM degradation in coastal sediments (Krumins et al., 2013). In these environments most $O_2$ is used to reoxidise reduced substances produced during anaerobic degradation (Canfield et al., 2005; Thullner et al., 2009). Thus, the in situ production of e.g. $NO_3$ and $SO_4$ through oxidation of $NH_4$ and $H_2S$ forms an important sink for $O_2$ which is entirely neglected in current
sediment representations in global models. In addition, the lack of anoxic degradation pathways in these models limits their application to oxic oceans. Currently no analytical sediment model exists that can be used under anoxic conditions. Due to the lack of an appropriate sedimentary P cycle (with the exception of the HAMOCC version of Palastanga et al., 2011), no current global ESM is able to model the redox dependent P release from marine sediments and its implications for primary productivity, global biogeochemical cycles and climate. A sediment model suitable for the coupling to an ESM and enabling a wide range of
paleo questions to be addressed has to provide a robust quantification of organic (and inorganic) carbon burial fluxes, benthic uptake/return fluxes of oxygen, growth-limiting nutrients and reduced species, as well as anoxic degradation pathways. As a consequence, the reaction network must account for the most important primary and secondary redox reactions, equilibrium reactions, mineral precipitation/dissolution and adsorption/desorption, resulting in a complex set of coupled reaction-transport equations.

Therefore, we developed the OrganicMatter ENabled SEDiment model (OMEN-SED), a new, one-dimensional, numerically efficient diagenetic model. OMEN-SED builds upon and stands in the tradition of earlier stand-alone, analytical diagenetic models (Vanderborght et al., 1977; Billen, 1982; Goloway and Bender, 1982; Boudreau, 1991), as well as of analytical diagenetic models developed for the coupling to regional scale ecosystem or global Earth system models (Ruardij and Van Raaphorst, 1995; Tromp et al., 1995; Heinze et al., 1999; Gypens et al., 2008).

OMEN-SED is the first analytical model to explicitly describe OM cycling as well as associated dynamics of the most important TEAs (i.e. $O_2$, $NO_3$, $SO_4$), related reduced substances ($NH_4$, $H_2S$), the full suite of secondary-redox reactions, macronutrients ($PO_4$) and associated pore water quantities (ALK, DIC). To represent a redox-dependent sedimentary P cycle we consider the formation and burial of Fe-bound P and authigenic Ca-P minerals. Thus, while OMEN-SED captures most of the features of a complex, numerical diagenetic model, its computational efficiency allows the coupling to global Earth
system models and therefore the investigation of coupled global biogeochemical dynamics over different timescales. Here,





the model is presented as a 2G-approach, however, OMEN-SED can be easily extended to a Multi-G approach. The first part of the paper provides a detailed description of OMEN-SED (Section 2). This includes descriptions of the general model approach (Section 2.1), of the conservation equations for all explicitly represented biogeochemical tracers (Section 2.2), as well as a summary of global relationships used to constrain reaction and transport parameters in OMEN-SED (Section 2.4). In addition, a generic algorithm is described which is used to match internal boundary conditions and to determine the integration constants for the analytical solutions (Section 2.3). In order to validate the stand-alone version of OMEN-SED, the second part of the paper performs an extensive sensitivity analysis for the most important model parameters and resulting sediment-water interface fluxes are compared with a global database (Section 3.1). In addition, results of the stand-alone model are compared with observed pore water profiles from different ocean depths (Section 3.2) and OMEN-SED simulations of TEA-fluxes along a typical ocean transect are compared with observations and results from a complete, numerical diagenetic model (Section 3.3). Thereafter, OMEN-SED is coupled to the carbon-centric version of the "GENIE" Earth system model (cGENIE, Ridgwell et al., 2007, Section 4.1). Sensitivity studies are carried out using this coupled model and modelled organic matter concentrations in the surface sediments are compared to a global database (Seiter et al., 2004, Section 4.2). We finally discuss potential applicabilities of OMEN-SED and critically analyse model limitations (Section 5).

## 2 Model Description

OMEN-SED is implemented as a FORTRAN version that can be easily coupled to any pelagic, biogeochemical model via the coupling routine **OMEN_SED_main**. In addition, OMEN-SED exists as a stand-alone version implemented in MATLAB and the entire model can be executed on a standard personal computer in less than 0.1 seconds. The source code of both, the FORTRAN and the MATLAB stand-alone version, as well as instructions for executing OMEN-SED and for plotting model results are available as a supplement to this paper.

The following section provides a detailed description of OMEN-SED and the fundamental equations underlying the model are highlighted. Tables 1 and A1 summarise the biogeochemical reaction network and Tables 9 and 10 provide a glossary of model parameters along with their respective units.

### 2.1 General Model Approach

In OMEN-SED, the calculation of benthic uptake, recycling and burial fluxes is based on the vertically resolved conservation equation for solid and dissolved species in porous media (e.g. Berner, 1980; Boudreau, 1997):

$$\frac{\partial \xi C_i}{\partial t} = -\frac{\partial}{\partial z}\left(-\xi D_i \frac{\partial C_i}{\partial z} + \xi w C_i\right) + \xi \sum_j R_i^j \tag{1}$$

where $C_i$ is the concentration of biogeochemical species $i$, $\xi$ equals the porosity $\phi$ for solute species and $(1-\phi)$ for solid species. The term $z$ is the sediment depth, $t$ denotes the time, $D_i$ is the apparent diffusion coefficient of species $i$, $w$ is the burial rate and $\sum_j R_i^j$ represents the sum of all biogeochemical rates $j$ affecting species $i$.





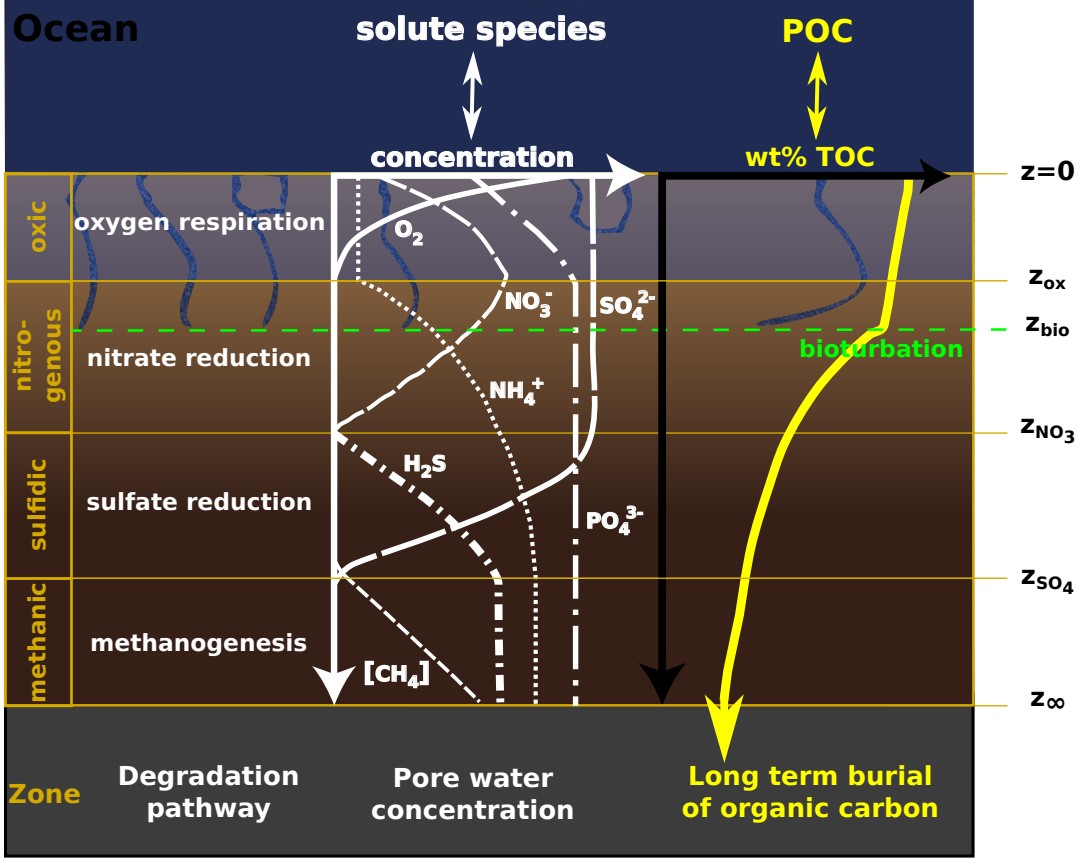

**Figure 1.** Schematic of the different modelled species and zones in OMEN-SED. Here showing the case $z_{\mathrm{ox}} < z_{\mathrm{bio}} < z_{\mathrm{NO_3}} < z_{\mathrm{SO_4}}$.

OMEN-SED accounts for both the advective, as well as the diffusive transport of solid and dissolved species. They are buried in the sediment according to a constant burial rate $w$, thus neglecting the effect of sediment compaction (i.e. $\frac{\partial \phi}{\partial z} = 0$) due to mathematical constraints. The molecular diffusion of dissolved species is described by Fick's law applying a species-specific apparent diffusion coefficient, $D_{\mathrm{mol},i}$. In addition, the activity of infaunal organisms in the bioturbated zone is simulated

5   using a diffusive term (e.g. Boudreau, 1986), with a constant bioturbation coefficient $D_{\mathrm{bio}}$ in the bioturbated zone, while $D_{\mathrm{bio}}$ is set to zero below the maximum bioturbation depth, $z_{bio}$. The pumping activity by burrow-dwelling animals and the resulting ventilation of tubes, the so-called bioirrigation, is encapsulated in a factor $f_{ir}$ that enhances the molecular diffusion coefficient (hence, $D_{i,0} = D_{\mathrm{mol},i} \cdot f_{ir}$, Soetaert et al., 1996). The reaction network of OMEN-SED accounts for the most important primary and secondary redox reactions, equilibrium reactions, mineral dissolution and precipitation, as well as

10   adsorption and desorption processes associated with OM dynamics that affect the dissolved and solid species explicitly resolved in the model. Tables 1 and A1 provide a summary of the reactions and biogeochemical tracers considered in OMEN-SED together with their respective reaction stoichiometries.





**Table 1.** Reactions and biogeochemical tracers implemented in the reaction network of OMEN-SED. The primary and secondary redox reactions are listed in the sequence they occur with increasing sediment depth.

|  | Description |
|---|---|
| Primary redox reactions | Degradation of organic matter via aerobic degradation, denitrification, sulfate reduction, methanogenesis (implicit) |
| Secondary redox reactions | Oxidation of ammonium and sulfide by oxygen, anaerobic oxidation of methane by sulfate |
| Adsorption/Desorption | Ad-/Desorption of P on/from $Fe(OH)_3$, $NH_4$ adsorption, $PO_4$ adsorption |
| Mineral precipitation | Formation of authigenic P |
| Biogeochemical tracers | Organic matter (2-G or pseudo 3-G), oxygen, nitrate, ammonium, sulfate, sulfide (hydrogen sulfide), phosphate, Fe-bound P, DIC, ALK |

All parameters in Eq. (1) may vary with depth and many reaction rate expressions depend on the concentration of other species. Expressing Eq. (1) for a set of chemical species thus results in a non-linear, coupled set of equations that can only be solved numerically. However, OMEN-SED is designed for the coupling to Earth system models and, therefore, cannot afford a computationally expensive numerical solution. Instead, similar to early, analytical diagenetic models, a computationally

efficient analytical solution of Eq. (1) can be derived by 1) assuming steady state conditions (i.e. $\frac{\partial C_i}{\partial t} = 0$) and 2) reducing the vertical variability in parameters and reaction rate expressions by dividing the sediment column into a number of functional biogeochemical zones (Fig. 1, compare e.g. Billen, 1982; Goloway and Bender, 1982; Ruardij and Van Raaphorst, 1995; Tromp et al., 1995; Gypens et al., 2008, for similar solutions). More specifically, OMEN-SED follows Berner (1980) by dividing the sediment column into: I) a bioturbated and II) a non-bioturbated zone defined by an imposed, constant bioturbation depth $z_{bio}$

(Fig. 1). Furthermore, it resolves the dynamic redox stratification of marine sediments by dividing the sediment into 1) an oxic zone delineated by the oxygen penetration depth $z_{ox}$; 2) a denitrification (or nitrogenous) zone situated between $z_{ox}$ and the nitrate penetration depth $z_{NO_3}$; 3) a sulfate reduction zone situated between $z_{NO_3}$ and the sulfate penetration depth $z_{SO_4}$; and 4) a methanogenic zone situated below $z_{SO_4}$ (Fig. 1). Although in each of these zones Eq. (1) is applied with depth invariant parameters, parameter values may differ across zones. The biogeochemical zones are linked by stating continuity in both

concentrations and fluxes at the dynamic, internal boundaries ($z_b \in \{z_{bio}, z_{ox}, z_{NO_3}, z_{SO_4}\}$, compare e.g. Billen, 1982; Ruardij and Van Raaphorst, 1995). Note that these boundaries are dynamic because their depth varies in response to changing ocean boundary conditions and forcings (see Section 2.3.1 for details). Furthermore, the maximum bioturbation depth is not restricted to a specific biogeochemical zone, hence OMEN-SED allows bioturbation to occur in the anoxic zones of the sediment (here all zones $z > z_{ox}$ combined).

The formulation of the reaction term in Eq. (1) varies between zones and encapsulates the most pertinent reaction processes within the respective zone (see Section 2.2), thus simplifying the mathematical description of the reaction network while retaining most of its biogeochemical complexity. One such simplification is that the solid phase iron and manganese oxidants and its reductants are not considered in the reaction network. All consumption or production processes of dissolved species related to the degradation of organic matter are a function of the organic matter concentration. Because organic matter degradation is





described as a first-order degradation, these processes can be expressed as a series of exponential terms ($\sum_j \alpha_j \exp(-\beta_j z)$, see Eq. (2)). In addition, slow adsorption/desorption and mineral precipitation processes can be expressed as zero or first order (reversible) reaction ($Q_m$ or $k_l \cdot C_i$, in Eq. (2)). Fast adsorption is described as an instantaneous equilibrium reaction using a constant adsorption coefficient $K_i$. The reoxidation of reduced substances is accounted for implicitly by adding a (consump-

tion/production) flux to the internal boundary conditions (see Sections 2.2.2, 2.2.3 and 2.2.4). This simplification has been used previously by Gypens et al. (2008) for nitrate and ammonium and can be justified as it has been shown that the reoxidation mainly occurs within a thin layer at the oxic/anoxic interface (Soetaert et al., 1996). The general reaction-transport equation underlying OMEN-SED is thus given by:

$$\frac{\partial C_i}{\partial t} = 0 = \frac{D_i}{1+K_i}\frac{\partial^2 C_i}{\partial z^2} - w\frac{\partial C_i}{\partial z} - \frac{1}{1+K_i}\left(\sum_j \alpha_j \exp(-\beta_j z) + \sum_l k_l \cdot C_i - \sum_m Q_m\right) \tag{2}$$

where $1/\beta_j$ can be interpreted as the length scale and $\alpha_j$ as the relative importance (or the magnitude at $z = 0$) of reaction $j$ (Boudreau, 1997), $k_l$ are generic first order reaction rate constants and $Q_m$ are zero-order (or constant) reaction rates.

The analytical solution of Eq. (2) is of the general form:

$$C_i(z) = A \cdot \exp(az) + B \cdot \exp(bz) + \sum_j \frac{\alpha_j}{D\beta_j^2 - w\beta_j - \sum_l k_l} \cdot \exp(-\beta_j z) + \frac{\sum_m Q_m}{\sum_l k_l} \tag{3}$$

with

$$a = \frac{w - \sqrt{w^2 + 4 \cdot D \cdot \sum_l k_l}}{2 \cdot D}, \quad b = \frac{w + \sqrt{w^2 + 4 \cdot D \cdot \sum_l k_l}}{2 \cdot D} \tag{4}$$

where $A$ and $B$ are integration constants that can be determined by applying a set of internal boundary conditions (see Section 2.3) and $D = \frac{D_i}{1+K_i}$.

Based on Eq. (2) and its analytical solution Eq. (3), OMEN-SED returns the fraction of particulate organic carbon (POC)

buried in the sediment, $f_{\text{POC}}$, as well as the benthic uptake/return fluxes $F_{C_i}$ of dissolved species $C_i$ (in mol cm$^{-2}$ year$^{-1}$) in response to the dynamic interplay of transport and reaction processes under changing boundary conditions and forcings:

$$f_{\text{POC}} = \frac{\text{POC}(z_\infty)}{\text{POC}(0)} \tag{5}$$

$$F_{C_i} = \phi(0)\left(D_i\frac{\partial C_i(z)}{\partial z}\bigg|_{z=0} - w \cdot C_i(0)\right) \tag{6}$$

where $w$ is the deposition rate, $D_i$ is the diffusion coefficient and $\text{POC}(0)$, $\text{POC}(z_\infty)$, $C_i(0)$ denote the concentration of POC

and dissolved species $i$ at the SWI and at the lower sediment boundary, respectively.

## 2.2   Conservation Equations and Analytical Solution

The following sections provide a detailed description of the conservation equations and analytical solutions for each chemical species that is resolved in this version of OMEN-SED.





### 2.2.1 Organic Matter or Particulate Organic Carbon (POC)

In marine sediments, organic matter (or in the following called particulate organic carbon, POC) is degraded by heterotrophic activity coupled to the sequential utilisation of terminal electron acceptors (TEAs) according to the free energy gain of the half-reaction ($O_2 > NO_3^- > MnO_2 > Fe(OH)_3 > SO_4^{2-}$, e.g. Stumm and Morgan, 2012). Once all TEAs are depleted, organic

matter is degraded via methanogenesis. Here, organic matter degradation is described via a multi G-model approach (Jørgensen, 1978), dividing the bulk OM into a number $i$ of discrete compound classes $POC_i$ characterised by class-specific first-order degradation rate constants $k_i$. The conservation equation for organic matter dynamics is thus given by:

$$\frac{\partial POC_i}{\partial t} = 0 = D_{POC_i} \frac{\partial^2 POC_i}{\partial z^2} - w \frac{\partial POC_i}{\partial z} - k_i \cdot POC_i \tag{7}$$

with $D_{POC_i} = D_{bio}$ for $z \leq z_{bio}$ and $D_{POC_i} = 0$ for $z > z_{bio}$. Integration of equations (7) yields the following general solu-

tions for the bioturbated and non-bioturbated layers:

I. Bioturbated zone ($z \leq z_{bio}$)

$$POC_i^I(z) = A_{1i} \cdot exp(a_{1i}z) + B_{1i} \cdot exp(b_{1i}z) \tag{8}$$

II. Non-bioturbated zone ($z_{bio} < z$)

$$POC_i^{II}(z) = A_{2i} \cdot exp(a_{2i}z) \tag{9}$$

where

$$a_{1i} = \frac{w - \sqrt{w^2 + 4 \cdot D_{POC_i} \cdot k_i}}{2 \cdot D_{POC_i}}, \quad b_{1i} = \frac{w + \sqrt{w^2 + 4 \cdot D_{POC_i} \cdot k_i}}{2 \cdot D_{POC_i}}, \quad a_{2i} = -\frac{k_i}{w} \tag{10}$$

Determining the integration constants ($A_{1,i}$, $B_{1,i}$, $A_{2,i}$) requires the definition of a set of boundary conditions (Table 2). For organic matter, OMEN-SED applies a known concentration/flux at the sediment-water interface and assumes continuity across the bottom of the bioturbated zone, $z_{bio}$. The integration constants ($A_{1,i}$, $B_{1,i}$, $A_{2,i}$) are thus given by:

$$B_{1i} \overset{BC1)}{=} POC_{0i} - A_{1i} \tag{11}$$

$$A_{2i} \overset{BC2)}{=} \frac{A_{1i} \cdot exp(a_{1i}z_{bio}^-) + B_{1i} \cdot exp(b_{1i}z_{bio}^-)}{exp(a_{2i}z_{bio}^+)}$$

$$A_{1i} \overset{BC3)}{=} -\frac{B_{1i}b_{1i} \cdot exp(b_{1i}z_{bio}^-)}{a_{1i} \cdot exp(a_{1i}z_{bio}^-)}$$

See Section 2.3.1 for further details on how to find the analytical solution.

### 2.2.2 Oxygen

OMEN-SED explicitly accounts for oxygen consumption by the aerobic degradation of organic matter within the oxic zone, as well as the oxidation of reduced species (i.e. $NH_4$, $H_2S$) produced in the anoxic zones of the sediment. In the oxic zone





**Table 2.** OM Boundary conditions applied in OMEN-SED. For the boundaries we define: $z_{\mathrm{bio}}^- := \lim_{h\to 0}(z_{\mathrm{bio}} - h)$ and $z_{\mathrm{bio}}^+ := \lim_{h\to 0}(z_{\mathrm{bio}} + h)$.

| Boundary | Condition | | |
|---|---|---|---|
| $z = 0$ | known concentration | 1) | $\mathrm{POC}_i(0) = \mathrm{POC}_{0i}$ |
| $z = z_{\mathrm{bio}}$ | continuity | 2) | $\mathrm{POC}_i(z_{\mathrm{bio}}^-) = \mathrm{POC}_i(z_{\mathrm{bio}}^+)$ |
| | | 3) | $-D_{\mathrm{bio}} \cdot \frac{\partial \mathrm{POC}_i}{\partial z}\big|_{z_{\mathrm{bio}}^-} = 0$ |

($z < z_{ox}$), the aerobic degradation consumes oxygen with a fixed $\mathrm{O_2 : C}$ ratio ($\mathrm{O_2C}$, Tab. 10). A predefined fraction, $\gamma_{\mathrm{NH_4}}$, of the ammonium produced during the aerobic degradation of OM is nitrified to nitrate, consuming two moles of oxygen per mole of ammonium produced. In addition, OMEN-SED implicitly accounts for the oxygen consumption due to oxidation of reduced species ($\mathrm{NH_4}$, $\mathrm{H_2S}$) produced below the oxic zone through the flux boundary condition at the dynamically calculated
5  (see section 2.4.2 for details) oxygen penetration depth $z_{\mathrm{ox}}$. All oxygen consumption processes can thus be formulated as a function of organic matter degradation. The conservation equation for oxygen is given by:

$$\frac{\partial \mathrm{O_2}}{\partial t} = 0 = D_{\mathrm{O_2}} \frac{\partial^2 \mathrm{O_2}}{\partial z^2} - w \frac{\partial \mathrm{O_2}}{\partial z} - \frac{1-\phi}{\phi} \sum_i k_i \cdot [\mathrm{O_2C} + 2\gamma_{\mathrm{NH_4}} \mathrm{NC_i}] \cdot \mathrm{POC}_i(z) \tag{12}$$

For illustrative purposes, we here substitute the analytical solution for the POC depth profile and provide the analytical solution. The remaining paragraphs only outline the general equation, whose analytical solution can be derived in an identical manner. Substituting Eq. (8) and (9) for $\mathrm{POC_i(z)}$ and Eq. (11) for $B_{1i}$ gives:

I Bioturbated zone ($z \leq z_{\mathrm{bio}}$)

$$\frac{\partial \mathrm{O_2^I}}{\partial t} = 0 \overset{8\&11}{=} D_{\mathrm{O_2}}^I \frac{\partial^2 \mathrm{O_2}}{\partial z^2} - w \frac{\partial \mathrm{O_2}}{\partial z} - \frac{1-\phi}{\phi} \sum_i k_i \cdot [\mathrm{O_2C} + 2\gamma_{\mathrm{NH_4}} \mathrm{NC_i}] \cdot \Big( A_{1i} \cdot [exp(a_{1i}z) - exp(b_{1i}z)] + \mathrm{POC}_{0i} \cdot exp(b_{1i}z) \Big)$$

II Non-bioturbated zone ($z_{\mathrm{bio}} < z < z_{\mathrm{ox}}$)

$$\frac{\partial \mathrm{O_2}^{II}}{\partial t} = 0 \overset{9}{=} D_{\mathrm{O_2}}^{II} \frac{\partial^2 \mathrm{O_2}}{\partial z^2} - w \frac{\partial \mathrm{O_2}}{\partial z} - \frac{1-\phi}{\phi} \sum_i k_i \cdot [\mathrm{O_2C} + 2\gamma_{\mathrm{NH_4}} \mathrm{NC_i}] \cdot \Big( A_{2i} \cdot exp(a_{2i}z) \Big)$$

where $D_{\mathrm{O_2}}^I$ and $D_{\mathrm{O_2}}^{II}$ denote the $\mathrm{O_2}$ diffusion coefficient for the bioturbated and non-bioturbated zone, respectively. The term $\frac{1-\phi}{\phi}$ accounts for the volume conversion from solid to dissolved phase and $\mathrm{NC_i}$ is the nitrogen to carbon ratio in POC.
15  Integration yields the following analytical solution for each zone:

I Bioturbated zone ($z \leq z_{\mathrm{bio}}$):

$$\mathrm{O_2}^I(z) = A_{\mathrm{O_2}}^1 + B_{\mathrm{O_2}}^1 \cdot \exp(b_{\mathrm{O_2}}^1 z) + \sum_i \Phi_{1,i}^I \cdot \exp(a_{1i}z) + \sum_i \Phi_{1,i}^{II} \cdot \exp(b_{1i}z) + \sum_i \Phi_{1,i}^{III} \cdot \exp(b_{1i}z) \tag{13}$$



**Table 3.** Boundary conditions for oxygen. For the boundaries we define: $z_{\text{bio}}^{-} := \lim_{h \to 0}(z_{\text{bio}} - h)$ and $z_{\text{bio}}^{+} := \lim_{h \to 0}(z_{\text{bio}} + h)$.

| Boundary | Condition | | |
|---|---|---|---|
| $z = 0$ | known concentration | 1) | $O_2(0) = O_{20}$ |
| $z = z_{\text{bio}}$ | continuity | 2) | $O_2(z_{\text{bio}}^{-}) = O_2(z_{\text{bio}}^{+})$ |
| | | 3) | $-(D_{O_2,0} + D_{\text{bio}}) \cdot \frac{\partial O_2}{\partial z}\big|_{z_{\text{bio}}^{-}} = -D_{O_2,0} \cdot \frac{\partial O_2}{\partial z}\big|_{z_{\text{bio}}^{+}}$ |
| $z = z_{\text{ox}}$ | $O_2$ consumption | 4) | **IF** $(O_2(z_\infty) > 0)$ |
| $(z_{\text{ox}} = z_\infty)$ | | 4.1) | $\frac{\partial O_2}{\partial z}\big|_{z_{\text{ox}}} = 0$ |
| | | | **ELSE** |
| $(z_{\text{ox}} < z_\infty)$ | | 4.2) | $O_2(z_{\text{ox}}) = 0 \quad$ and $\quad -D_{O_2} \cdot \frac{\partial O_2}{\partial z}\big|_{z_{\text{ox}}} = F_{red}(z_{\text{ox}})$ |
| | with | | $F_{red}(z_{\text{ox}}) = \frac{1-\phi}{\phi} \cdot \int_{\tilde{z}_{\text{NO}_3}}^{\infty} \sum_i (2\gamma_{\text{NH}_4}\text{NC}_i + 2\gamma_{\text{H}_2\text{S}}\text{SO}_4\text{C}) k_i \text{POC}_i \, dz$ |

Note: $\tilde{z}_{\text{NO}_3} = z_{\text{ox}}$ as upper boundary here, as $z_{\text{NO}_3}$ is not known at this point.

II Non-bioturbated zone ($z_{\text{bio}} < z < z_{\text{ox}}$)

$$O_2{}^{II}(z) = A_{O_2}^2 + B_{O_2}^2 \cdot \exp(b_{O_2}^2 z) + \sum_i \Phi_{i,2}^I \cdot \exp(a_{2i}z) \tag{14}$$

with

$$b_{O_2}^1 = \frac{w}{D_{O_2}^I}, \quad b_{O_2}^2 = \frac{w}{D_{O_2}^{II}}$$

$$\Phi_{1,i}^I = \frac{1-\phi}{\phi} \cdot \frac{k_i \cdot (O_2\text{C} + 2\gamma_{\text{NH}_4}\text{NC}_i) \cdot A_{1i}}{D_{O_2}^I(-a_{1i})^2 - w \cdot (-a_{1i})}, \qquad\qquad \Phi_{1,i}^{II} = -\frac{1-\phi}{\phi} \cdot \frac{k_i \cdot (O_2\text{C} + 2\gamma_{\text{NH}_4}\text{NC}_i) \cdot A_{1i}}{D_{O_2}^I(-b_{1i})^2 - w \cdot (-b_{1i})}$$

$$\Phi_{1,i}^{III} = \frac{1-\phi}{\phi} \cdot \frac{k_i \cdot (O_2\text{C} + 2\gamma_{\text{NH}_4}\text{NC}_i) \cdot \text{POC}_{0i}}{D_{O_2}^I(-b_{1i})^2 - w \cdot (-b_{1i})}$$

$$\Phi_{i,2}^I := \frac{1-\phi}{\phi} \cdot \frac{k_i \cdot (O_2\text{C} + 2\gamma_{\text{NH}_4}\text{NC}_i) \cdot A_{2i}}{D_{O_2}^{II}(-a_{2i})^2 - w \cdot (-a_{2i})}$$

Determining the four integration constants ($A_{O_2}^1, B_{O_2}^1, A_{O_2}^2, B_{O_2}^2$, see Section 2.3 for details), as well as the *a priori* unknown oxygen penetration depth requires the definition of five boundary conditions (see Table 3). At the sediment-water interface, OMEN-SED applies a Dirichlet condition (i.e. known concentration) and assumes concentration and flux continuity across the bottom of the bioturbated zone, $z_{\text{bio}}$. The oxygen penetration depth $z_{\text{ox}}$ marks the lower boundary and is dynamically calculated as the depth at which $O_2(z) = 0$. Therefore, OMEN-SED applies a Dirichlet boundary condition $O_2(z_{\text{ox}}) = 0$. In addition, a flux boundary is applied that implicitly accounts for the oxygen consumption by the partial oxidation of $\text{NH}_4$ and $\text{H}_2\text{S}$ diffusing into the oxic zone from below (BC 4.2, Table 3). It is assumed that respective fractions ($\gamma_{\text{NH}_4}$ and $\gamma_{\text{H}_2\text{S}}$) are directly reoxidised at the oxic/anoxic interface and the remaining fraction escapes reoxidation. OMEN-SED iteratively solves for $z_{\text{ox}}$ by first testing if there is oxygen left at $z_\infty$ (i.e. $O_2(z_\infty) > 0$). If that is not the case, it determines the root for the flux boundary condition 4.2 (Table 3). If $z_{\text{ox}} = z_\infty$, a zero diffusive flux boundary condition is applied as lower boundary condition.





### 2.2.3 Nitrate and Ammonium

Nitrogen dynamics in OMEN-SED are controlled by the metabolic production of ammonium, nitrification, denitrification as well as ammonium adsorption. Ammonium is produced by organic matter degradation in both the oxic and anoxic zones, while denitrification consumes nitrate in the denitrification zone with a fixed $NO_3 : C$ ratio ($NO_3C$, Tab. 10).

The adsorption of ammonium to sediment particles is formulated as an equilibrium process with constant equilibrium adsorption coefficient $K_{NH_4}$, thus assuming that the adsorption is fast compared to the characteristic time scales of transport processes (Wang and Van Cappellen, 1996). In addition, a defined fraction, $\gamma_{NH_4}$, of metabolically produced ammonium is directly nitrified to nitrate in the oxic zone, while the nitrification of upward diffusing ammonium produced in the sulfidic and methanic zones is implicitly accounted for in the boundary conditions. The conservation equations for ammonium and nitrate
are thus given by:

1. Oxic zone ($z \leq z_{ox}$)

$$\frac{\partial NO_3{}^I}{\partial t} = 0 = D_{NO_3} \frac{\partial^2 NO_3{}^I}{\partial z^2} - w \frac{\partial NO_3{}^I}{\partial z} + \gamma_{NH_4} \frac{1-\phi}{\phi} \cdot \sum_i NC_i \cdot k_i \cdot POC_i(z) \tag{15}$$

$$\frac{\partial NH_4{}^I}{\partial t} = 0 = \frac{D_{NH_4}}{1+K_{NH_4}} \frac{\partial^2 NH_4{}^I}{\partial z^2} - w \frac{\partial NH_4{}^I}{\partial z} + \frac{1-\gamma_{NH_4}}{1+K_{NH_4}} \cdot \frac{1-\phi}{\phi} \cdot \sum_i NC_i \cdot k_i \cdot POC_i(z) \tag{16}$$

2. Denitrification (or nitrogenous) zone ($z_{ox} < z \leq z_{NO_3}$)

$$\frac{\partial NO_3{}^{II}}{\partial t} = 0 = D_{NO_3} \frac{\partial^2 NO_3{}^{II}}{\partial z^2} - w \frac{\partial NO_3{}^{II}}{\partial z} - \frac{1-\phi}{\phi} NO_3C \cdot \sum_i k_i \cdot POC_i(z) \tag{17}$$

$$\frac{\partial NH_4{}^{II}}{\partial t} = 0 = \frac{D_{NH_4}}{1+K_{NH_4}} \frac{\partial^2 NH_4{}^{II}}{\partial z^2} - w \frac{\partial NH_4{}^{II}}{\partial z} \tag{18}$$

3. Sulfidic and methanic zone ($z_{NO_3} < z \leq z_\infty$)

$$\frac{\partial NH_4{}^{III}}{\partial t} = 0 = \frac{D_{NH_4}}{1+K_{NH_4}} \frac{\partial^2 NH_4{}^{III}}{\partial z^2} - w \frac{\partial NH_4{}^{III}}{\partial z} + \frac{1}{1+K_{NH_4}} \cdot \frac{1-\phi}{\phi} \cdot \sum_i NC_i \cdot k_i \cdot POC_i(z) \tag{19}$$

where $D_{NO_3}$ and $D_{NH_4}$ denote the diffusion coefficients for $NO_3$ and $NH_4$ which depend on the bioturbation status of the respective geochemical zone (compare Section 2.3.1). Integration of Eq. (15) - (19) yields the analytical solutions, which are not further developed here but follow the procedure outlined in Section 2.2.2 for oxygen (also see Section 2.3.1 for more details on how to find the analytical solution). Table 4 summarises the boundary conditions applied in OMEN-SED to solve Eq. (15) - (19) and to find the *a priori* unknown nitrate penetration depth, $z_{NO_3}$. The model assumes known bottom water concentrations
for both $NO_3$ and $NH_4$, the complete consumption of nitrate at the nitrate penetration depth (in case $z_{NO_3} < z_\infty$) and no change in ammonium flux at $z_\infty$. In addition, concentration and diffusive flux continuity across $z_{bio}$ and $z_{ox}$ is considered for $NO_3$ and $NH_4$. Furthermore, the reoxidation of upward-diffusing reduced ammonium is accounted for in the oxic-anoxic boundary condition for nitrate and ammonium. OMEN-SED iteratively solves for $z_{NO_3}$ by first testing if there is nitrate left at $z_\infty$ (i.e. $NO_3(z_\infty) > 0$) and, otherwise, by finding the root for the flux boundary condition 6.2 (Table 4).





**Table 4.** Boundary conditions for nitrate and ammonium. For the boundaries we define: $z_-^- := \lim_{h \to 0}(z_- - h)$ and $z_-^+ := \lim_{h \to 0}(z_- + h)$.

| Boundary | Condition | | |
|---|---|---|---|
| $z = 0$ | known concentration | 1) | $\mathrm{NO_3}(0) = \mathrm{NO_{30}}$ |
| $z = z_\mathrm{bio}$ | continuity | 2) | $\mathrm{NO_3}(z_\mathrm{bio}^-) = \mathrm{NO_3}(z_\mathrm{bio}^+)$ |
| | | 3) | $-(D_{\mathrm{NO_3},0} + D_\mathrm{bio}) \cdot \frac{\partial \mathrm{NO_3}}{\partial z}\big|_{z_\mathrm{bio}^-} = -D_{\mathrm{NO_3},0} \cdot \frac{\partial \mathrm{NO_3}}{\partial z}\big|_{z_\mathrm{bio}^+}$ |
| $z = z_\mathrm{ox}$ | continuity | 4) | $\mathrm{NO_3}(z_\mathrm{ox}^-) = \mathrm{NO_3}(z_\mathrm{ox}^+)$ |
| | | 5) | $-D_{\mathrm{NO_3}} \cdot \frac{\partial \mathrm{NO_3}}{\partial z}\big|_{z_\mathrm{ox}^-} + \gamma_{\mathrm{NH_4}} \cdot F_{\mathrm{NH_4}}(z_\mathrm{ox}) = -D_{\mathrm{NO_3}} \cdot \frac{\partial \mathrm{NO_3}}{\partial z}\big|_{z_\mathrm{ox}^+}$ |
| | where: | | $F_{\mathrm{NH_4}}(z_\mathrm{ox}) = \frac{1}{1 + K_{\mathrm{NH_4}}} \cdot \frac{1 - \phi}{\phi} \cdot \int_{z_{\mathrm{NO_3}}}^{\infty} \sum_i \mathrm{NC_i} \cdot k_i \cdot \mathrm{POC}_i\, dz$ |
| $z = z_{\mathrm{NO_3}}$ | NO$_3$ consumption | 6) | **IF** $(\mathrm{NO_3}(z_\infty) > 0)$ |
| | $(z_{\mathrm{NO_3}} = z_\infty)$ | 6.1) | $\frac{\partial \mathrm{NO_3}}{\partial z}\big|_{z_{\mathrm{NO_3}}} = 0$ |
| | | | **ELSE** |
| | $(z_{\mathrm{NO_3}} < z_\infty)$ | 6.2) | $\mathrm{NO_3}(z_{\mathrm{NO_3}}) = 0$    and    $\frac{\partial \mathrm{NO_3}}{\partial z}\big|_{z_{\mathrm{NO_3}}} = 0$ |
| $z = 0$ | known concentration | 1) | $\mathrm{NH_4}(0) = \mathrm{NH_{40}}$ |
| $z = z_\mathrm{bio}$ | continuity | 2) | $\mathrm{NH_4}(z_\mathrm{bio}^-) = \mathrm{NH_4}(z_\mathrm{bio}^+)$ |
| | | 3) | $-\frac{D_{\mathrm{NH_4},0} + D_\mathrm{bio}}{1 + K_{\mathrm{NH_4}}} \cdot \frac{\partial \mathrm{NH_4}}{\partial z}\big|_{z_\mathrm{bio}^-} = -\frac{D_{\mathrm{NH_4},0}}{1 + K_{\mathrm{NH_4}}} \cdot \frac{\partial \mathrm{NH_4}}{\partial z}\big|_{z_\mathrm{bio}^+}$ |
| $z = z_\mathrm{ox}$ | continuity | 4) | $\mathrm{NH_4}(z_\mathrm{ox}^-) = \mathrm{NH_4}(z_\mathrm{ox}^+)$ |
| | | 5) | $-\frac{D_{\mathrm{NH_4}}}{1 + K_{\mathrm{NH_4}}} \cdot \frac{\partial \mathrm{NH_4}}{\partial z}\big|_{z_\mathrm{ox}^-} - \gamma_{\mathrm{NH_4}} \cdot F_{\mathrm{NH_4}}(z_\mathrm{ox}) = -\frac{D_{\mathrm{NH_4}}}{1 + K_{\mathrm{NH_4}}} \cdot \frac{\partial \mathrm{NH_4}}{\partial z}\big|_{z_\mathrm{ox}^+}$ |
| | where: | | $F_{\mathrm{NH_4}}(z_\mathrm{ox}) = \frac{1}{1 + K_{\mathrm{NH_4}}} \cdot \frac{1 - \phi}{\phi} \cdot \int_{z_{\mathrm{NO_3}}}^{\infty} \sum_i \mathrm{NC_i} \cdot k_i \cdot \mathrm{POC}_i\, dz$ |
| $z = z_{\mathrm{NO_3}}$ | continuity | 6) | $\mathrm{NH_4}(z_{\mathrm{NO_3}}^-) = \mathrm{NH_4}(z_{\mathrm{NO_3}}^+)$ |
| | flux | 7) | $-\frac{D_{\mathrm{NH_4}}}{1 + K_{\mathrm{NH_4}}} \cdot \frac{\partial \mathrm{NH_4}}{\partial z}\big|_{z_{\mathrm{NO_3}}^-} = -\frac{D_{\mathrm{NH_4}}}{1 + K_{\mathrm{NH_4}}} \cdot \frac{\partial \mathrm{NH_4}}{\partial z}\big|_{z_{\mathrm{NO_3}}^+}$ |
| $z = z_\infty$ | zero NH$_4$ flux | 8) | $\frac{\partial \mathrm{NH_4}}{\partial z}\big|_{z_\infty} = 0$ |

### 2.2.4 Sulfate and Sulfide

Below the denitrification zone ($z > z_{\mathrm{NO_3}}$), organic matter degradation is coupled to sulfate reduction, consuming sulfate and producing hydrogen sulfide with a fixed $\mathrm{SO_4 : C}$ ratio ($\mathrm{SO_4C}$, Tab. 10). In addition, the anaerobic oxidation of upward diffusing methane (AOM) produced below the sulfate penetration and the associated consumption of sulfate and production of sulfide; as well as the production of sulfate and consumption of sulfide through sulfide oxidation are implicitly accounted for through the boundary conditions (Table 5). The conservation equations for sulfate and sulfide are thus given by:

1. Oxic and nitrogenous zone ($z \leq z_{\mathrm{NO_3}}$)

$$\frac{\partial \mathrm{SO_4}^I}{\partial t} = 0 = D_{\mathrm{SO_4}} \frac{\partial^2 \mathrm{SO_4}^I}{\partial z^2} - w \frac{\partial \mathrm{SO_4}^I}{\partial z} \tag{20}$$

$$\frac{\partial \mathrm{H_2S}^I}{\partial t} = 0 = D_{\mathrm{H_2S}} \frac{\partial^2 \mathrm{H_2S}^I}{\partial z^2} - w \frac{\partial \mathrm{H_2S}^I}{\partial z} \tag{21}$$



2. Sulfidic zone ($z_{\mathrm{NO_3}} < z \leq z_{\mathrm{SO_4}}$)

$$\frac{\partial \mathrm{SO_4}^{II}}{\partial t} = 0 = D_{\mathrm{SO_4}} \frac{\partial^2 \mathrm{SO_4}^{II}}{\partial z^2} - w \frac{\partial \mathrm{SO_4}^{II}}{\partial z} - \frac{1-\phi}{\phi} \cdot \sum_i \mathrm{SO_4C} \cdot k_i \cdot \mathrm{POC}_i(z) \tag{22}$$

$$\frac{\partial \mathrm{H_2S}^{II}}{\partial t} = 0 = D_{\mathrm{H_2S}} \frac{\partial^2 \mathrm{H_2S}^{II}}{\partial z^2} - w \frac{\partial \mathrm{H_2S}^{II}}{\partial z} + \frac{1-\phi}{\phi} \cdot \sum_i \mathrm{SO_4C} \cdot k_i \cdot \mathrm{POC}_i(z) \tag{23}$$

3. Methanic zone ($z_{\mathrm{SO_4}} < z \leq z_\infty$)

$$\frac{\partial \mathrm{H_2S}^{III}}{\partial t} = 0 = D_{\mathrm{H_2S}} \frac{\partial^2 \mathrm{H_2S}^{III}}{\partial z^2} - w \frac{\partial \mathrm{H_2S}^{III}}{\partial z} \tag{24}$$

where $D_{\mathrm{SO_4}}$ and $D_{\mathrm{H_2S}}$ denote the diffusion coefficients for $\mathrm{SO_4}$ and $\mathrm{H_2S}$ which depend on the bioturbation status of the respective geochemical zone (compare Section 2.3.1). Integration of Eq. (20) - (24) yields the analytical solution and Table 5 summarises the boundary conditions applied. OMEN-SED assumes known concentrations at the sediment-water interface and continuity across the bioturbation depth and the nitrate penetration depth. The reoxidation of reduced $\mathrm{H_2S}$ to $\mathrm{SO_4}$ is accounted for implicitly via the oxic-anoxic boundary condition for both species, while reduction of $\mathrm{SO_4}$ and the associated production of $\mathrm{H_2S}$ via AOM is accounted for through the respective boundary conditions at $z_{\mathrm{SO_4}}$. In case $z_{\mathrm{SO_4}} < z_\infty$, OMEN-SED assumes zero sulfate concentration at $z_{\mathrm{SO_4}}$ and its diffusive flux must equal the amount of methane produced below (with a methane to carbon ratio of MC); or, in case $z_{\mathrm{SO_4}} = z_\infty$, a zero diffusive flux condition for sulfate is considered. OMEN-SED iteratively solves for $z_{\mathrm{SO_4}}$ by first testing if there is sulfate left at $z_\infty$ (i.e. $\mathrm{SO_4}(z_\infty) > 0$) and, otherwise, by finding the root for the flux boundary condition 8.2 (Table 5). At the lower boundary $z_\infty$ zero diffusive flux of $\mathrm{H_2S}$ is considered.

### 2.2.5 Phosphate

The biogeochemical description of phosphorus (P) dynamics builds on earlier models developed by Slomp et al. (1996) and accounts for phosphorus recycling through organic matter degradation, adsorption onto sediments and iron(III) hydroxides (Fe-bound P), as well as carbonate fluorapatite (CFA or authigenic P) formation (see Figure 2 for a schematic overview of the sedimentary P cycle). In the oxic zone of the sediment, $\mathrm{PO_4}$ liberated through organic matter degradation can adsorb to iron(III) hydroxides forming Fe-bound P (or FeP, Slomp et al., 1998). Below the oxic zone, $\mathrm{PO_4}$ is not only produced via organic matter degradation but can also be released from the Fe-bound P pool due to the reduction of iron(III) hydroxides under anoxic conditions. Furthermore, in these zones phosphate concentrations build up and pore waters can thus become supersaturated with respect to carbonate fluorapatite, thus triggering the authigenic formation of CFA (Van Cappellen and Berner, 1988). Phosphorus bound in these authigenic minerals represents a permanent sink for reactive phosphorus (Slomp et al., 1996). As for ammonium, the adsorption of P to the sediment matrix is treated as an equilibrium processes, parameterised with dimensionless adsorption coefficients for the oxic and anoxic zone, respectively ($K_{\mathrm{PO_4}}^{\mathrm{ox}}$, $K_{\mathrm{PO_4}}^{\mathrm{anox}}$ Slomp et al., 1998). The sorption and desorption of P to iron(III) hydroxides as well as the authigenic fluorapatite formation are described as first-order reactions with rate constants $k_s$, $k_m$ and $k_a$, respectively (Table 10). The rate of the respective process is calculated as the product of the rate constant and the difference between the current concentration (of $\mathrm{PO_4}$ and FeP) and an equilibrium or asymptotic concentration Slomp et al. (1996). The asymptotic Fe-bound P concentration is $\mathrm{FeP}^\infty$ and the equilibrium





**Table 5.** Boundary conditions for sulfate and sulfide. For the boundaries we define: $z_-^- := \lim_{h \to 0}(z_- - h)$ and $z_-^+ := \lim_{h \to 0}(z_- + h)$.

| Boundary | Condition | | |
|---|---|---|---|
| $z = 0$ | known concentration | 1) | $\mathrm{SO_4}(0) = \mathrm{SO_{40}}$ |
| $z = z_{\mathrm{bio}}$ | continuity | 2) | $\mathrm{SO_4}(z_{\mathrm{bio}}^-) = \mathrm{SO_4}(z_{\mathrm{bio}}^+)$ |
| | flux | 3) | $-(D_{\mathrm{SO_4},0} + D_{\mathrm{bio}}) \cdot \frac{\partial \mathrm{SO_4}}{\partial z}\big|_{z_{\mathrm{bio}}^-} = -D_{\mathrm{SO_4},0} \cdot \frac{\partial \mathrm{SO_4}}{\partial z}\big|_{z_{\mathrm{bio}}^+}$ |
| $z = z_{\mathrm{ox}}$ | continuity | 4) | $\mathrm{SO_4}(z_{\mathrm{ox}}^-) = \mathrm{SO_4}(z_{\mathrm{ox}}^+)$ |
| | flux | 5) | $-D_{\mathrm{SO_4}} \cdot \frac{\partial \mathrm{SO_4}}{\partial z}\big|_{z_{\mathrm{ox}}^-} + \gamma_{\mathrm{H_2S}} \cdot F_{\mathrm{H_2S}}(z_{\mathrm{ox}}) = -D_{\mathrm{SO_4}} \cdot \frac{\partial \mathrm{SO_4}}{\partial z}\big|_{z_{\mathrm{ox}}^+}$ |
| | where: | | $F_{\mathrm{H_2S}}(z_{\mathrm{ox}}) = \frac{1-\phi}{\phi} \cdot \left( \int_{z_{\mathrm{NO_3}}}^{\mathrm{SO_4}} \sum_i \mathrm{SO_4C} \cdot k_i \cdot \mathrm{POC}_i \, dz + \gamma_{\mathrm{CH_4}} \cdot \int_{z_{\mathrm{SO_4}}}^{\infty} \sum_i \mathrm{MC} \cdot k_i \cdot \mathrm{POC}_i \, dz \right)$ |
| $z = z_{\mathrm{NO_3}}$ | continuity | 6) | $\mathrm{SO_4}(z_{\mathrm{NO_3}}^-) = \mathrm{SO_4}(z_{\mathrm{NO_3}}^+)$ |
| | flux | 7) | $-D_{\mathrm{SO_4}} \cdot \frac{\partial \mathrm{SO_4}}{\partial z}\big|_{z_{\mathrm{NO_3}}^-} = -D_{\mathrm{SO_4}} \cdot \frac{\partial \mathrm{SO_4}}{\partial z}\big|_{z_{\mathrm{NO_3}}^+}$ |
| $z = z_{\mathrm{SO_4}}$ | $\mathrm{SO_4}$ consumption | 8) | **IF** $(\mathrm{SO_4}(z_\infty) > 0)$ |
| | $(z_{\mathrm{SO_4}} = z_\infty)$ | 8.1) | $\frac{\partial SO_4}{\partial z}\big|_{z_{\mathrm{SO_4}}} = 0$ |
| | | | **ELSE** |
| | $(z_{\mathrm{SO_4}} < z_\infty)$ | 8.2) | $\mathrm{SO_4}(z_{\mathrm{SO_4}}) = 0 \quad$ and $\quad -D_{\mathrm{SO_4}} \cdot \frac{\partial \mathrm{SO_4}}{\partial z}\big|_{z_{\mathrm{SO_4}}} = \gamma_{\mathrm{CH_4}} \cdot F_{\mathrm{CH_4}}(z_{\mathrm{SO_4}})$ |
| | with | | $F_{\mathrm{CH_4}}(z_{\mathrm{SO_4}}) = \frac{1-\phi}{\phi} \cdot \int_{z_{\mathrm{SO_4}}}^{\infty} \sum_i \mathrm{MC} \cdot k_i \cdot \mathrm{POC}_i \, dz$ |
| $z = 0$ | known concentration | 1) | $\mathrm{H_2S}(0) = \mathrm{H_2S_0}$ |
| $z = z_{\mathrm{bio}}$ | continuity | 2) | $\mathrm{H_2S}(z_{\mathrm{bio}}^-) = \mathrm{H_2S}(z_{\mathrm{bio}}^+)$ |
| | flux | 3) | $-(D_{\mathrm{H_2S},0} + D_{\mathrm{bio}}) \cdot \frac{\partial \mathrm{H_2S}}{\partial z}\big|_{z_{\mathrm{bio}}^-} = -D_{\mathrm{H_2S},0} \cdot \frac{\partial \mathrm{H_2S}}{\partial z}\big|_{z_{\mathrm{bio}}^+}$ |
| $z = z_{\mathrm{ox}}$ | continuity | 4) | $\mathrm{H_2S}(z_{\mathrm{ox}}^-) = \mathrm{H_2S}(z_{\mathrm{ox}}^+)$ |
| | flux | 5) | $-D_{\mathrm{H_2S}} \cdot \frac{\partial \mathrm{H_2S}}{\partial z}\big|_{z_{\mathrm{ox}}^-} - \gamma_{\mathrm{H_2S}} F_{\mathrm{H_2S}}(z_{\mathrm{ox}}) = -D_{\mathrm{H_2S}} \cdot \frac{\partial \mathrm{H_2S}}{\partial z}\big|_{z_{\mathrm{ox}}^+}$ |
| | where: | | $F_{\mathrm{H_2S}}(z_{\mathrm{ox}}) = \frac{1-\phi}{\phi} \cdot \left( \int_{z_{\mathrm{NO_3}}}^{\mathrm{SO_4}} \sum_i \mathrm{SO_4C} \cdot k_i \cdot \mathrm{POC}_i \, dz + \gamma_{\mathrm{CH_4}} \cdot \int_{z_{\mathrm{SO_4}}}^{\infty} \sum_i \mathrm{MC} \cdot k_i \cdot \mathrm{POC}_i \, dz \right)$ |
| $z = z_{\mathrm{NO_3}}$ | continuity | 6) | $\mathrm{H_2S}(z_{\mathrm{NO_3}}^-) = \mathrm{H_2S}(z_{\mathrm{NO_3}}^+)$ |
| | flux | 7) | $-D_{\mathrm{H_2S}} \cdot \frac{\partial \mathrm{H_2S}}{\partial z}\big|_{z_{\mathrm{NO_3}}^-} = -D_{\mathrm{H_2S}} \cdot \frac{\partial \mathrm{H_2S}}{\partial z}\big|_{z_{\mathrm{NO_3}}^+}$ |
| $z = z_{\mathrm{SO_4}}$ | continuity | 8) | $\mathrm{H_2S}(z_{\mathrm{SO_4}}^-) = \mathrm{H_2S}(z_{\mathrm{SO_4}}^+)$ |
| | flux (with AOM) | 9) | $-D_{\mathrm{H_2S}} \cdot \frac{\partial \mathrm{H_2S}}{\partial z}\big|_{z_{\mathrm{SO_4}}^-} + \gamma_{\mathrm{CH_4}} \cdot F_{\mathrm{CH_4}}(z_{\mathrm{SO_4}}) = -D_{\mathrm{H_2S}} \cdot \frac{\partial \mathrm{H_2S}}{\partial z}\big|_{z_{\mathrm{SO_4}}^+}$ |
| | where: | | $F_{\mathrm{CH_4}}(z_{\mathrm{SO_4}}) = \frac{1-\phi}{\phi} \cdot \int_{z_{\mathrm{SO_4}}}^{\infty} \sum_i \mathrm{MC} \cdot k_i \cdot \mathrm{POC}_i \, dz$ |
| $z = z_\infty$ | zero $\mathrm{H_2S}$ flux | 10) | $\frac{\partial \mathrm{H_2S}}{\partial z}\big|_{z_\infty} = 0$ |



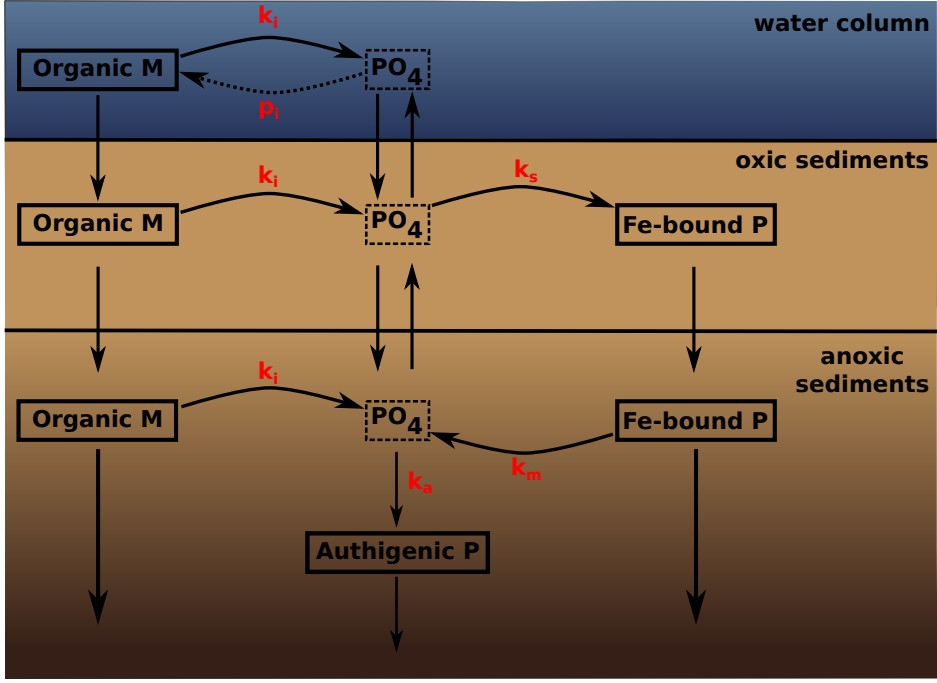

**Figure 2.** A schematic of the sedimentary P cycle in OMEN-SED. Red numbers represent kinetic rate constants for phosphorus dynamics (compare Table 10; $p_i$ represents uptake rate of $PO_4$ via primary production in shallow environments). Adapted from Slomp et al. (1996).

concentration for P sorption and authigenic fluorapatite formation are $PO_4{}^s$ and $PO_4{}^a$, respectively (Table 10). The last term in Eq. (25) and (26) represents sorption of $PO_4$ to FeP in the oxic zone, the last term in Eq. (27) and (28) is the release of $PO_4$ from the FeP pool and the 4th term in Eq. (28) represents the permanent loss of $PO_4$ to authigenic fluorapatite formation. The conservation equations for phosphate and Fe-bound P are thus given by:

5    1. Oxic zone ($z \leq z_{\text{ox}}$)

$$\frac{\partial PO_4{}^I}{\partial t} = \frac{D_{PO_4}}{1+K_{PO_4}^{\text{ox}}}\frac{\partial^2 PO_4{}^I}{\partial z^2} - w\frac{\partial PO_4{}^I}{\partial z} + \frac{1-\phi}{\phi}\frac{1}{1+K_{PO_4}^{\text{ox}}}\sum_i (PC_i \cdot k_i \cdot POC_i(z)) - \frac{k_s}{1+K_{PO_4}^{\text{ox}}}(PO_4{}^I - PO_4{}^s) \quad (25)$$

$$\frac{\partial FeP^I}{\partial t} = D_{FeP}\frac{\partial^2 FeP^I}{\partial z^2} - w\frac{\partial FeP^I}{\partial z} + \frac{\phi}{1-\phi}k_s(PO_4{}^I - PO_4{}^s) \quad (26)$$

2. Anoxic zones ($z_{\text{ox}} < z \leq z_{\infty}$)

$$\frac{\partial FeP^{II}}{\partial t} = D_{FeP}\frac{\partial^2 FeP^{II}}{\partial z^2} - w\frac{\partial FeP^{II}}{\partial z} - k_m(FeP^{II} - FeP^{\infty}) \quad (27)$$

$$\frac{\partial PO_4{}^{II}}{\partial t} = \frac{D_{PO_4}}{1+K_{PO_4}^{\text{anox}}}\frac{\partial^2 PO_4{}^{II}}{\partial z^2} - w\frac{\partial PO_4{}^{II}}{\partial z} + \frac{1-\phi}{\phi}\frac{1}{1+K_{PO_4}^{\text{anox}}}\sum_i (PC_i \cdot k_i \cdot POC_i(z))$$

$$- \frac{k_a}{1+K_{PO_4}^{\text{anox}}}(PO_4{}^{II} - PO_4{}^a) + \frac{(1-\phi)}{\phi}\frac{k_m}{1+K_{PO_4}^{\text{anox}}}(FeP^{II} - FeP^{\infty}) \quad (28)$$





**Table 6.** Boundary conditions for phosphate and Fe-bound P (FeP). For the boundaries we define: $z_-^- := \lim_{h\to 0}(z_- - h)$ and $z_-^+ := \lim_{h\to 0}(z_- + h)$.

| Boundary | Condition | |
|---|---|---|
| $z = 0$ | known concentration | 1) $PO_4(0) = PO_{40}$ |
| $z = z_{bio}$ | continuity | 2) $PO_4(z_{bio}^-) = PO_4(z_{bio}^+)$ |
| | flux | 3) $(D_{PO_4,0} + D_{bio}) \cdot \frac{\partial PO_4}{\partial z}\big|_{z_{bio}^-} = D_{PO_4,0} \cdot \frac{\partial PO_4}{\partial z}\big|_{z_{bio}^+}$ |
| $z = z_{ox}$ | continuity | 4) $PO_4(z_{ox}^-) = PO_4(z_{ox}^+)$ |
| | flux | 5) $-\frac{D_{PO_4}}{1 + K_{PO_4}^{ox}} \cdot \frac{\partial PO_4}{\partial z}\big|_{z_{ox}^-} = -\frac{D_{PO_4}}{1 + K_{PO_4}^{anox}} \cdot \frac{\partial PO_4}{\partial z}\big|_{z_{ox}^+}$ |
| $z = z_\infty$ | flux | 10) $\frac{\partial PO_4}{\partial z}\big|_{z_\infty} = 0$ |
| $z = 0$ | known concentration | 1) $FeP(0) = FeP_0$ |
| $z = z_{bio}$ | continuity | 2) $FeP(z_{bio}^-) = FeP(z_{bio}^+)$ |
| | flux | 3) $\frac{\partial FeP}{\partial z}\big|_{z_{bio}^-} = \frac{\partial FeP}{\partial z}\big|_{z_{bio}^+}$ |
| $z = z_{ox}$ | continuity | 4) $FeP(z_{ox}^-) = FeP(z_{ox}^+)$ |
| | flux | 5) $\frac{\partial FeP}{\partial z}\big|_{z_{ox}^-} = \frac{\partial FeP}{\partial z}\big|_{z_{ox}^+}$ |
| $z = z_\infty$ | asymptotic concentration | 10) $FeP(z_\infty) = FeP_\infty$ |

where $D_{PO_4}$ denotes the diffusion coefficient for $PO_4$ which depends on the bioturbation status of the respective geochemical zone and $D_{FeP} = D_{bio}$ for $z \leq z_{bio}$ and $D_{FeP} = 0$ for $z > z_{bio}$ (compare Section 2.3.1). Integration of Eq. (25) - (28) yields the analytical solution and Table 6 summarises the boundary conditions applied in OMEN-SED. The model assumes known bottom water concentrations and equal concentrations and diffusive fluxes at $z_{bio}$ and $z_{ox}$ for both species. Additionally OMEN-SED considers no change in phosphate flux and an asymptotic Fe-bound P concentration at $z_\infty$.

### 2.2.6 Dissolved Inorganic Carbon (DIC)

OMEN-SED accounts for the production of dissolved inorganic carbon (DIC) through organic matter degradation, as well as methane oxidation. Organic matter degradation produces dissolved inorganic carbon with a stoichiometric DIC : C ratio of 1:2 in the methanic zone and 1:1 in the rest of the sediment column ($DICC^{II}$ and $DICC^I$ respectively). DIC production through methane oxidation is implicitly taken into account through the boundary condition at $z_{SO_4}$. A mechanistic description of DIC production from $CaCO_3$ dissolution would lead to significant mathematical problems and is therefore not included in the current version of OMEN-SED. The conservation equations for DIC are thus given by:

1. Oxic, nitrogenous and sulfidic zone ($z \leq z_{SO_4}$)

$$\frac{\partial DIC^I}{\partial t} = 0 = D_{DIC} \frac{\partial^2 DIC^I}{\partial z^2} - w \frac{\partial DIC^I}{\partial z} + \frac{1-\phi}{\phi} \cdot \sum_i DICC^I \cdot k_i \cdot POC_i(z) \tag{29}$$



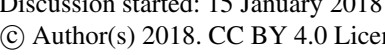



**Table 7.** Boundary conditions for DIC. For the boundaries we define: $z_-^- := \lim_{h\to 0}(z_- - h)$ and $z_-^+ := \lim_{h\to 0}(z_- + h)$.

| Boundary | Condition | | |
|---|---|---|---|
| $z = 0$ | known concentration | 1) | $\mathrm{DIC}(0) = \mathrm{DIC}_0$ |
| $z = z_{\mathrm{bio}}$ | continuity | 2) | $\mathrm{DIC}(z_{\mathrm{bio}}^-) = \mathrm{DIC}(z_{\mathrm{bio}}^+)$ |
| | flux | 3) | $-(D_{\mathrm{DIC},0} + D_{\mathrm{bio}}) \cdot \frac{\partial \mathrm{DIC}}{\partial z}\big|_{z_{\mathrm{bio}}^-} = -D_{\mathrm{DIC},0} \cdot \frac{\partial \mathrm{DIC}}{\partial z}\big|_{z_{\mathrm{bio}}^+}$ |
| $z = z_{\mathrm{SO}_4}$ | continuity | 4) | $\mathrm{DIC}(z_{\mathrm{SO}_4}^-) = \mathrm{DIC}(z_{\mathrm{SO}_4}^+)$ |
| | flux (with AOM) | 5) | $-D_{\mathrm{DIC}} \cdot \frac{\partial \mathrm{DIC}}{\partial z}\big|_{z_{\mathrm{SO}_4}^-} + \gamma_{\mathrm{CH}_4} \cdot F_{\mathrm{CH}_4}(z_{\mathrm{SO}_4}) = -D_{\mathrm{DIC}} \cdot \frac{\partial \mathrm{DIC}}{\partial z}\big|_{z_{\mathrm{SO}_4}^+}$ |
| | where: | | $F_{\mathrm{CH}_4}(z_{\mathrm{SO}_4}) = \frac{1-\phi}{\phi} \cdot \int_{z_{\mathrm{SO}_4}}^{\infty} \sum_i \mathrm{MC} \cdot k_i \cdot \mathrm{POC}_i \, dz$ |
| $z = z_\infty$ | zero DIC flux | 6) | $\frac{\partial \mathrm{DIC}}{\partial z}\big|_{z_\infty} = 0$ |

2. Methanic zone ($z_{\mathrm{SO}_4} < z \leq z_\infty$)

$$\frac{\partial \mathrm{DIC}^{II}}{\partial t} = 0 = D_{\mathrm{DIC}} \frac{\partial^2 \mathrm{DIC}^{II}}{\partial z^2} - w \frac{\partial \mathrm{DIC}^{II}}{\partial z} + \frac{1-\phi}{\phi} \cdot \sum_i \mathrm{DICC}^{II} \cdot k_i \cdot \mathrm{POC}_i(z) \qquad (30)$$

where $D_{\mathrm{DIC}}$ denotes the diffusion coefficient for DIC (taking the values for $\mathrm{HCO}_3^-$ from Schulz (2006)) which depends on the bioturbation status of the respective geochemical zone. Integration of Eq. (29) and (30) yields the analytical solution and

Table 7 summarises the boundary conditions applied in OMEN-SED. A Dirichlet condition is applied at the sediment-water interface. In addition, the model assumes a zero diffusive flux through the lower boundary $z_\infty$ and continuity across the bottom of the bioturbated zone, as well as the sulfate penetration depth. An additional flux boundary condition at $z_{\mathrm{SO}_4}$, implicitly accounts for DIC production through anaerobic oxidation of methane (Table 7 Eq. 5).

### 2.2.7 Alkalinity

Organic matter degradation and secondary redox reactions exert a complex influence on alkalinity (e.g. Jourabchi et al., 2005; Wolf-Gladrow et al., 2007; Krumins et al., 2013). To model alkalinity, OMEN-SED divides the sediment column is into four geochemical zones, where different equations describe the biogeochemical processes using variable stoichiometric coefficients (compare values in Table 10). Above $z_{\mathrm{ox}}$, the combined effects of $\mathrm{NH}_4$ and P release due to aerobic OM degradation increases alkalinity according to $\mathrm{ALK}^{\mathrm{OX}}$ whereas nitrification decreases alkalinity with stoichiometry $\mathrm{ALK}^{\mathrm{NIT}}$. In the remaining three

zones anaerobic OM degradation generally results in an increase in alkalinity, with the exact magnitude depending on the nature of the terminal electron acceptor used (i.e. $\mathrm{ALK}^{\mathrm{DEN}}$, $\mathrm{ALK}^{\mathrm{SUL}}$, $\mathrm{ALK}^{\mathrm{MET}}$). In addition, the effect of secondary redox reactions, such as nitrification, sulfide and methane oxidation are implicitly accounted for in the boundary conditions. Note that the alkalinity description in the current version of OMEN-SED does not account for $\mathrm{CaCO}_3$ dissolution/precipitation due to the mathematical complexity of the problem. In OMEN-SED, the conservation equations for alkalinity are thus given by:

1. Oxic zone ($z \leq z_{\mathrm{ox}}$)

$$\frac{\partial \mathrm{ALK}^I}{\partial t} = 0 = D_{\mathrm{ALK}} \frac{\partial^2 \mathrm{ALK}^I}{\partial z^2} - w \frac{\partial \mathrm{ALK}^I}{\partial z} + \frac{1-\phi}{\phi} \cdot \sum_i \left( \mathrm{ALK}^{\mathrm{NIT}} \cdot \frac{\gamma_{\mathrm{NH}_4}}{1 + K_{\mathrm{NH}_4}} \mathrm{NC_i} + \mathrm{ALK}^{\mathrm{OX}} \right) \cdot k_i \cdot \mathrm{POC}_i(z) \qquad (31)$$



**Table 8.** Boundary conditions for alkalinity. For the boundaries we define: $z_-^- := \lim_{h\to 0}(z_- - h)$ and $z_-^+ := \lim_{h\to 0}(z_- + h)$.

| Boundary | Condition | | |
|---|---|---|---|
| $z = 0$ | known concentration | 1) | $\mathrm{ALK}(0) = \mathrm{ALK}_0$ |
| $z = z_\mathrm{bio}$ | continuity | 2) | $\mathrm{ALK}(z_\mathrm{bio}^-)=\mathrm{ALK}(z_\mathrm{bio}^+)$ |
| | flux | 3) | $-(D_{\mathrm{ALK},0} + D_\mathrm{bio}) \cdot \frac{\partial \mathrm{ALK}}{\partial z}\big|_{z_\mathrm{bio}^-} = -D_{\mathrm{ALK},0} \cdot \frac{\partial \mathrm{ALK}}{\partial z}\big|_{z_\mathrm{bio}^+}$ |
| $z = z_\mathrm{ox}$ | continuity | 4) | $\mathrm{ALK}(z_\mathrm{ox}^-)=\mathrm{ALK}(z_\mathrm{ox}^+)$ |
| | flux | 5) | $-D_\mathrm{ALK} \cdot \frac{\partial \mathrm{ALK}}{\partial z}\big|_{z_\mathrm{ox}^-} + F_\mathrm{ALK}(z_\mathrm{ox}) = -D_\mathrm{ALK} \cdot \frac{\partial \mathrm{ALK}}{\partial z}\big|_{z_\mathrm{ox}^+}$ |
| | where: | | $F_\mathrm{ALK}(z_\mathrm{ox}) = \frac{1-\phi}{\phi} \cdot \left( \mathrm{ALK}^{\mathrm{H_2S}} \cdot \gamma_{\mathrm{H_2S}} \int_{z_{\mathrm{NO_3}}}^{\mathrm{SO_4}} \sum_i \mathrm{SO_4C} \cdot k_i \cdot \mathrm{POC}_i \, dz \right)$ |
| | | | $+ \frac{1-\phi}{\phi} \cdot \left( \mathrm{ALK}^{\mathrm{NIT}} \frac{\gamma_{\mathrm{NH_4}}}{1+k_{\mathrm{NH_4}}} \int_{z_{\mathrm{NO_3}}}^{\infty} \sum_i \mathrm{NC}_i \cdot k_i \cdot \mathrm{POC}_i \, dz \right)$ |
| $z = z_{\mathrm{NO_3}}$ | continuity | 6) | $\mathrm{ALK}(z_{\mathrm{NO_3}}^-)=\mathrm{ALK}(z_{\mathrm{NO_3}}^+)$ |
| | flux | 7) | $-D_\mathrm{ALK} \cdot \frac{\partial \mathrm{ALK}}{\partial z}\big|_{z_{\mathrm{NO_3}}^-} = -D_\mathrm{ALK} \cdot \frac{\partial \mathrm{ALK}}{\partial z}\big|_{z_{\mathrm{NO_3}}^+}$ |
| $z = z_{\mathrm{SO_4}}$ | continuity | 8) | $\mathrm{ALK}(z_{\mathrm{SO_4}}^-)=\mathrm{ALK}(z_{\mathrm{SO_4}}^+)$ |
| | flux (with AOM) | 9) | $-D_\mathrm{ALK} \cdot \frac{\partial \mathrm{ALK}}{\partial z}\big|_{z_{\mathrm{SO_4}}^-} + F_\mathrm{ALK}(z_{\mathrm{SO_4}}) = -D_\mathrm{ALK} \cdot \frac{\partial \mathrm{ALK}}{\partial z}\big|_{z_{\mathrm{SO_4}}^+}$ |
| | where: | | $F_\mathrm{ALK}(z_{\mathrm{SO_4}}) = \frac{1-\phi}{\phi} \cdot \left( \mathrm{ALK}^{\mathrm{AOM}} \gamma_{\mathrm{CH_4}} \cdot \int_{z_{\mathrm{SO_4}}}^{\infty} \sum_i k_i \cdot \mathrm{POC}_i \, dz \right)$ |
| $z = z_\infty$ | zero ALK flux | 10) | $\frac{\partial \mathrm{ALK}}{\partial z}\big|_{z_\infty} = 0$ |

2. Denitrification or nitrogenous zone ($z_\mathrm{ox} < z \le z_{\mathrm{NO_3}}$)

$$\frac{\partial \mathrm{ALK}^{II}}{\partial t} = 0 = D_\mathrm{ALK} \frac{\partial^2 \mathrm{ALK}^{II}}{\partial z^2} - w \frac{\partial \mathrm{ALK}^{II}}{\partial z} + \frac{1-\phi}{\phi} \cdot \sum_i \mathrm{ALK}^{\mathrm{DEN}} \cdot k_i \cdot \mathrm{POC}_i(z) \tag{32}$$

3. Sulfidic zone ($z_{\mathrm{NO_3}} < z \le z_{\mathrm{SO_4}}$)

$$\frac{\partial \mathrm{ALK}^{III}}{\partial t} = 0 = D_\mathrm{ALK} \frac{\partial^2 \mathrm{ALK}^{III}}{\partial z^2} - w \frac{\partial \mathrm{ALK}^{III}}{\partial z} + \frac{1-\phi}{\phi} \cdot \sum_i \mathrm{ALK}^{\mathrm{SUL}} \cdot k_i \cdot \mathrm{POC}_i(z) \tag{33}$$

5   4. Methanic zone ($z_{\mathrm{SO_4}} < z \le z_\infty$)

$$\frac{\partial \mathrm{ALK}^{IV}}{\partial t} = 0 = D_\mathrm{ALK} \frac{\partial^2 \mathrm{ALK}^{IV}}{\partial z^2} - w \frac{\partial \mathrm{ALK}^{IV}}{\partial z} + \frac{1-\phi}{\phi} \cdot \sum_i \mathrm{ALK}^{\mathrm{MET}} \cdot k_i \cdot \mathrm{POC}_i(z) \tag{34}$$

where $D_\mathrm{ALK}$ denotes the diffusion coefficient for alkalinity (taking the values for $\mathrm{HCO_3^-}$ from Schulz (2006)) which depends on the bioturbation status of the respective geochemical zone. Integration of Eq. (31) - (34) yields the analytical solution and Table 8 summarises the boundary conditions applied in OMEN-SED. A Dirichlet boundary condition is applied at the

10   sediment-water interface. The decrease of alkalinity due to oxidation of reduced species produced in the anoxic zones (with stoichiometry $\mathrm{ALK}^{\mathrm{NIT}}$ and $\mathrm{ALK}^{\mathrm{H_2S}}$) is implicitly taken into account through the flux boundary condition at $z_\mathrm{ox}$ (Table 8 Eq. 5). Furthermore, the oxidation of methane by sulfate reduction increases alkalinity with stoichiometry $\mathrm{ALK}^{\mathrm{AOM}}$ which is accounted for through the flux boundary condition at $z_{\mathrm{SO_4}}$ (Table 8 Eq. 9). At the lower boundary $z_\infty$ a zero diffusive flux condition is applied.



## 2.3 Determination of Integration Constants

The integration constants of all general analytical solutions derived above change in response to changing boundary conditions. Thus, OMEN-SED has to re-determine integration constants for each dynamic zone (i.e. $z_{\mathrm{ox}}, z_{\mathrm{bio}}, z_{\mathrm{NO_3}}$ and $z_{\mathrm{SO_4}}$) at every time step for all biogeochemical species. The bioturbation boundary poses a particular challenge as it can theoretically occur in any

of the dynamic geochemical zones (Fig. 3). Therefore, in order to generalise and simplify this recurring boundary matching problem, an independent, generic algorithm (Generic Boundary Condition Matching) is implemented (rather than using multiple fully-worked-out algebraic solutions for each possible case and every biogeochemical species). As a consequence, the algorithm only has to solve a two-simultaneous-equation problem.

### 2.3.1 Generic Boundary Condition Matching (GBCM)

As discussed in Section 2.1, the solution of the general steady-state transport-reaction equation (Eq. (2)) for a generic species $C$ is of the general form:

$$C(z) = A \cdot \exp(az) + B \cdot \exp(bz) + \sum_j \frac{\alpha_j}{D\beta_j^2 - w\beta_j - k} \cdot \exp(-\beta_j z) + \frac{Q}{k} \tag{35}$$

and can therefore be expressed as:

$$C(z) = A \cdot E(z) + B \cdot F(z) + G(z) \tag{36}$$

where $E(z)$, $F(z)$ are the homogeneous solutions of the ODE, G(z) the particular integral (collectively called the basis functions), and A, B are the integration constants that must be determined using the boundary conditions (shown in Fig. 3 for the whole sediment column).

Each internal boundary matching problem (i.e. excluding $z = 0$ and $z = z_\infty$) involves matching continuity and flux for the two solutions of the respective reaction-transport equation above, $C_U(z)$ (= 'upper'), and below, $C_L(z)$ (= 'lower'), the

dynamic boundary at $z = z_b$:

$$C_U(z) = A_U \cdot E_U(z) + B_U \cdot F_U(z) + G_U(z) \tag{37}$$
$$C_L(z) = A_L \cdot E_L(z) + B_L \cdot F_L(z) + G_L(z). \tag{38}$$

OMEN-SED generally applies concentration continuity and flux boundary conditions at its internal, dynamic boundaries:
Continuity (where for generality we allow a discontinuity $V_b$)

$$C_U(z_b) = C_L(z_b) + V_b \tag{39}$$

Flux

$$D_U C_U'(z_b) + wC_U(z_b) = D_L C_L'(z_b) + wC_L(z_b) + F_b \tag{40}$$





where $w$ is advection, $D$ are the diffusion coefficients and $F_b$ is any flux discontinuity (e.g. resulting from secondary redox reactions).

Considering that the advective flux above and below the boundary is equal (i.e. $wC_U(z_b) = wC_L(z_b)$) and substituting the general ODE solutions (37), (38), the boundary conditions can be represented as two equations connecting the four integration

constants:

$$
\begin{pmatrix} E_U & F_U \\ D_U E_U' & D_U F_U' \end{pmatrix} \begin{pmatrix} A_U \\ B_U \end{pmatrix} = \begin{pmatrix} E_L & F_L \\ D_L E_L' & D_L F_L' \end{pmatrix} \begin{pmatrix} A_L \\ B_L \end{pmatrix} + \begin{pmatrix} G_L - G_U + V_b \\ D_L G_L' - D_U G_U' + F_b - wV_b \end{pmatrix} \tag{41}
$$

where the ODE solutions $E$, $F$, $G$ are all evaluated at $z_b$.

Equation (41) can now be solved to give $A_U$ and $B_U$ as a function of the integration constants from the layer below ($A_L$ and $B_L$), thereby constructing a piecewise solution for both layers, with just two integration constants (this is implemented in the

function **benthic_utils.matchsoln** of OMEN-SED):

$$
\begin{pmatrix} A_U \\ B_U \end{pmatrix} = \begin{pmatrix} c_1 & c_2 \\ c_3 & c_4 \end{pmatrix} \begin{pmatrix} A_L \\ B_L \end{pmatrix} + \begin{pmatrix} d_1 \\ d_2 \end{pmatrix}. \tag{42}
$$

Using Eq. (42), $C_U(z)$ in (37) can now be rewritten as a function of $A_L$ and $B_L$ (implemented in **benthic_utils.xformsoln**)):

$$
C_U(z) = (c_1 A_L + c_2 B_L + d_1) \cdot E_U(z) + (c_3 A_L + c_4 B_L + d_2) \cdot F_U(z) + G_U(z) \tag{43}
$$

and hence define the "transformed" basis functions $E_U^*(z)$, $F_U^*(z)$, $G_U^*(z)$ such that:

$$
C_U(z) = A_L \cdot E_U^*(z) + B_L \cdot F_U^*(z) + G_U^*(z) \tag{44}
$$

where

$$
\begin{aligned}
E_U^*(z) &= c_1 E_U(z) + c_3 F_U(z) \\
F_U^*(z) &= c_2 E_U(z) + c_4 F_U(z) \\
G_U^*(z) &= G_U(z) + d_1 E_U(z) + d_2 F_U(z)
\end{aligned} \tag{45}
$$

Equations (42), (44) and (45) can now be consecutively applied for each of the dynamic biogeochemical zone boundaries (Fig. 3), starting at the bottom of the sediment column. The net result is a piecewise solution of the whole sediment column with just two integration constants (coming from the lowest layer), which can then be solved for by applying the boundary conditions at the sediment-water interface and the bottom of the sediments.

### 2.3.2   Abstracting out the bioturbation boundary

The bioturbation boundary affects the diffusion coefficient of the modelled solutes, as well as the conservation equation of organic matter (and thereby the exact form of each reaction-transport equation). This boundary is particularly inconvenient as it





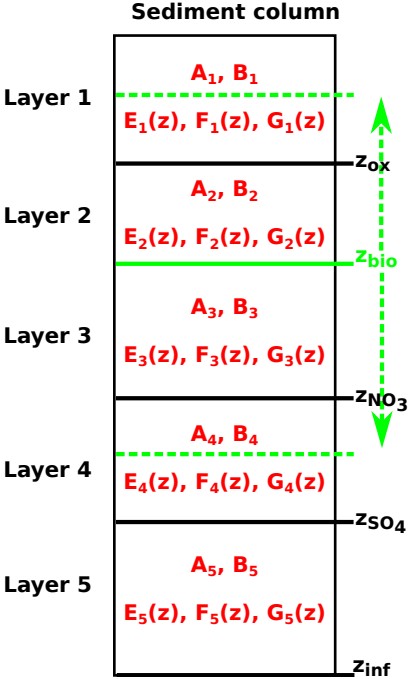

**Figure 3.** Schematic of the generic boundary condition matching (GBCM) problem. Showing the resulting integration constants ($A_i$, $B_i$) and ODE solutions ($E_i$, $F_i$, $G_i$) for the different sediment layers and the variable bioturbation boundary.

can, in principle, occur in the middle of any of the dynamically shifting biogeochemical zones and therefore generate multiple cases (Fig. 3). The GBCM algorithm described above is thus not only used to construct a piecewise solution of the whole sediment column, but also to abstract out the bioturbation boundary. For each biogeochemical zone the "bioturbation-status" is initially tested (i.e. fully bioturbated, fully non-bioturbated, or crossing the bioturbation boundary). Therefore, the upper and lower boundaries for the different zones (e.g. for the nitrogenous zone: $z_U = z\mathrm{ox}$, $z_L = z_{\mathrm{NO}_3}$), as well as the respective reactive terms and diffusion coefficients (bioturbated and non-bioturbated) are passed over to the routine **zTOC.prepfg_l12** where the bioturbation-status is determined. In case the bioturbation depth is located within this zone (i.e. $z_U < z_{\mathrm{bio}} < z_L$) a piecewise solution for this layer is constructed. Therefore, the reactive terms and diffusion coefficients are handed over to the routines **zTOC.calcfg_l1** and **zTOC.calcfg_l2** which calculate the basis functions ($E_U, F_U, G_U$ and $E_L, F_L, G_L$) and their derivatives for the bioturbated and the non-bioturbated part of this specific geochemical zone. The concentration and flux for both solutions at $z_{\mathrm{bio}}$ are matched and the coefficients $c_1, c_2, c_3, c_4, d_1, d_2$ (as in Eq. (42)) are calculated by the routine **benthic_utils.matchsoln**. These coefficients and the "bioturbation-status" of the layer are passed back to the main GBCM algorithm where they can be used by the routine **benthic_utils.xformsoln** to calculate the "transformed" basis functions ($E_U^*(z), F_U^*(z), G_U^*(z)$) such that both layers are expressed in the same basis (compare Eq. (43 - 45)).





For instance, in the case of sulfate, **zTOC.prepfg_l12** is called three times before the actual profile is calculated (once per zone: oxic, nitrogenous, sulfidic) and hands back the information about the "bioturbation-status" of the three layers and the coefficients $c_1, c_2, c_3, c_4, d_1, d_2$ for the biogeochemical zone including the bioturbation depth. When calculating the complete piecewise solution for the sediment column, this information is passed to the function **zTOC.calcfg_l12** which sorts out the
correct solution type to use. The main GBCM algorithm therefore never needs to know whether it is dealing with a piecewise solution (i.e. matched across the bioturbation boundary) or a "simple" solution (i.e. the layer is fully bioturbated or fully non-bioturbated).

## 2.4 Model Parameters

The following section provides a summary of global relationships used to constrain reaction and transport parameters in
OMEN-SED. Table 9 synthesises sediment and transport parameters, while Tab. 10 provides an overview of all biogeochemical parameters used in OMEN-SED.

### 2.4.1 Transport Parameters

The burial of sediments and pore water is directly related to the accumulation of new material on the seafloor (i.e. sedimentation, Burdige, 2006). This results in a downward advective flux of older sediment material and pore water in relation to the sediment-
water interface. When coupled to an ocean model, its sedimentation flux can be readily used in OMEN-SED. The stand-alone version of OMEN-SED uses the empirical global relationship between sediment accumulation rate ($w$ in cm yr$^{-1}$) and seafloor depth ($z$ in m) of Middelburg et al. (1997):

$$w = 3.3 \cdot 10^{-0.87478367 - 0.00043512 \cdot z}. \tag{46}$$

As an option we include the parameterisation of Burwicz et al. (2011)

$$w = \frac{w_1}{1 + (\frac{z}{z1})^{c1}} + \frac{w_2}{1 + (\frac{z}{z2})^{c2}} \tag{47}$$

with parameter values as found in the original study (i.e. $w_1 = 0.117$ cm yr$^{-1}$, $w_2 = 0.006$ cm yr$^{-1}$, $z1 = 200$ m, $z2 = 4000$ m, $c1 = 3$, $c2 = 10$). As mentioned before (Section 2.1), the diffusion coefficient of species $i$ is calculated as $D_i = D_{i,0} + D_{bio} = D_{mol,i} \cdot f_{ir} + D_{bio}$ for dissolved species and $D_i = D_{bio}$ for solid species. The bioturbation coefficient $D_{bio}$ (cm$^2$ yr$^{-1}$) is constant in the bioturbated zone and also follows the empirical relationship by Middelburg et al. (1997):

$$D_{bio} = 5.2 \cdot 10^{0.76241122 - 0.00039724 \cdot z} \tag{48}$$

Observations indicate that bioturbation is largely restricted to the upper 10 cm of the sediments and is only marginally related to seafloor depth (e.g. Boudreau, 1998; Teal et al., 2010). Therefore, OMEN-SED imposes a globally invariant bioturbation depth $z_{bio}$ of 10 cm. In case the bottom water oxygen concentration is low (here $<4.5$ nmol cm$^{-3}$ which is often used to define suboxic waters, e.g. Morrison et al., 1999; Karstensen et al., 2008) infaunal activity is assumed to cease and $z_{bio} = 0.01$ cm.





We choose a low value unequal to zero in order to simplify the implementation of the model. This approach ensures that the sediment column always consists of a bioturbated (even though very small for the low oxygen condition) and a non-bioturbated zone, thus the same GBCM algorithm can be used to solve the conservation equations. Furthermore, when OMEN-SED is coupled to an Earth system model the same method can be used to convert the POC depositional flux into a SWI concentration
(i.e. the flux needs to be converted assuming bioturbation, see Section 4.1).

Bioirrigation (i.e the pumping activity by burrow-dwelling animals) exchanges burrow water with overlying water and may enhance the SWI-flux of solutes (Aller, 1984, 1988). Several approaches exist to incorporate this into a 1-D diagenetic model, for instance as a non-local transport/exchange process (Boudreau, 1984; Emerson et al., 1984) or as an enhancement factor of the molecular diffusion coefficient (Devol and Christensen, 1993; Soetaert et al., 1996). In OMEN-SED the latter approach
is applied and the apparent "bio-diffusion" coefficient is calculated as $D_{i,0} = D_{\mathrm{mol},i} \cdot f_{ir}$. Soetaert et al. (1996) derived an empirical relationship between $f_{ir}$ and seafloor depth ($f_{ir} = \mathrm{Min}\{1; 15.9 \cdot z^{-0.43}\}$) based on observations from Archer and Devol (1992) and Devol and Christensen (1993). As this relationship just varies for depth above ∼623 m (with a maximum value of 3 at ∼50 m) a constant value of $f_{ir} = 1$ is used in the default OMEN-SED configuration. The specific molecular diffusion coefficients $D_{\mathrm{mol},i}$ are corrected for sediment porosity $\phi$, tortuosity F and are linearly interpolated for an ambient
temperature $T$ (in °C) using zero-degree coefficients $D_i^0$ and temperature-dependent diffusion coefficients $D_i^T$ (Soetaert et al., 1996):

$$D_{\mathrm{mol},i} = (D_i^0 + D_i^T \cdot T) \cdot \frac{1}{\phi \cdot F}.$$

Tortuosity can be expressed in terms of porosity as $F = \frac{1}{\phi^m}$ (Ullman and Aller, 1982) with the exponent $m$ varying according to the type of sediment (here $m = 3$ is used representing muddy sediments with high porosity). Values for $D_i^T$ and $D_i^0$ are
summarised in Table 9 and are adapted from Li and Gregory (1974), Schulz (2006) and Gypens et al. (2008).

### 2.4.2 Stoichiometries and reaction parameters

The first-order organic matter degradation constants of compound class $i$, $k_i$ (yr$^{-1}$), are assumed invariant along the sediment column and therefore independent of the nature of the terminal electron acceptor. The rate constants can be altered manually to fit observed sediment profiles (compare modelled profiles in Section 3.2) or related to a master variable provided by a coupled
Earth system model (e.g. sedimentation rate, see Section 4.2). The partitioning of the bulk OM pool into reactivity classes ($f_i$) needs to be specified manually in the stand-alone version or can be provided by the ESM. Organic matter degradation releases N, P and DIC to the pore water using Redfield molar ratios (Redfield, 1963) and consumes TEA with specific stoichiometries ($O_2C$, $NO_3C$, $SO_4C$) as summarised in Table 10. Table A1 in the appendix provides a list of reactions and their stoichiometries as implemented in OMEN-SED. The effect of OM degradation and secondary redox reactions on total alka-
linity is also accounted for via reaction specific stoichiometries representing the release of $NH_4$, $H_2S$ and P and is based on Jourabchi et al. (2005). In reality, the reoxidation of reduced substances produced during OM degradation may be incomplete. Yet, in OMEN-SED we have to assume their complete, instantaneous reoxidation at $z_{\mathrm{ox}}$ to allow for an analytical solution. In order to relax this assumption, for cases where it can be justified, we include a "switch" to allow part of the $NH_4$, $H_2S$ and





**Table 9.** Sediment characteristics and transport parameters.

| Parameter | Unit | Value | Description/Source |
|---|---|---|---|
| $\rho_{\mathrm{sed}}$ | $\mathrm{g\,cm^{-3}}$ | 2.6 | Sediment density |
| $w$ | $\mathrm{cm\,yr^{-1}}$ | Fct. of seafloor | Advection/Sediment accumulation rate |
| | | depth or from ESM | (Middelburg et al., 1997) |
| $z_{\mathrm{bio}}$ | cm | 10 or 0.01 | Bioturbation depth |
| | | | (Boudreau, 1998; Teal et al., 2010) |
| $D_{\mathrm{bio}}$ | $\mathrm{cm^2\,yr^{-1}}$ | Fct. of seafloor | Bioturbation coefficient |
| | | depth | (Middelburg et al., 1997) |
| $\phi$ | - | 0.85 | Porosity |
| F | - | $\frac{1}{\phi^m}$ | Tortuosity, here m=3 |
| $f_{ir}$ | - | 1 | Irrigation factor |
| **Diffusion coefficients** (Li and Gregory, 1974; Schulz, 2006; Gypens et al., 2008) | | | |
| $D_{\mathrm{O_2}}^0$ | $\mathrm{cm^2\,yr^{-1}}$ | 348.62 | Molecular diffusion coefficient of oxygen at $0^{\circ}\mathrm{C}$ |
| $D_{\mathrm{O_2}}^T$ | $\mathrm{cm^2\,yr^{-1}\,^{\circ}C^{-1}}$ | 14.09 | Diffusion coefficient for linear temp. dependence of oxygen |
| $D_{\mathrm{NO_3}}^0$ | $\mathrm{cm^2\,yr^{-1}}$ | 308.42 | Molecular diffusion coefficient of nitrate at $0^{\circ}\mathrm{C}$ |
| $D_{\mathrm{NO_3}}^T$ | $\mathrm{cm^2\,yr^{-1}\,^{\circ}C^{-1}}$ | 12.26 | Diffusion coefficient for linear temp. dependence of nitrate |
| $D_{\mathrm{NH_4}}^0$ | $\mathrm{cm^2\,yr^{-1}}$ | 309.05 | Molecular diffusion coefficient of ammonium at $0^{\circ}\mathrm{C}$ |
| $D_{\mathrm{NH_4}}^T$ | $\mathrm{cm^2\,yr^{-1}\,^{\circ}C^{-1}}$ | 12.26 | Diffusion coefficient for linear temp. dependence of ammonium |
| $D_{\mathrm{SO_4}}^0$ | $\mathrm{cm^2\,yr^{-1}}$ | 157.68 | Molecular diffusion coefficient of sulfate at $0^{\circ}\mathrm{C}$ |
| $D_{\mathrm{SO_4}}^T$ | $\mathrm{cm^2\,yr^{-1}\,^{\circ}C^{-1}}$ | 7.88 | Diffusion coefficient for linear temp. dependence of sulfate |
| $D_{\mathrm{H_2S}}^0$ | $\mathrm{cm^2\,yr^{-1}}$ | 307.48 | Molecular diffusion coefficient of sulfide at $0^{\circ}\mathrm{C}$ |
| $D_{\mathrm{H_2S}}^T$ | $\mathrm{cm^2\,yr^{-1}\,^{\circ}C^{-1}}$ | 9.64 | Diffusion coefficient for linear temp. dependence of sulfide |
| $D_{\mathrm{PO_4}}^0$ | $\mathrm{cm^2\,yr^{-1}}$ | 112.91 | Molecular diffusion coefficient of phosphate at $0^{\circ}\mathrm{C}$ |
| $D_{\mathrm{PO_4}}^T$ | $\mathrm{cm^2\,yr^{-1}\,^{\circ}C^{-1}}$ | 5.59 | Diffusion coefficient for linear temp. dependence of phosphate |
| $D_{\mathrm{DIC}}^0$ | $\mathrm{cm^2\,yr^{-1}}$ | 151.69 | Molecular diffusion coefficient of DIC at $0^{\circ}\mathrm{C}$ |
| $D_{\mathrm{DIC}}^T$ | $\mathrm{cm^2\,yr^{-1}\,^{\circ}C^{-1}}$ | 7.93 | Diffusion coefficient for linear temp. dependence of DIC |
| $D_{\mathrm{ALK}}^0$ | $\mathrm{cm^2\,yr^{-1}}$ | 151.69 | Molecular diffusion coefficient of ALK at $0^{\circ}\mathrm{C}$ |
| $D_{\mathrm{ALK}}^T$ | $\mathrm{cm^2\,yr^{-1}\,^{\circ}C^{-1}}$ | 7.93 | Diffusion coefficient for linear temp. dependence of ALK |

Note: DIC and ALK coefficients are the values of $\mathrm{HCO_3^-}$ from Schulz (2006).





$CH_4$ flux to escape reoxidation. The secondary redox parameters (i.e. $\gamma_{NH_4}$, $\gamma_{H_2S}$, $\gamma_{CH_4}$) therefore account for the fraction of reduced substances that are reoxidised and would be ideally parameterised for instance in relation to bottom water oxygen concentration or oxygen penetration depth ($z_{ox}$). Gypens et al. (2008) for example expressed $\gamma_{NH_4}$ as a function of oxygen penetration depth ($\gamma_{NH_4} = 0.243 \cdot \ln(z_{ox}) + 1.8479$) based on a fitting exercises to a numerical model and showed that the frac-

tion varies between 0.2 for $z_{ox} = 0.1$cm and 1.0 for $z_{ox} > 3$cm. Due to mathematical constraints in OMEN-SED for finding an analytical solution to the model equations these fractions take constant values generally representing oxygenated deep sea conditions. The instantaneous equilibrium adsorption coefficients of $NH_4$ and $PO_4$ ($K_{NH_4}$, $K_{PO_4}^{ox}$, $K_{PO_4}^{anox}$) are based on Wang and Van Cappellen (1996) and Slomp et al. (1998), respectively. The first order rate constants for sorption of $PO_4$ to Fe oxides ($k_s$), release of $PO_4$ from Fe-bound P due to Fe-oxide reduction ($k_m$) and authigenic CFA precipitation ($k_a$), as well as the

pore water equilibrium concentrations for P sorption and CFA precipitation ($PO_4{}^s$, $PO_4{}^a$) and the asymptotic concentration for Fe-bound P ($FeP^\infty$) are taken from Slomp et al. (1996). See Table 10 for a complete summary of the parameters and their values.

## 3 Stand-alone sensitivity analysis and case studies

### 3.1 Sensitivity Analysis

#### 3.1.1 Methodology

Model parameters implicitly account for processes that are not explicitly resolved. They are notoriously difficult to constrain and thus a primary source of uncertainty for numerical and analytical models - in particular on the global scale and/or in data-poor areas. A comprehensive sensitivity analysis can help quantify this uncertainty and identify the most sensitive parameters. More specifically, sensitivity analysis is used to investigate how the variations in the outputs ($y_1$, ..., $y_N$) of a model can be

attributed to variations in the different input parameters ($x_1$, ..., $x_M$, Pianosi et al., 2016). Different types of sensitivity indices, which quantify the relative influence of parameter $x_i$ on output $y_j$ with a scalar $S_{i,j}$ (for $i \in \{1, ..., M\}$ and $j \in \{1, ..., N\}$), can be calculated, ranging from simple one-at-a-time methods to statistical evaluations of the output distribution (e.g. variance-based or density-based approaches Pianosi et al., 2016). The latter indices take values between zero and one ($S_{i,j} \in [0,1]$), where zero indicates a non-influential parameter and a higher value a more influential parameter. Here, sensitivity analysis is

used mainly to identify which parameters have the largest impact on the different model outputs and therefore require more careful calibration. As the probability density functions of our model outputs (i.e. the resulting SWI-fluxes) are generally highly-skewed towards extreme organic matter degradation rates (not shown) variance-based sensitivity indices may not be a suitable proxy for output uncertainty (Pianosi et al., 2016). Hence, instead the density-based PAWN method by Pianosi and Wagener (2015) is employed which considers the entire conditional and unconditional Cumulative Distribution Function (CDF)

of the model output rather than its variance only. The unconditional CDF, $F_y(y)$, of output $y$ is obtained when all uncertain parameters ($x_1$, ..., $x_M$) are varied simultaneously, and the conditional CDFs, $F_{y|x_i}(y)$, are obtained when all inputs but the $i$-th parameter are varied (i.e. $x_i$ is fixed to a so-called conditioning value). The sensitivity index of parameter $i$ is measured by

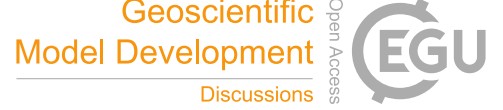

**Table 10.** Values for biogeochemical parameters used in OMEN-SED. The variables $x$, $y$ and $z$ denote the elemental ratio of carbon, nitrogen and phosphorus of the degrading organic matter (here set to $C : N : P = 106 : 16 : 1$).

| Parameter/Variable | Unit | Value | Description |
|---|---|---|---|
| **Stoichiometric factors and molecular ratios** | | | |
| $NC_i$ | mol/mol | $\frac{y}{x} = \frac{16}{106}$ | Nitrogen to carbon ratio |
| $PC_i$ | mol/mol | $\frac{z}{x} = \frac{1}{106}$ | Phosphorus to carbon ratio |
| $MC$ | mol/mol | 0.5 | Methane to carbon ratio |
| | | | produced during methanogenesis |
| $DICC^{I}$ | mol/mol | 1.0 | DIC to carbon ratio until $z_{SO_4}$ |
| $DICC^{II}$ | mol/mol | 0.5 | DIC to carbon ratio below $z_{SO_4}$ |
| $O_2C$ | mol/mol | $\frac{x+2y}{x} = \frac{138}{106}$ | Oxygen to carbon ratio |
| $NO_3C$ | mol/mol | $\frac{4x+3y}{5x} = \frac{94.4}{106}$ | Nitrate to carbon ratio |
| $SO_4C$ | mol/mol | $\frac{106}{212}$ | Sulfate to carbon ratio |
| $ALK^{OX}$ | mol/mol | $\frac{y-2z}{x} = \frac{14}{106}$ | ALK from aerobic degradation |
| $ALK^{NIT}$ | mol/mol | $-2$ | ALK from nitrification |
| $ALK^{DEN}$ | mol/mol | $\frac{4x+3y-10z}{5x} = \frac{92.4}{106}$ | ALK from denitrification |
| $ALK^{SUL}$ | mol/mol | $\frac{x+y-2z}{x} = \frac{120}{106}$ | ALK from sulfate reduction |
| $ALK^{MET}$ | mol/mol | $\frac{y-2z}{x} = \frac{14}{106}$ | ALK from methanogenesis |
| $ALK^{H_2S}$ | mol/mol | $-2$ | ALK from $H_2S$ oxidation |
| $ALK^{AOM}$ | mol/mol | 2 | ALK from AOM |
| **Secondary reaction parameters** | | | |
| $\gamma_{NH_4}$ | - | 0.9 | Fraction of $NH_4$ that is nitrified |
| $\gamma_{H_2S}$ | - | 0.95 | Fraction of $H_2S$ that is oxidised |
| $\gamma_{CH_4}$ | - | 0.99 | Fraction of $CH_4$ that is oxidised |
| **Adsorption coefficients** (Wang and Van Cappellen, 1996; Slomp et al., 1998) | | | |
| $K_{NH_4}$ | - | 1.4 | $NH_4$ adsorption coefficient |
| $K_{PO_4}^{ox}$, $\quad K_{PO_4}^{anox}$ | - | 200.0, 2.0 | $PO_4$ adsorption coefficient (oxic, anoxic) |
| **P related parameters** (Slomp et al., 1996) | | | |
| $k_s$ | $yr^{-1}$ | 94.9 | Rate constant for $PO_4$ sorption |
| $k_m$ | $yr^{-1}$ | 0.193 | Rate constant for Fe-bound P release |
| $k_a$ | $yr^{-1}$ | 0.365 | Rate constant for authigenic CFA precipitation |
| $PO_4{}^s$ | $mol\,cm^{-3}$ | $1 \cdot 10^{-9}$ | Equilibrium conc. for P sorption |
| $FeP^{\infty}$ | $mol\,cm^{-3}$ | $1.99 \cdot 10^{-10}$ | Asymptotic concentration for Fe-bound P |
| $PO_4{}^a$ | $mol\,cm^{-3}$ | $3.7 \cdot 10^{-9}$ | Equilibrium conc. for authigenic P precipitation |





**Table 11.** Range of model parameters used for sensitivity analysis of model predicted output.

| Parameter | Description | Units | Minimum | Maximum | Source |
|---|---|---|---|---|---|
| $k_1$ | labile OM degradation constant | $yr^{-1}$ | $1e^{-4}$ | 5.0 | (1) |
| $\widetilde{k_2}$ | order of refractory OM degradation constant ($k_2 = \widetilde{k_2} \cdot k_1$) | - | $1e^{-4}$ | $1e^{-1}$ | (1) |
| $f_1$ | fraction of labile OM | - | 0.02 | 0.98 | - |
| $K_{NH_4}$ | Adsorption coefficient | - | 0.8 | 1.7 | (2) |
| $\gamma_{NH_4}$ | $NH_4$ fraction oxidised | | 0.5 | 1.0 | - |
| $\gamma_{H_2S}$ | $H_2S$ fraction oxidised | | 0.5 | 1.0 | - |
| $K_{PO_4}^{ox}$ | Adsorption coeff. oxic | - | 100.0 | 400.0 | (3) |
| $K_{PO_4}^{anox}$ | Adsorption coeff. anoxic | - | 1.3 | 2.0 | (3) |
| $k_s$ | kinetic P sorption | $yr^{-1}$ | 0.1 | 100.0 | (4, 5) |
| $k_m$ | Fe-bound P release | $yr^{-1}$ | 0.015 | 0.02 | (4, 5) |
| $k_a$ | authigenic P formation | $yr^{-1}$ | 0.001 | 10.0 | (4, 6) |

Sources:    (1) Arndt et al. (2013); (2): Van Cappellen and Wang (1996); (3): Krom and Berner (1980)

(4): Gypens et al. (2008); (5): Slomp et al. (1996); (6): Van Cappellen and Berner (1988)

the distance between the two CDFs using the Kolmogorov-Smirnov statistic (Kolmogorov, 1933; Smirnov, 1939), i.e.:

$$S_i = \max_{x_i} \max_{y} |F_y(y) - F_{y|x_i}(y)|. \tag{49}$$

Since $F_{y|x_i}(y)$ accounts for what happens when the variability due to $x_i$ is removed, the distance between the two CDFs provides a measure of the effects of $x_i$ on the output $y$. Due to the model complexity it is impossible to compute the sensitivity
indices analytically. Therefore, they are approximated from a Latin-Hypercube sampling of parameter inputs and calculated outputs. For a brief description of the methodology, see Fig. 4. For more details, we refer the interested reader to Pianosi and Wagener (2015).

The PAWN method, as implemented within the Sensitivity Analysis for Everyone (SAFE) matlab toolbox (Pianosi et al., 2015), is used to investigate M = 11 model parameters for ranges as specified in Table 11. Sensitivity indices for all resulting
SWI-fluxes for two idealised sediment conditions (i.e. anoxic at 400 m and oxic at 4000 m, see Table 12) are calculated. We use NU = 200 samples to estimate the unconditional CDF, NC = 100 samples to estimate the conditional CDFs and n = 10 conditioning points. Thus as $N_{eval} = 200 + 100 \cdot 10 \cdot 11$, 11200 model evaluations are performed for each sediment condition. The resulting indices are then translated into a colour code and summarised in a pattern plot to simplify comparison (Fig. 5).

### 3.1.2    Results

Fig. 5 summarises results of the sensitivity analysis as a colour map. Results indicate that generally the most significant parameters for all model outputs are the degradation rate constant for the labile OM pool ($k_1$) and the fraction of this pool to





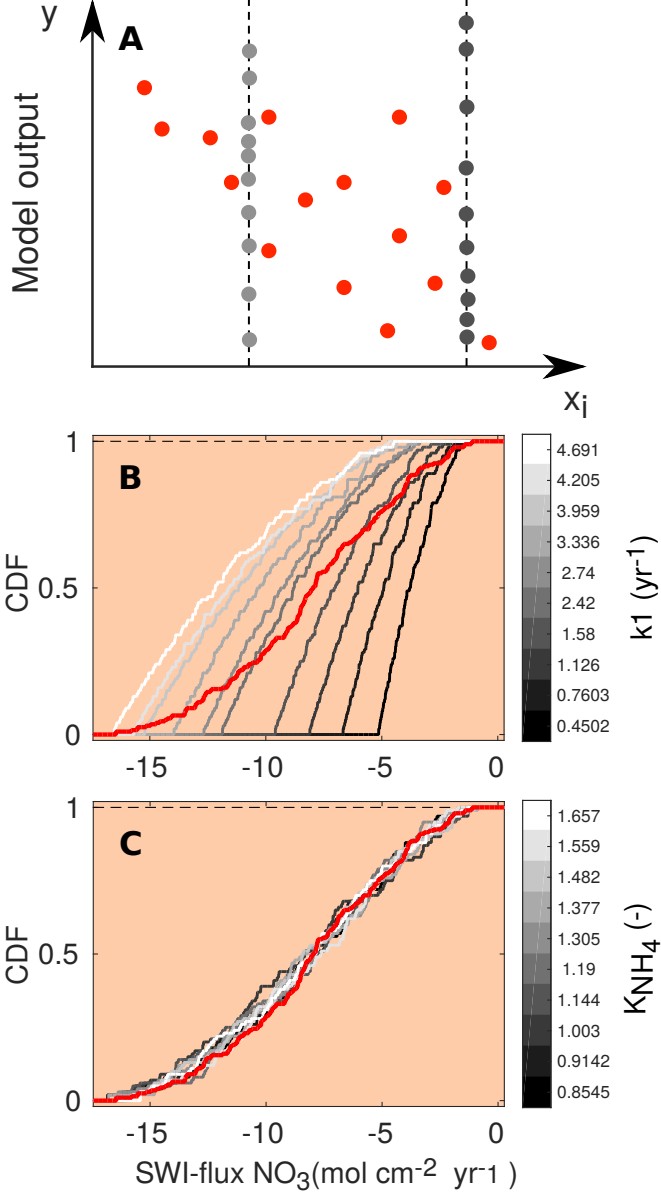

**Figure 4.** A: Schematic of the PAWN method, plotting an uncertain parameter ($x_i$) against a generic model output ($y$). Red dots represent points for calculating the unconditional CDF (NU, here 15), grey dots are points for calculating each conditional CDF (NC, here 10) with n = 2 conditioning points as an example. The user can change the values of NU, NC and n. The number of model evaluations equals $N_{eval}$ =NU + n·NC·M, where M is the number of uncertain input parameters. B + C: Two examples of CDFs of the model calculated SWI-flux of $NO_3$ using NU = 200, NC = 100 and n = 10. The red lines are the unconditional distribution functions $F_y(NO_3)$ and the grey lines are the conditional distribution functions $F_{y|x_i}(NO_3)$ at different fixed values of the input parameters $k_1$ (B) and $K_{NH_4}$ (C). As the maximal distance between conditional CDFs and unconditional CDF is greater for $k_1$, this parameter is more influential for the model output (here SWI-flux of $NO_3$, compare Fig. 5).



**Table 12.** Model boundary conditions for the two idealised sediment conditions used for the sensitivity analysis (Fig. 5 and 6). All solute concentrations are in nmol cm$^{-3}$.

| Depth (m) | Temp. (°C) | OC (wt%) | $O_2$ | $NO_3$ | $SO_4$ | $PO_4$ | $z_{bio}$ (cm) |
|---|---|---|---|---|---|---|---|
| 400 | 8.0 | 2.0 | 0.0 | 40.0 | 28,000 | 40.0 | 0.001 |
| 4000 | 1.5 | 1.0 | 300.0 | 20.0 | 28,000 | 40.0 | 10.0 |

the total OM stock ($f_1$). Other parameters play a minor role for the SWI-fluxes, with the exception of the secondary redox parameters (i.e. $\gamma_{NH_4}$, $\gamma_{H_2S}$) in the oxic scenario. Here, $NH_4$, $SO_4$ and $H_2S$ are very sensitive to changes in $\gamma_{NH_4}$ and $\gamma_{H_2S}$, as these parameters determine how much of the respective TEA is produced in situ via reoxidation, thus affecting the resulting SWI-fluxes. For the oxic scenario, the reoxidation of $H_2S$ produced in the sulfidic layer also has a strong influence on alkalinity

($\gamma_{H_2S}$, Table 8 Eq. 5) as it decreases alkalinity by 2 moles per mole of S oxidized (ALK$^{H_2S}$, Table 10). However, these high sensitivities are partially caused by the wide range of allowed values ($\gamma_{NH_4}$, $\gamma_{NH_4} \in [0.5; 1.0]$). Yet, for oxic deep sea conditions it is more likely that reduced substances are almost completely reoxidised (e.g. Hensen et al., 2006). For the anoxic scenario the secondary redox parameters are essentially non-influential as no $O_2$ is available for the reoxidation of reduced substances. Especially for the oxic condition the $PO_4$ SWI-flux appears to be insensitive to P-related parameters (i.e. $K_{PO_4}^{ox}$, $K_{PO_4}^{anox}$, $k_s$, $k_m$,

$k_a$) as the majority is absorbed to Fe-oxides. The sensitivities change if other $PO_4$ related equilibrium concentrations $PO_4{}^s$, $PO_4{}^a$ and FeP$^\infty$ are used (not shown). Overall the results of the sensitivity analysis are in line with what one would expect from a diagenetic model and thus provide ground to confirm that OMEN-SED provides sensible results. The parameterisation of the organic matter pools ($f_1$) and their degradation rate constants ($k_1$, $k_2$) is critical especially when the model is used in a global Earth system model framework, as these parameters, as well as the $\gamma$-parameters, can have a very important influence on

the flux of dissolved species through the SWI. At the same time these are the weakest constrained parameters. Thus, one should rather choose $\gamma$-values close to 1 and consider carefully where a relaxation of the "all reoxidised" assumption is appropriate. In contrast, the importance of the OM degradation rate constants can not be overemphasised. Therefore, much care should be given to how these are parameterised in coupled simulations and a range of different plausible scenarios should be tested to quantify uncertainty.

Because of the strong sensitivity of model results on OM degradation rate parameters, we further explore the sensitivity of simulated sediment-water exchange fluxes to variations in organic matter degradation parameters by varying $k_1$, $f_1$ and $\widetilde{k_2}$ while all other model parameters are set to their default values (Tables 9 and 10). Minimum and maximum values for $k_1$, $\widetilde{k_2}$ and $f_1$ in the shallow ocean are as in Table 11. For the deep sea condition we account for the presence of more refractory OM by sampling $f_1 \in [0.02, 0.3]$, whereas the variation of $k_1$ and $\widetilde{k_2}$ is as in the shallow ocean. The parameter space is sampled using

another Latin-Hypercube approach with sample sizes of $N = 3500$ for each idealised sediment condition. Figure 6 summarises the results of the sensitivity study and the ranges of observed $O_2$ and $NO_3$ sediment-water interface fluxes extracted from a global database (Stolpovsky et al., 2015) are indicated on the colour scale. The colour patterns in Figure 6 A and B reveal the complex interplay between the amount of labile OM $f_1$ and its degradation rate $k_1$ for the resulting SWI-fluxes of $NO_3$





**Figure 5.** Pattern plot, showing the output sensitivity for each SWI flux (i.e. the chemical compounds on the vertical axis) and each input factor (i.e. the model parameters on the horizontal axis) for two idealised sediment cores. White patterns are assigned where the SWI flux is independent of the specific parameter.





**Figure 6.** Scatter plots ($k_1$ vs $f_1$) of resulting OMEN-SED SWI-fluxes for the 400m anoxic (A: $NO_3$) and 4000m oxic (B: $O_2$, C: $NO_3$) scenario. Negative values represent a flux from the water column into the sediments. Ranges indicated in red on the colour scale correspond to observed benthic fluxes as reported in the global database of Stolpovsky et al. (2015).

in anoxic sediments and $O_2$ in aerobic sediments. In general, a higher degradation rate in combination with more labile OM available leads to a higher SWI-flux. However, higher fluxes extend over a larger range of $k_1$-values when the amount of labile OM $f_1$ is high. The absence of a colour pattern in Figure 6 C highlights the limited interaction of the two model parameters for $NO_3$ SWI-fluxes under oxic conditions. Figure 6 shows that SWI-fluxes can vary widely over the range of plausible organic

5    matter degradation parameters and that simulated fluxes generally fall within the range of observed SWI-fluxes. However, a large number of different k-f combinations can result in SWI-fluxes that fall within the observed ranges reported by Stolpovsky et al. (2015) further emphasising the care that should be devoted to constraining OM degradation parameters.





### 3.2 Case study: Simulations of sediment cores

#### 3.2.1 Methodology

In order to illustrate the capabilities of OMEN-SED, comprehensive datasets from the Santa Barbara Basin (Reimers et al., 1996), as well as from the Iberian margin and the Nazaré Canyon (Epping et al., 2002) are modelled. Modelled profiles are

compared with measured pore water data from different depths including the continental shelf (108 m) and the lower slope (2213 m) located at the Iberian margin, the upper slope (585 m) from the Santa Barbara Basin, and a deep sea site (4298 m) in the Nazaré Canyon. The Santa Barbara Basin is characterised by anoxic bottom waters, high POC concentrations and varved sediments (Reimers et al., 1990), therefore the depth of bioturbation in OMEN-SED is restricted to the upper 0.01 cm. In the uppermost sediments iron(III) hydroxides are reduced, releasing $Fe^{2+}$ which reacts with sulfide to form iron sulfides. Thus,

the Fe cycle exerts a strong control on sulfide concentrations in the sediments of this basin (Reimers et al., 1996). In addition, the sediments are generally supersaturated with respect to carbonate fluorapatite by and below 2 cm (Reimers et al., 1996). The Iberian margin, situated in the northeastern Atlantic, generally belongs to the more productive regions of the global ocean (Longhurst et al., 1995), however, seasonal changes in upwelling creates a strong temporal variability in primary productivity and organic carbon deposition. Submarine canyons in this area (like the Nazaré Canyon) may deliver organic carbon from the

shelf to the ocean interior (van Weering et al., 2002; Epping et al., 2002). For a more detailed description of the study areas and the experimental work, the interested reader is referred to the publications by Reimers et al. (1996) and Epping et al. (2002).

In OMEN-SED sediment characteristics and boundary conditions are set to the observed values where available (Table 13). Other sediment characteristics (e.g. sedimentation rate, porosity, density), stoichiometric factors and secondary reaction parameters are set to the default value (see Tables 9 and 10). Organic matter is modelled as two fractions, with different first-

order degradation rate constants. The POC and pore water profiles were manually fitted by optimizing the POC partitioning into the fast and slow degrading pool and their respective first-order degradation rate constants (priority is given to reproduce the POC and $O_2$ profiles). For phosphorus the equilibrium concentration for authigenic P formation ($PO_4^a$) was adjusted to fit the $PO_4$ concentration at $z_\infty$.

#### 3.2.2 Results

Figure 7 compares modelled and observed sediment profiles for the Santa Barbara Basin and the Iberian margin. Results show that OMEN-SED is able to capture the main diagenetic features across a range of different environments without changing model parameters (other than the 4 we tuned, i.e. $k_1$, $k_2$, $f_1$ and $PO_4^a$) to site specific conditions. For the two open Iberian margin stations (108 and 2213 m) OMEN-SED fits all observations well. OMEN-SED does especially well at seafloor depth (SFD) 2213 m by reproducing the deep $O_2$ penetration and the subsurface maximum in $NO_3$ concentration due to the nitrification

of $NH_4$ (note, that $NH_4$ is overestimated at this SFD). For the anoxic Santa Barbara Basin (585 m) the decrease in $SO_4$ and the increase in ALK concentration with sediment depth is well represented, indicating the importance of sulfate reduction as the primary pathway of OM degradation at this site (compare with Meysman et al., 2003). However, a misfit is observed for $H_2S$ and $PO_4$ in the upper 20 cm of this sediment core. The discrepancy for $H_2S$ can be explained by high iron(III)





**Table 13.** Model boundary conditions for the simulated sediment profiles in the Santa Barbara basin (108 and 2213 m) and Iberian margin (585 and 4298 m) reported in Figure 7. For all sites a DIC bottom water concentration of 2,400 nmols cm$^{-3}$ is assumed.

**Sediment characteristics:**

| Depth (m) | Temp. ($^\circ$C) | $z_{bio}$ (cm) | $D_{bio}$ (cm$^2$yr$^{-1}$) | POC$_1$ (wt%) | POC$_2$ (wt%) | $k_1$ (yr$^{-1}$) | $k_2$ (yr$^{-1}$) | PO$_4^a$ (nmol cm$^{-3}$) |
|---|---|---|---|---|---|---|---|---|
| 108 | 12.50 | 1.00 | 0.02 | 2.64 | 1.8 | 0.650 | $1.0e^{-5}$ | 15.0 |
| 585 | 5.85 | 0.01 | 0.02 | 2.00 | 3.5 | 0.200 | $8.0e^{-4}$ | 90.0 |
| 2213 | 3.20 | 10.00 | 0.17 | 0.45 | 0.5 | 0.100 | $4.0e^{-4}$ | 5.0 |
| 4298 | 2.50 | 4.20 | 0.18 | 0.83 | 1.2 | 0.052 | $1.0e^{-5}$ | 5.0 |

**Bottom water concentrations of solutes** (all in nmol cm$^{-3}$):

| Depth | O$_2$ | NO$_3$ | SO$_4$ | NH$_4$ | H$_2$S | PO$_4$ | Alkalinity |
|---|---|---|---|---|---|---|---|
| 108 | 210.0 | 9.6 | 28,000 | 0.40 | 0.0 | 0.0 | 2,400 |
| 585 | 10.0 | 25.0 | 28,000 | 0.00 | 0.0 | 50.0 | 2,480 |
| 2213 | 250.0 | 25.0 | 28,000 | 0.60 | 0.0 | 0.0 | 2,400 |
| 4298 | 243.0 | 30.1 | 28,000 | 0.22 | 0.0 | 0.0 | 2,400 |

hydroxide concentrations, which is reduced to degrade organic matter (especially in the $2-4$ cm depth interval), therefore placing the beginning of the sulfate reduction zone and the production of H$_2$S to the deeper sediments (Reimers et al., 1996). Iron processes are currently not dynamically represented in OMEN-SED. In addition, produced dissolved Fe reacts with H$_2$S to form iron sulfides (e.g. pyrite, FeS$_2$) and thus further inhibits the rise of H$_2$S (Reimers et al., 1990). The iron cycle also

plays a critical role for phosphorus, as the reduction of iron(III) hydroxides in the surface sediments releases sorbed phosphate, leading to pore waters around and below 2 cm which are supersaturated with respect to fluorapatite, thus initiating CFA precipitation. Reimers et al. (1996) could even show that the accumulation of CFA is mainly restricted to the near-surface sediments ($\sim 5$ cm) instead of throughout the sediment column. As OMEN-SED currently does not include an iron-cycle, and Fe-bound P and CFA processes are highly parameterised, the model is not able to capture these complex, non-steady state

phosphorus dynamics at this specific site. For the Nazaré Canyon station (4298 m) satisfactory fits could be realised apart from NH$_4$. However, also Epping et al. (2002) could not obtain a better fit using the diagenetic model OMEXDIA. They suggested non-local solute exchange resulting from bioirrigation being responsible for the higher NH$_4$ concentrations at this site which is neglected in their model, as well as in OMEN-SED. Furthermore, the fractured POC profile (indicating episodic depositional events through the canyon) could have been approximated using a different partitioning of the bulk POC into

labile and refractory pool with different degradation rate constants, thus potentially leading to a better fit of the NH$_4$ profile. In general, better approximations of the data could have potentially been acquired by applying a sensitivity study using different NC-ratios (e.g. Epping et al., 2002, report different ratios from Redfield stoichiometry) and exploring the parameter space for the secondary reaction parameters ($\gamma_{NH_4}$, $\gamma_{H_2S}$). However, considering these generalisations and our assumption of steady-state, which might not be valid, particularly for the complex Santa Barbara basin, the shallow core and the Nazaré Canyon,





which are affected by seasonality and biology, OMEN-SED generally reproduces the observed pore water trends and hence captures the main diagenetic processes.

### 3.3 Case study: Stand-alone simulations of global ocean transect

#### 3.3.1 Methodology

In this section we explore to what degree OMEN-SED is capable of capturing the dynamics of organic matter degradation pathways and related TEA-fluxes as simulated with a more complete and complex numerical diagenetic model (Thullner et al., 2009). Therefore, we reproduce the simulations of typical conditions along a global ocean hypsometry of Thullner et al. (2009) and compare our modelled TEA-fluxes with the results of the complex model as well as with observations from Middelburg et al. (1996). To explore the global degradation of OM in the seafloor Thullner et al. (2009) quantified various diagenetic

processes using the Biogeochemical Reaction Network Simulator (BRNS, Aguilera et al., 2005), a flexible simulation environment suitable for reactive transport simulations of complex biogeochemical problems (e.g. Jourabchi et al., 2005; Thullner et al., 2005). Thullner et al. (2009) used seafloor depth (SFD) as the master variable and calculated model parameters, such as $w$, $D_{bio}$ and $\phi$, from existing empirical relationships (e.g. Van Cappellen and Wang, 1995; Middelburg et al., 1997). Organic matter degradation was described with a 1G approach, thus assuming a single pool of organic matter of uniform reactivity. The

first order rate constant was related to the burial velocity, $w$ (cm year$^{-1}$), following the empirical relationship of Boudreau (1997):

$$k = 0.38 \cdot w^{0.59}. \tag{50}$$

This rate constant can be assumed as the mean reactivity of the organic matter fractions which are degraded in the upper, bioturbated $10-20$ cm of the sediments. Thus, more reactive fractions (degraded during days/weeks close to the SWI) and more

refractory fractions (degraded on longer time scales deeper in the sediments) are not captured by this relationship (Boudreau, 1997). BRNS simulations were performed using boundary conditions and parameters for depths representative for shelf, slope and deep sea sediments (i.e. SFD of 100m, 200m, 500m, 1000m, 2000m, 3500m and 5000m). In order to reproduce these results, OMEN-SED is configured here as a 1G model and boundary conditions and model parameters are defined as in Thullner et al. (2009, see Table 14). As OMEN-SED assumes a fixed fraction (i.e. $\gamma_{NH_4}$, $\gamma_{H_2S}$) of reduced substances to

be reoxidised, which exerts a large impact on the resulting SWI-fluxes (compare Section 3.1), two sets of simulations are performed in order to show the range of possible model outputs. In the first setup 95% of the reduced substances are reoxidised (i.e. $\gamma_{NH_4} = \gamma_{H_2S} = 0.95$) and in the second, less realistic case, only 5% are reoxidised (all other model parameters and boundary conditions are equal).

#### 3.3.2 Results

Figure 8 compares simulated SWI-fluxes of TEAs (i.e. O$_2$, NO$_3$ and SO$_4$) along the global hypsometry using OMEN-SED (black lines) with the results of Thullner et al. (2009) (red lines). Observations for O$_2$ and NO$_3$ fluxes are taken from Middel-





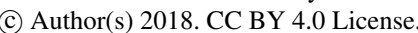

**Figure 7.** Modelled (curves) and measured (filled dots) solid phase and dissolved pore water profiles for four different sediment cores. Note that different scales are used for different stations. The blue POC curve represents the sum of the refractory (green) and labile (red) POC fraction. The horizontal dashed lines in each panel indicate the bioturbation depth (black) and the penetration depths of oxygen (blue), nitrate (green) and sulfate (red) as calculated by OMEN-SED.



**Table 14.** Seafloor depth dependency of key model parameters and boundary conditions (adapted from Thullner et al., 2009).

| | Seafloor depth | | | | | | |
| --- | --- | --- | --- | --- | --- | --- | --- |
| | 100 m | 200 m | 500 m | 1000 m | 2000 m | 3500 m | 5000 m |
| **Model parameters** | | | | | | | |
| $w^a$ (cm yr$^{-1}$) | $3.98 \times 10^{-1}$ | $3.60 \times 10^{-1}$ | $2.67 \times 10^{-1}$ | $1.62 \times 10^{-1}$ | $5.94 \times 10^{-2}$ | $1.32 \times 10^{-2}$ | $2.94 \times 10^{-3}$ |
| $D_{bio}^a$ (cm$^2$ yr$^{-1}$) | 27.5 | 25.1 | 19.0 | 12.1 | 4.83 | 1.23 | 0.310 |
| $\phi^b$ | 0.85 | 0.85 | 0.80 | 0.80 | 0.80 | 0.80 | 0.80 |
| T$^c$ (°C) | 10.3 | 9.7 | 8.1 | 5.8 | 3.0 | 1.5 | 1.4 |
| $\rho_{sed}^c$ (g cm$^{-3}$) | 2.5 | 2.5 | 2.5 | 2.5 | 2.5 | 2.5 | 2.5 |
| $k^d$ (yr$^{-1}$) | 0.221 | 0.208 | 0.174 | 0.130 | 0.0718 | 0.0296 | 0.0122 |
| **Upper boundary conditions** | | | | | | | |
| POC$_{flux}^a$ ($\mu$mol cm$^{-2}$yr$^{-1}$) | 510 | 467 | 357 | 228 | 93.0 | 24.3 | 6.33 |
| POC$^e$ (wt%) | 0.79 | 0.78 | 0.55 | 0.50 | 0.42 | 0.32 | 0.25 |
| O$_{2,0}^c$ (nmol cm$^{-3}$) | 132 | 129 | 121 | 114 | 116 | 135 | 141 |
| NO$_{3,0}^c$ (nmol cm$^{-3}$) | 17.3 | 18.6 | 22.1 | 26.5 | 31.0 | 31.6 | 31.6 |
| SO$_{4,0}^b$ (nmol cm$^{-3}$) | 28,000 | 28,000 | 28,000 | 28,000 | 28,000 | 28,000 | 28,000 |

[a] Derived from Middelburg et al. (1997).
[b] Derived from Van Cappellen and Wang (1995).
[c] Derived from Conkright et al. (2002).
[d] Derived from Boudreau (1997).
[e] Calculated with OMEN-SED from POC$_{flux}$.





burg et al. (1996). Due to the applied empirical relations organic matter flux to the seafloor decreases by 2 orders of magnitude from 100 to 5000 m and its degradation rate constant by 1 order of magnitude (Table 14). Therefore, the rate of organic matter degradation is about 50 times greater at 100 m than at 5000 m (compare Thullner et al., 2009), thus resulting in a decrease of TEA-fluxes along the hypsometry (Figure 8). The 95%-reoxidation experiments (dots) show proportionally higher $O_2$ in-
fluxes than the 5%-reoxidation experiments (triangles) because more $O_2$ is utilised for in situ production of $NO_3$ and $SO_4$ in the sediments. This is also mirrored by the increased $NO_3$ out-flux and decreased $SO_4$ in-flux for shallower SFDs. This is in line with the results of Thullner et al. (2009) which showed that in situ production is an important pathway of $SO_4$ supply in the sediment, which is responsible for ∼80% of the total OM degradation at depths between 100 and 2000 m (in our results $SO_4$ is not used for OM degradation in OMEN-SED below 2000m). In general, Figure 8 shows that OMEN-SED captures the
main trends in observed and numerically simulated TEA fluxes well. Results also confirm that higher $\gamma$-values better represent SWI-fluxes for most of the global hypsometry. A slight overestimation of shallow ocean SWI-fluxes (SFD < 200 m) for the high $\gamma$ scenario indicates that slightly lower $\gamma$-values would better capture SWI-fluxes in these areas, where rapid oxygen consumption favours the escape of reduced species across the SWI.

In addition, observed $O_2$ fluxes in the upper 2000m are generally encompassed by our total range in predicted OMEN-SED
fluxes. Oxygen fluxes for the deep-sea sediments, however, are slightly underestimated. These deviations can presumably be related to the assumed 1G description of organic matter degradation, which neglects the more labile OM pool. This highly reactive pool is degraded close to the sediment surface, thus promoting higher aerobic degradation rates and higher $O_2$ fluxes. Nitrate fluxes in the upper 500m of the Atlantic Ocean are well predicted. However, as in Middelburg et al. (1996) the direction of calculated nitrate fluxes in the upper 1000m of the Pacific Ocean differ from the observations. Middelburg et al. (1996)
related these discrepancies to the globally averaged model parameters and the applied boundary conditions. They could reduce the disagreements significantly by using more representative bottom water concentrations for the eastern Pacific and a higher flux of labile organic matter for their 2G model. By changing the boundary conditions and the N:C elemental ratio of organic matter for the whole hypsometry, it is possible to obtain a better model-data fit with OMEN-SED for the shallow Pacific Ocean (green line in Fig. 8B). Bohlen et al. (2012) report that the elemental N:C ratio strongly deviates from Redfield stoichiometry
(0.151) with specifically lower values for the East Pacific Ocean. The use of their globally averaged value of 0.067 allows reconciling modelled and observed values provided that bottom water conditions are also changed to the low oxygen/high nitrate levels more likely to be found in the shallow Pacific Ocean ($O_2 = 10$ nmol cm$^{-3}$ and $NO_3 = 80$ nmol cm$^{-3}$).

## 4 Coupled pre-industrial Earth system model simulations

### 4.1 Coupling to the cGENIE Earth system model

In a final step, we couple OMEN-SED to the carbon-centric version of the "GENIE" Earth system model (cGENIE, Ridgwell et al., 2007) in order to illustrate how a fully coupled ocean-sediment system can be configured and applied. We start by providing a brief description of cGENIE and the coupling procedure (Fig. 9).



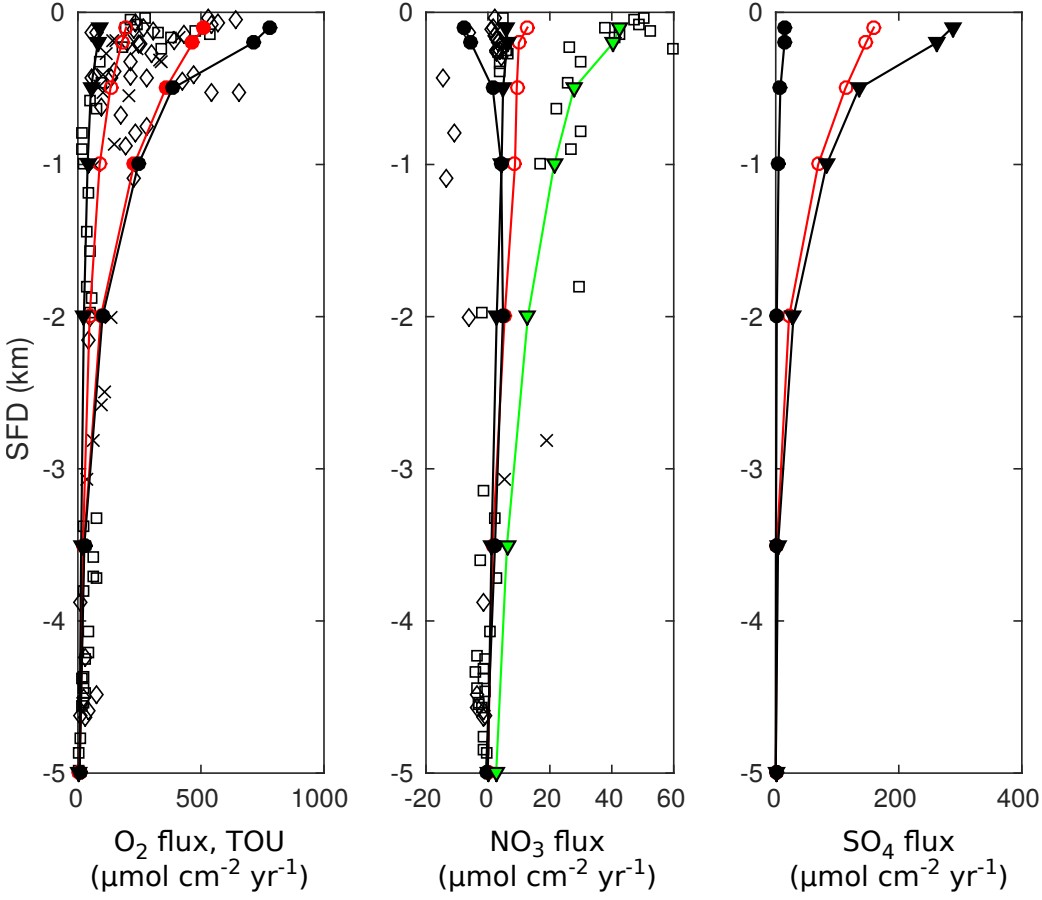

**Figure 8.** Fluxes of $O_2$, $NO_3$ and $SO_4$ to the sediment along the global hypsometry. Red lines (with open symbols) are modelled fluxes from Thullner et al. (2009) using BRNS; black lines are results from OMEN-SED (• : $\gamma_{NH_4} = \gamma_{H_2S} = 0.95$; ▼: $\gamma_{NH_4} = \gamma_{H_2S} = 0.05$). Observations of TEA fluxes are taken from Middelburg et al. (1996) (◇: Atlantic, □: Pacific, ×: Arctic/Indian Ocean). Also plotted in Figure (A) are the total oxygen uptake (TOU) estimates of Thullner et al. (2009) (filled red symbols). The green line indicates OMEN-SED results for low oxygen/high nitrate levels and the lower NC-ratio. Positive fluxes are directed from the ocean into the sediments.





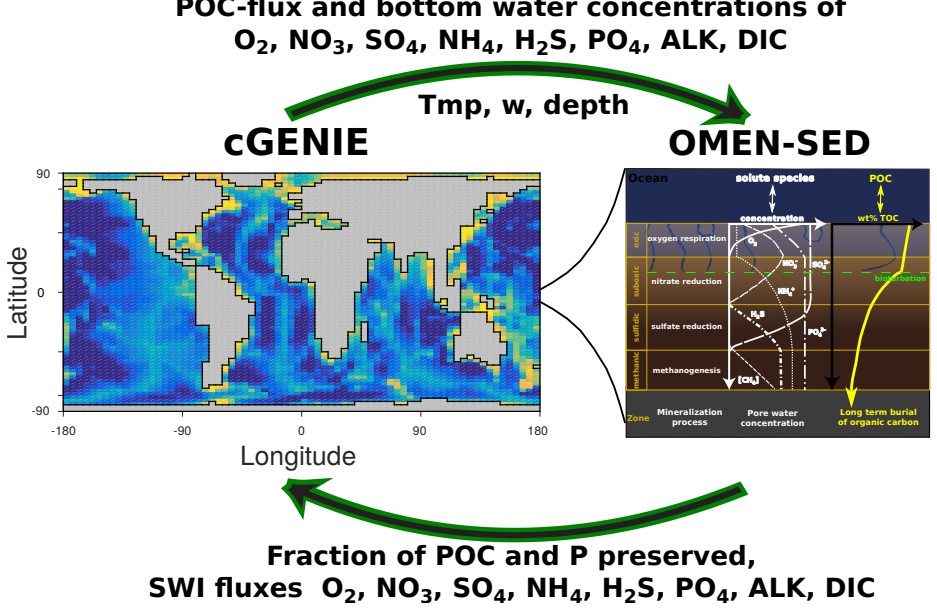

**Figure 9.** Schematic of the relationship between OMEN-SED and cGENIE. Arrows represent the information transferred between models.

cGENIE is a model of Intermediate Complexity based on the efficient climate model "C-GOLDSTEIN" of Edwards and Marsh (2005), featuring a frictional-geostrophic 3D-ocean circulation model coupled to a fast Energy-Moisture Balance 2D-atmosphere together with a dynamic-thermodynamic sea-ice component. The version of cGENIE used here includes the marine geochemical cycling of carbon, oxygen, phosphorus and sulfur (Ridgwell et al., 2007), preservation of carbonates in deep-sea sediments (SEDGEM, Ridgwell and Hargreaves, 2007) and terrestrial weathering (Colbourn et al., 2013). The ocean model is implemented on a $36 \times 36$ equal-area horizontal grid with 16 vertical levels using the pre-industrial continental configuration and bathymetry as in Archer et al. (2009). A finer grid ($72 \times 72$) is used for the sediments (see Fig. 11C and Ridgwell and Hargreaves, 2007) and OMEN-SED is called by SEDGEM for each wet ocean grid point.

In our Earth system model set-up, we prescribe the burial sediment fluxes of detrital material, opal and $CaCO_3$, while leaving OMEN-SED to calculate organic matter preservation. This assumption serves two purposes. First, the run-time of the model is minimized as steady-state conditions are reached earlier, compared to the ca. 20-50 kyr adjustment time for surface sediment $CaCO_3$ (Ridgwell and Hargreaves, 2007). Second, invariant flux fields remove feedbacks between OMEN-SED and the calculation of $CaCO_3$ preservation (changes in organic matter preservation affect $CaCO_3$ dissolution and hence burial rates which in turn affects weight percent of organic matter in the sediments) that would not only lengthen the sediment adjustment time but also make it impossible to carry out unbiased comparisons between different assumptions regarding organic matter reactivity in OMEN-SED.





We derive these fields form the data compilation of Archer (1996) as follows. First, we re-grid the Archer (1996) interpolated non carbonate mass accumulation rate field ($NC_{flux}$) to the $72 \times 72$ cGENIE sediment grid. This field includes detrital material plus opal (plus a minor contribution from organic matter). We could then directly calculate $\sum_{flux}$ (total burial flux of all components or total sediment accumulation rate) from this plus measurements of coretop wt% $CaCO_3$ ($C_{wtpct}$) (Archer,

1996) as $\sum_{flux} = NC_{flux} \cdot (1 - \frac{C_{wtpct}}{100})^{-1}$. However, some of the Archer (1996) database $C_{wtpct}$ values are both close to 100% and associated with high $NC_{flux}$, and hence would lead to unrealistically high values for $\sum_{flux}$. We therefore impose a plausibility filter, by also re-gridding coretop wt% opal ($O_{wtpct}$) and quartz ($Q_{wtpct}$) and for grid points in which more than one component is reported and the sum exceeds 100 wt%, normalizing the individual components. (Note that for grid points with only a single solid component, no change is made.) We then calculate the individual solid component burial fluxes, and

sum them up. To interpolate between the grid points associated with data, we iteratively average nearest (adjoining) neighbours. The distribution of the total burial flux $\sum_{flux}$ (in $g\,cm^{-2}\,kyr^{-1}$) is shown in Figure C1 in the Appendix.

Depending on the configuration of the overlying biogeochemical ocean model, processes can be included or excluded in OMEN-SED and stoichiometric factors (Tab. 10) need to be matched between models to ensure preservation of mass. As nitrogen is not modelled explicitly in the employed cGENIE configuration, $NC_i$, $ALK^{NIT}$ and $ALK^{DEN}$ in OMEN-SED are set

to zero. cGENIE, however, implicitly includes the effects of $NH_4$ release and its complete nitrification on alkalinity but neglects the impact of P release. Therefore, alkalinity stoichiometries for aerobic degradation, sulfate reduction and methanogenesis are changed to $ALK^{OX} = -16/106$, $ALK^{SUL} = 122/106$ and $ALK^{MET} = -16/106$, respectively (compare to default in Table 10).

Various biogeochemical tracers and parameters are transferred from SEDGEM to OMEN-SED (see Fig. 9) and are converted

into the required units. Bottom water concentrations of solutes are converted from $mol\,kg^{-1}$ to $mol\,cm^{-3}$ and the depositional flux of POC ($POC_{flux}$) is converted from $cm^3\,cm^{-2}\,yr^{-1}$ to $mol\,cm^{-2}\,yr^{-1}$ assuming an average density of POC of $1.0\,g\,cm^{-3}$. Within the water column in cGENIE, POC is partitioned into two fractions with different degradation length scales of $\sim 590\,m$ and $1000000\,m$, respectively. The labile pool thus degrades while sinking through the water column, whereas the refractory pool is assumed relatively unreactive (Ridgwell et al., 2007). Thus, depending on seafloor depth, the partitioning of bulk

POC reaching the sediments is different (Fig. 10A+B). This information is used by OMEN-SED to define the parameter $f_1$. Other parameters used from cGENIE are seafloor depth and local temperature. The advection/burial rate ($w$) is taken from the previous time-step of cGENIE, however, it is assured that $w$ is not smaller than the detrital flux ($Det_{flux}$) to the sediments (e.g. $w < 0$ can occur if initially carbonate rich sediments are eroded during the spin-up of cGENIE). In case ($w \leq Det_{flux}$ & $Det_{flux} = 0.0$) all POC is remineralised at the ocean floor. Furthermore, a minimum value of $w = 0.4\,cm\,kyr^{-1}$ is imposed

as OMEN-SED tends to be less stable for lower values. For comparison, this threshold is crossed for seafloor depths below 7000 m when applying the relationship between burial rate and water depth of Middelburg et al. (1997) and below 5200 m for the Burwicz et al. (2011) parameterisation. The bulk $POC_{flux}$ is separated into the labile and refractory component and the routine to find the steady-state solution for POC is called. Here, the two POC depositional fluxes are first converted into SWI





concentrations ($\text{POC}_i(z = 0)$, in $\text{mol cm}^{-3}$) by solving the flux divergence equation:

$$\frac{\partial F}{\partial z} = -\frac{\partial}{\partial z} \left( -\xi D_i \frac{\partial \text{POC}_i}{\partial z} + \xi w \text{POC}_i \right) \tag{51}$$

for z=0. OMEN-SED then computes the fraction of POC preserved in the sediment ($f_{\text{POC}}$, see Eq. (5)) and subsequently calls the routines to find the steady-state solutions for the solute substances. Note, that in this initial coupling the calculated benthic

uptake/return fluxes $F_{C_i}$ of dissolved species $C_i$ (compare Eq. (6)) are adjusted for the advective loss at the lower sediment boundary ($w \cdot C_i(z_\infty)$) to assure the conservation of mass in the coupled model:

$$F_{C_i} = \phi(0) \left( D_i \frac{\partial C_i(z)}{\partial z} \bigg|_{z=0} - w \left[ C_i(0) - C_i(z_\infty) \right] \right). \tag{52}$$

In case OMEN-SED computes unrealistic results for POC preservation (i.e. $f_{\text{POC}} < 0.0$ or $f_{\text{POC}} > 1.0$) we discard the results of OMEN-SED and all POC is remineralised at the ocean floor. For the modern ocean set-up, using the adjustments for $w$

described above, this has not occurred and is just installed as a safety check. Finally, $f_{\text{POC}}$ and the SWI-fluxes of solutes ($F_{C_i}$, in $\text{mol cm}^{-2} \text{yr}^{-1}$) are returned to cGENIE. In case no POC is deposited on the seafloor (i.e. $\text{POC}_{\text{flux}} = 0$), OMEN-SED is not executed and $f_{\text{POC}}$ and $F_{C_i}$ for all $i$ are set to zero. In order to reduce memory requirements, the sediment profiles (e.g. as shown in Fig. 7) are not calculated in the FORTRAN version of OMEN-SED, however, the boundary conditions are saved and sediment profiles for specific grid-cells, ocean basins and ocean transects can be plotted at the end of each experiment using

the stand-alone MATLAB version of OMEN-SED.

## 4.2    Parameterising the OM degradation rate constants in a global model

As shown in our sensitivity analysis (Section 3.1) and discussed by Arndt et al. (2013), the degradation rate constants for OM ($k_i$) are the most influential parameters and exert a dominant control on the SWI-flux of redox-sensitive elements as well as the preservation of organic matter. Yet, their spatial variability is unknown at the global scale and reported rate constants in

the sediments can vary by about 10 orders of magnitude or more (Middelburg et al., 1993; Arndt et al., 2013). Furthermore, when OMEN-SED is coupled to cGENIE, very different timescales have to be considered for OM degradation in the sediments compared to the water column (Fig. 10A+B) and thus the diagenetic rate constants cannot be easily implied by the assumed water column POC flux profiles in cGENIE. To illustrate this, lets consider the degradation of fresh, marine organic matter as it is transported and degraded along the ocean-sediment continuum. The bulk material is composed of a complex mixture

of different organic carbon compounds that can be described by a reactivity continuum. Microbes preferentially degrade the more reactive organic matter compounds first (Emerson and Hedges, 1988; Wakeham et al., 1997; Lee et al., 2000), resulting in the preferential preservation of more unreactive compounds, rendering the remaining mixture less and less reactive with time. Thus depending on the age of OM (or depth in the water and sediment column) the reactivity distribution of its compounds changes significantly (Fig. 10C) and the multi-G (2G in this case) approximation of this continuum has to take this shift

into account. Fig. 10 illustrates these changes in the original reactivity distribution within a ocean-sediment framework. The reactivity distribution $t < 1$ year represents the organic matter mixture after it settled through the water column (Fig. 10C). Only the most reactive OM compounds are remineralised. This explains why the POC flux in the ocean can be represented



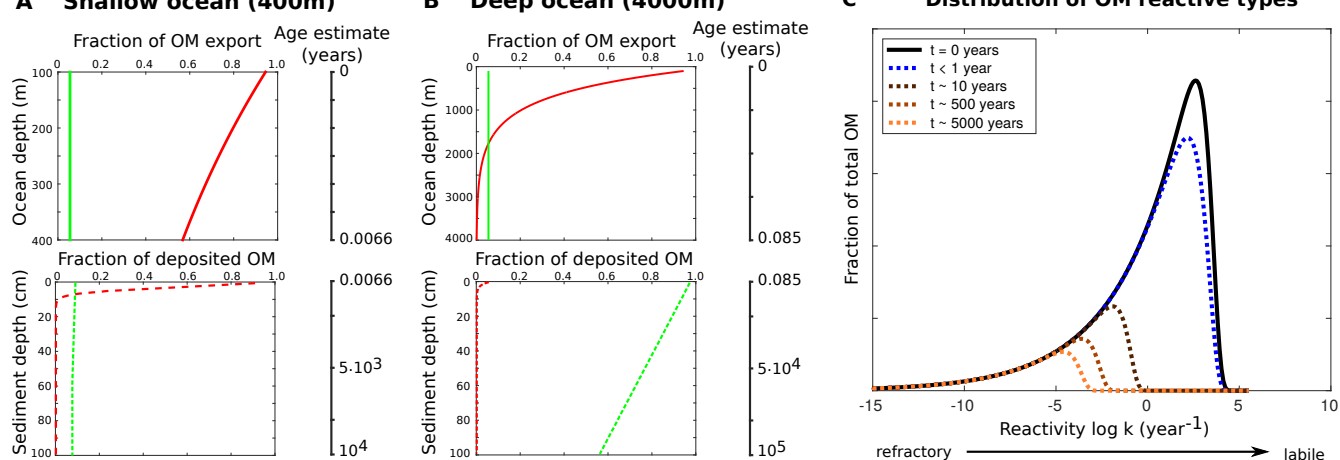

**Figure 10.** Idealised relationship of organic matter decomposition during remineralisation in the water column and the sediments. **A+B - Upper panels:** Water column development of the two organic matter fractions as represented in cGENIE for two ocean depths (red: labile OM with degradation length scale of 589m; green: refractory OM which is unreactive in the water column). The values are normalised to OM export at 100m. Age estimates for the OM since its export from the euphotic zone are calculated using a sinking velocity of 125m/day. **A+B - Lower panels:** Schematic representation of the development of the two OM fractions in the sediments (normalised to OM deposited on the seafloor). For the age estimates in the sediment column an advection rate of 0.01 and 0.001cm/yr is assumed, respectively. **C:** Idealised distribution functions of OM reactive types during remineralisation for different OM ages assuming a reactive continuum model for OM degradation. The initial distribution (at $t = 0$) represents fresh OM when it is exported from the euphotic zone (characterised by $a = 3e^{-4}$ yr$^{-1}$ and $\nu = 0.125$ Boudreau et al., 2008).

with a 1G or pseudo 2G degradation model. In the sediments, however, much longer timescales have to be considered and a wider range of more unreactive compounds are degraded. As a consequence, significant changes in the reactivity distribution already take place in the upper millimetres of the sediments ($t \sim 10$ years, Fig. 10C). Therefore, a broader range of OM reactive types must be represented by the degradation model to capture the reactivity spectrum of OM in surface sediments, explaining

5  why at least a pseudo 3G model (including two degradable and one refractory fraction Soetaert et al., 1996; Boudreau, 1997; Stolpovsky et al., 2015) is required. To complicate the situation even further, different sediment depths can represent very different timescales. For instance, half a meter of sediment can be deposited within a year in a coastal setting, while it will represent thousands of years (if not more) of sedimentation in a deep ocean setting. Therefore, residence times and thus degradation rate timescales (or OM age) are mainly controlled by advection rates. For instance, assuming an advection rate of

10  0.01 cm/yr for the shallow ocean, OM at 5cm depth is has been degraded for approximately 500 years, whereas a deep ocean advection rate of 0.001 cm/year allowed for OM degradation of approximately 5000 years at the same depth. As a consequence, organic matter degradation in deep ocean sediments affects a much wider range of the reactivity continuum and our simple pseudo 3G approximation of the complex OM mixture needs to reflect this by allowing for different k and f values (Fig. 10C).



Thus defining appropriate OM degradation rate constants is a major challenge and source of uncertainty for diagenetic models. The rate constants in models are either determined through profile fitting for a specific site or, for global applications, they are related to a single, readily available characteristic (or master variable) of the local environmental conditions. For instance, considerable effort has been expended to relate the apparent rate constant for oxic and anoxic OM degradation to

sedimentation rate ($w$) and various empirical relations have been proposed (Toth and Lerman, 1977; Tromp et al., 1995; Boudreau, 1997; Stolpovsky et al., 2015). Nevertheless, these relationships are generally based and/or tested on limited data sets and their global applicability, especially under past or projected future environmental conditions is questionable (Arndt et al., 2013). We hence test several alternative schemes in the coupled OMEN-cGENIE framework. Our objective is not to perform and discuss a detailed calibration of the coupled models as this is beyond the scope of this sediment model development

paper. Rather we want to showcase the feasibility of the model coupling, illustrate the range of results and thus information that can be generated with OMEN-SED and verify that model results capture the main observed global benthic biogeochemical features.

### 4.2.1 Methodology

In this section we compare modelled mean POC weight percentages (wt%) in the upper 5cm of the sediments ($POC_{5cm}$)

to the global distribution pattern of POC content in surface sediments (< 5cm sediment depth) of Seiter et al. (2004) using different parameterisations for the degradation rate constants $k_1$ and $k_2$. For our observational target we take the original POC distribution pattern in $1° \times 1°$ grid resolution (interpolated from > 5500 measurements, compare Seiter et al., 2004) and transform it onto the $72 \times 72$ SEDGEM grid (Figure 11). The regridding of the original POC distribution obviously affects the resolution of the data, especially for the continental margin, as some sites with higher POC wt% are lost in the regridding

process (compare e.g. maximum values for the East Pacific and upwelling waters of the Namibian shelf, Figure 11A + B). The colour of the points in Figures 12 - 14 indicates the seafloor depth (SFD) of the respective cGENIE grid-cell. As the individual data-points are highly scattered and in order to see if a certain relation between $k_1$ and $k_2$ performs better for specific ocean depths, the data-points are binned into 6 uniform depth-classes of 1000m each (respective mean POC wt% and SFD are represented by the triangles). The regression line (and the corresponding $R^2$-value) is calculated for the 6 bin-classes

and included in the figures.

To parameterise the reactivity of organic matter in OMEN-SED two different schemes are tested and compared. First, spatially uniform degradation rate constants $k_1$ and $k_2$ are assumed. By simulating two different pools of POC in the water-column characterised by different degradation length scales (Ridgwell et al., 2007), cGENIE implicitly accounts for the decrease in mean POC reactivity with water-depth. The rate constants for the more refractory OM pool, $k_2$, is systematically varied be-

tween 0.004 and 0.006 year$^{-1}$ and the more labile OM component, described by $k_1$, is assumed to degrade a multiple times faster (i.e. $x \in \{1.1, 1.2, 1.3, 1.5, 2\}$ ). However, although accounting for the decrease in mean POC reactivity with seafloor depth, this approach does not take into account the change in distribution of organic matter reactivity types caused by different burial velocities and thus different residence time scales in the sediments (Fig. 10). Therefore, the second approach uses the empirical relationship proposed by Boudreau (1997), which relates the apparent OM degradation rate constant in the upper



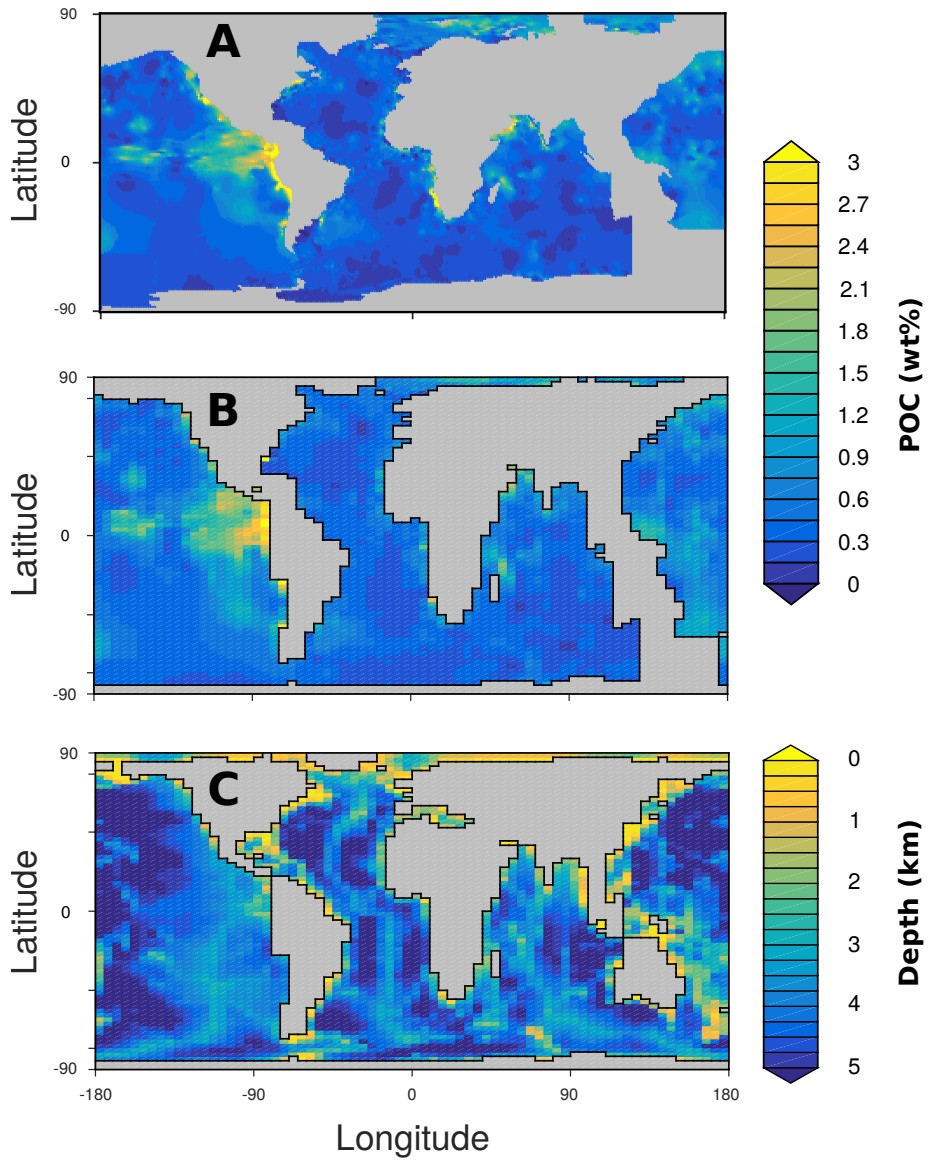

**Figure 11.** Observed distribution of sediment surface (< 5cm) POC wt% (A, B) and cGENIE bathymetry (C). (A) Original global distribution of POC wt% interpolated on a $1° \times 1°$ grid from more than 5500 individual data points (compare Seiter et al., 2004, for the interpolation procedure). (B) Observed POC wt% data transformed onto the $72 \times 72$ SEDGEM grid. Grid points without any observations are left blank (grey). (C) Gridded continental configuration and ocean bathymetry of the 16-level, $72 \times 72$ equal-area cGENIE grid.





sediments to the burial velocity, $w$ (cm year$^{-1}$, see also Section 3.3):

$$k_{\mathrm{app}} = 0.38 \cdot w^{0.59}. \tag{53}$$

Following Boudreau (1997) and Stolpovsky et al. (2015) it can be assumed that $k_{\mathrm{app}}$ represents the mean OM reactivity within the upper 10-20cm of the sediments. The following assumptions are made in order to calculate the two degradation rate constants for OMEN-SED:

$$k_{\mathrm{app}} = f_1 \cdot k_1 + (1 - f_1) \cdot k_2 \tag{54}$$
$$k_1 = x \cdot k_2 \tag{55}$$

where $x$ describes the relation between $k_1$ and $k_2$ and is subject to sensitivity experiments (with values of $x \in \{2, 5, 8, 10, 12, 15, 20, 25\}$). Note that the differences between $k_1$ and $k_2$ using this approach is significantly larger as in the globally uniform approach. As the fractions of labile and refractory OM reaching the sediments $f_1$ is known from cGENIE, $k_1$ and $k_2$ can be calculated independently for each grid-cell.

To simulate steady state sediment composition we configure the model as a "closed" system, i.e., one in which there is no loss of $CaCO_3$ through burial. The redox dependent P-cycle in OMEN-SED is not used in these experiments and all organic phosphorus is returned at the seafloor. To speed up the calculation and to assure that ocean redox changes caused by OMEN-SED do not impact the sediment composition of $CaCO_3$, we use the prescribed solid fields as described earlier. Apart from the prescribed fields and the $72 \times 72$ sediment grid the model is configured as in Archer et al. (2009) and atmospheric $CO_2$ is restored to a pre-industrial value of 278 ppmv. First a 20,000 year spinup is performed without OMEN-SED being coupled. All presented coupled cGENIE-OMEN simulations are run for 10,000 years to steady state from this spinup. OMEN-SED is called for each grid-cell in every time step, feeding back the resulting SWI-fluxes and the fraction of POC preserved in the sediments to cGENIE.

### 4.2.2 Results

Figure 12 presents results for the spatially uniform degradation rate experiments. In general, using spatially uniform degradation rate constants 5 of the 6 bin-classes are located closer to the 1:1 line as in the experiments using the Boudreau (1997) relation (Fig. 13). Also the slope of some regression lines is close to 1.0 (e.g. $(k_2, x) \in \{(0.004, 1.5), (0.0045, 1.3), (0.005, 1.2), (0.005, 1.3), (0.0055, 1.1), (0.0055, 1.2), (0.006, 1.1)\}$), indicating that the simpler parameterisation adequately captures the relationship between between depth and observed POC wt% by bin-class. The reason for this is that BIOGEM provides a depth dependent POC flux and partitioning between the two fractions (Fig. 10). The shallowest bin-class (between 0 and 1000m) represents an exception, as OMEN-SED tends to overestimate POC preservation for this depth class. However, this could also be related to the regridding of the original POC distribution pattern of (Seiter et al., 2004) on to the SEDGEM grid, as some data grid-cells with higher POC wt% on the continental margin are lost due to the restricted SEDGEM resolution (compare Section 4.2). Overall, using this parameterisation, a relationship where the labile POC fraction degrades not more than 1.5 times faster than the refractory fraction fits the Seiter et al. (2004) data better than a larger spread between both POC pools





**Figure 12.** Crossplots comparing modelled and observed mean POC wt% in the upper 5 cm of the sediments using spatially uniform degradation rate constants $k_1$ and $k_2$. Data-points are binned into 6 uniform depth-classes of 1000m as in Fig. 13, each class is represented by a triangle. Grid-points with more than 4.0 POC wt% are not shown.





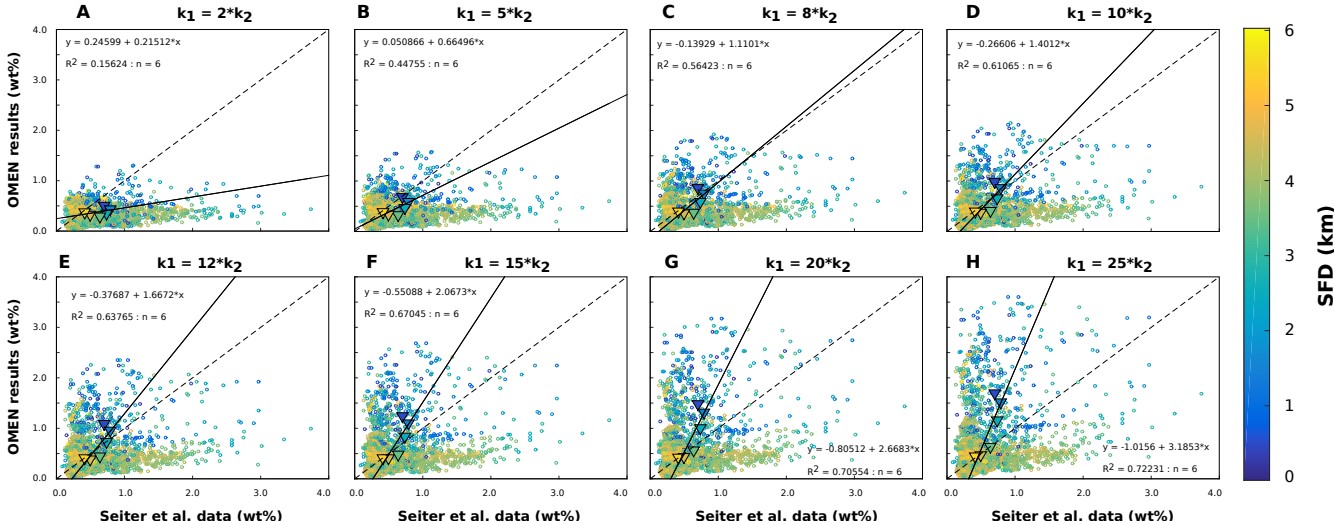

**Figure 13.** Crossplots comparing modelled and observed mean POC wt% in the upper 5 cm of the sediments using the relationship of Boudreau (1997) and the assumptions of Eq. (54) and (55) to calculate $k_1$ and $k_2$. Data-points are binned into 6 uniform depth-classes of 1000m, each class is represented by a triangle. Grid-points with more than 4.0 POC wt% are not shown.

(i.e. $x > 2.0$). We discuss later the implications of collapsing the $k$ values in this specific calibration for long-term, geological carbon preservation.

Next the relationship of Boudreau (1997) and the assumptions of Eq. (54) and (55) are used to calculate $k_1$ and $k_2$. In Figure 13 (A-H) the relation between the two degradation rate constants (Eq. (55)) is changed globally, thus independent of the seafloor depth. The crossplots show that it is not possible to achieve a solution where all bin-classes fall onto, or close to, the 1:1 line. Also, the slope of the regression lines are generally much larger or smaller than 1.0 (with the exception of Figure 13C), indicating that the relationship between depth and observed POC wt% by bin-class is not adequately represented by the model. The $R^2$ values are strictly monotonically increasing for increasing $x$ because a depth-dependency is artificially imposed for the modelled POC wt% through the relation between $k_1$ and $k_2$. When looking at the individual bin-classes it can be seen that shallow ocean depths are better represented by smaller differences between $k_1$ and $k_2$ (e.g. $k_1 = 5 \cdot k_2$ for SFD < 1000m, Figure 13B), and the deep ocean by a larger spread (e.g. $k_1 = 25 \cdot k_2$ for SFD > 3000m, Figure 13H). These results reflect the preferential degradation of more reactive organic matter types (Wakeham et al., 1997; Lee et al., 2000) and thus the change in the distribution functions of OM reactive types for different OM ages (Fig. 10C). In the shallow ocean bulk POC consists of fresher organic matter types on average and is therefore generally more reactive overall (i.e. higher $k_{app}$ due to higher $w$ in the model) as in the deep ocean. In addition, OM at 5cm sediment depth in the deep ocean is generally older as in the shallow ocean due to lower burial rates, therefore more reactive types are affected by degradation and a larger spread between k-values is needed to capture these dynamics (compare Fig. 10C).





**Figure 14.** Mean POC concentrations in the upper 5cm of the sediments ($POC_{5cm}$) using the globally uniform model ($k_2 = 0.005$, $k_1 = 1.3 \cdot k_2$) and the depth dependent parameterisation $k_1 = x(SFD) \cdot k_2$ adapted from Boudreau (1997). A+B: Crossplots as shown before in Fig. 12 and 13. C+D: Histograms of the residuals of modelled minus observed $POC_{5cm}$. E+F: $POC_{5cm}$ as calculated with OMEN-SED. G+H: Difference map of $POC_{5cm}$ as calculated with OMEN-SED and interpolated data from Seiter et al. (2004).

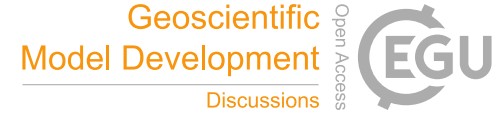

Departing from our theoretical considerations (see discussion of Fig. 10C), we use these observations to create a depth dependent relationship between the two degradation rate constants, where $x$ in Eq. (55) is a function of SFD and takes values of $x = 5$ for SFD < 1000m, $x = 8$ for 1000m $\leq$ SFD < 2000m, $x = 12$ for 2000m $\leq$ SFD < 3000m and $x = 25$ for SFD $\geq$ 3000m for the 6 SFD bin-classes, respectively. In this depth dependent approach all bin-classes are close to the 1:1 line and the resulting regression model accounts for 92.6% of the variance of the modelled POC wt% around the observed mean of the bin-classes (Figure 14B). Furthermore, the slope of the regression line (0.9662) indicates that the relationship between depth and observed POC wt% for the bin-classes is well represented by the model. The histograms (Fig. 14C+D) visualize the difference between modelled and observed mean POC concentrations and demonstrate the high density of data points close to the 1:1 line. For the depth dependent approach, 92.5% of the cGENIE grid-cells show a difference between modelled and observed POC concentration of less than 1.0 POC wt%; in 79.9% of the grid-cells, the difference is less than 0.5 POC wt% (for the globally uniform approach the percentages are 95.37% and 70.95%, respectively).

Both experiments reproduce minimal POC concentrations in the subtropical gyres and generally higher concentrations along the continental margins (Fig. 14E+F). Both experiments, however, underestimate mean POC wt% in the surface sediments of the equatorial east Pacific and overestimate POC concentrations in the North Pacific and Southern Oceans (Fig. 14G+H). The depth dependent approach of Boudreau (1997) shows more spatial variability in POC preservation than the other parameter-isation. In general, implementing lower, anaerobic degradation rate constants when bottom water oxygen concentrations fall below a threshold value could potentially improve the simulation of higher POC concentrations in areas with high POC input to the sediments (Palastanga et al., 2011).

### 4.3 Modelled fluxes and sediment characteristics

For the depth dependent Boudreau (1997) approach modelled SWI-fluxes and sediment characteristics are shown in Figure 15. Modelled total POC degradation ($POC_{degr}$) rates in the upper sediments decrease from the shelves to the deep sea by up to 2 orders of magnitude (Fig. 15B). This is in agreement with data from the literature (e.g. Middelburg et al., 1993, 1997; Burdige, 2007) and other model results (e.g. Thullner et al., 2009) which indicate that the highest degradation rates in marine sediments are found in the coastal ocean (SFD < 200 m). Oxygen fluxes into the sediments (Fig. 15C) range from 0.0 for the deep ocean and sites without OM deposition to values of about 300 $\mu$mol cm$^{-2}$yr$^{-1}$ for the shallow ocean with the highest POC degradation rates. Influx of SO$_4$ into the sediments is rather low (between 0.0 and 23.9 $\mu$mol cm$^{-2}$yr$^{-1}$) because in OMEN-SED 95% of produced H$_2$S is reoxidised to SO$_4$, therefore sulfate reduction is mainly driven by in situ sulfide oxidation. However, in general the coupled model fluxes fall well within the ranges predicted by the stand-alone global hypsometry experiments (O$_2$ between 0.0 and 800 $\mu$mol cm$^{-2}$yr$^{-1}$ and SO$_4$ between 0.0 and about 300 $\mu$mol cm$^{-2}$yr$^{-1}$, compare Section 3.3). In accordance with the total POC degradation rates the release of PO$_4$ shows a maximum value of 8.12 nmol cm$^{-2}$yr$^{-1}$ on the shelves (Fig. 15D). The relative contribution of aerobic POC degradation in the upper sediments increases from the shelves to the deep sea (Fig. 15G) which is also consistent with estimates from Thullner et al. (2009) who found that oxygen is responsible for less than 10% of $POC_{degr}$ at 100 m SFD and for more than 80% in the deep sea. The oxygen penetration depth in OMEN-SED increases from values below 1cm at the shelves to more than 10cm in the deep ocean (Fig. 15H and 16). Small



## Boudreau $k_1 = x(\text{SFD}) * k_2$

**Figure 15.** Sediment characteristics related to POC degradation and oxygen consumption for the depth dependent paramaterisation after Boudreau (1997) with $k_1 = x(\text{SFD}) \cdot k_2$. Total $\text{POC}_{\text{degr}}$ rate and fraction of aerobic $\text{POC}_{\text{degr}}$ are the respective values for the first 5cm in the sediments.





oxygen penetration depths of a few millimetres are typical for bioturbated sediments in the coastal ocean (e.g. Wenzhöfer and Glud, 2002) and the oxygen penetration depth has been shown to increase rapidly with SFD to more than 10 cm in the deep sea (Meile and Van Cappellen, 2003; Glud, 2008). Fischer et al. (2009) and D'Hondt et al. (2015) even found cores along a transect in the South Pacific gyre being oxygenated over their entire length (up to 8 m or even 75 m, respectively) which is consistent

5  with our model results (not shown). Simulated mean oxygen penetration depths for the 6 depth bin-classes also agree well with observations compiled by Glud (2008) and Meile and Van Cappellen (2003, Fig. 16).

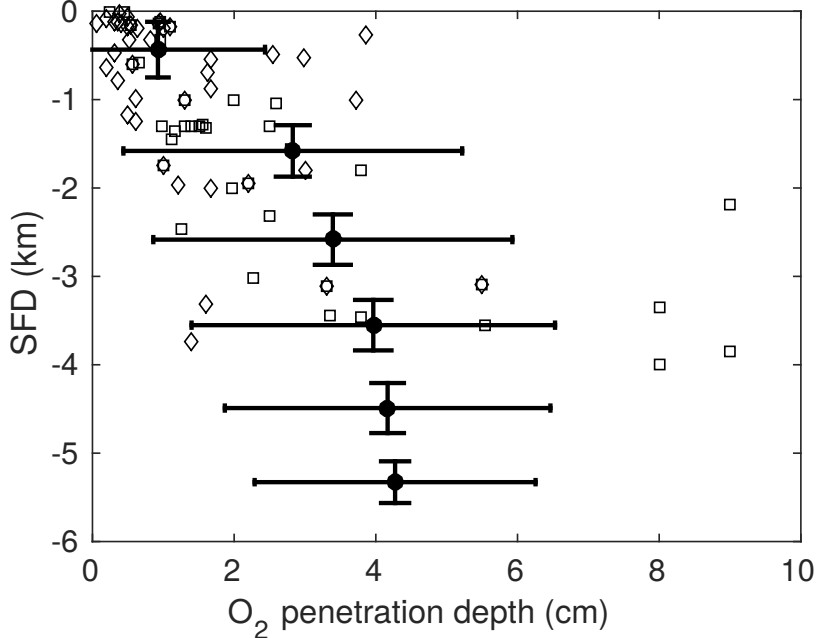

**Figure 16.** Seafloor depth versus $O_2$ penetration depth for the depth dependent paramaterisation after Boudreau (1997) with $k_1 = x(\mathrm{SFD}) \cdot k_2$. Diamonds represent observations compiled by Meile and Van Cappellen (2003) and squares observations from Glud (2008). Circles are the mean model results for the 6 SFD bin-classes (with standard deviations). Grid-cells where the entire sediment column is oxic (i.e. $z_{\mathrm{ox}} = 100\mathrm{cm}$) are not considered in these statistics (17, 32, 102, 300, 477 and 307 cells for the 6 bin-classes, respectively).

## 5   Scope of applicability and model limitations

Because of the high computational cost associated with resolving benthic dynamics, most Earth system Models of Intermediate Complexity (EMICs) and also some of the higher resolution Earth system models either completely neglect or merely include a

10  highly simplified representation of benthic-pelagic exchange processes (Hülse et al., 2017). However, benthic-pelagic coupling plays an important role for carbon cycling and the lack of its representation in EMICs compromises our ability to assess the response and recovery of the Earth system to major past, present and future carbon cycle and climate perturbations. As a conse-





quence, there is a need for benthic biogeochemical models that are able to capture the main features of benthic biogeochemical dynamics, but that are, at the same time, computationally efficient enough to allow for a direct, dynamic coupling to ocean biogeochemical model. Therefore, we have developed OMEN-SED, a, one-dimensional analytical early diagenetic model that offers a predictive ability similar to complex, numerical diagenetic models at a significantly reduced computational cost.

OMEN-SED is thus not problem-specific. Its reaction network resolves the most pertinent benthic biogeochemical species as well as the most important processes that control their cycling and burial in marine sediments. OMEN-SED can thus be coupled to a wide range of regional to global ocean biogeochemical models, as well as EMICs and higher resolution Earth system models to investigate a wide range of research questions associated with past, present or future carbon and macro-nutrient cycling. For instance, OMEN-SED can be used to i) quantify benthic macro-nutrient recycling from the shallow coastal to the

deep, open ocean, ii) investigate the role of benthic-pelagic coupling in the development of past or future ocean anoxia/euxinia, or to iii) estimate global organic carbon burial in marine sediments. In theory, its scope of applicability thus ranges from the regional to the global and from the seasonal to the millennial time-scale. In order to simulate organic matter preservation in the deeper sediments and thus addressing questions concerning long-term, geological carbon burial the degradation rate constant for the refractory OM pool has to be scaled down. The resulting larger difference between degradation rate constants can be

interpreted as being needed to capture the broader range of OM reactivities degraded over the entire sediment column (see Fig. 10). Instead, more collapsed degradation rate constants are needed to model OM degradation in the upper sediments, such as the first 5 centimetres as shown in Section 4.2.2. In addition, the computational efficiency of OMEN-SED allows calculating quantitative sensitivity indices requiring large sample sizes such as variance- or density-based approaches. Therefore, OMEN-SED can also help quantitatively investigate sensitivity of benthic model output to systematic variations in model parameters

when the model has been tuned to a site-specific problem.

However, OMEN-SED is inevitably associated with a certain degree of simplifications that may compromise the applicability of the model in its current version under certain circumstances. First, we have assumed steady state conditions to allow for an analytical solution of the coupled diagenetic equations. This steady-state assumption is only valid if the variability in boundary conditions and fluxes is generally longer than the characteristic timescales of the reaction-transport processes. As a

consequence, OMEN-SED is well suited for the coupling to EMICs and the investigation of long-term dynamics in sediment-water exchange fluxes, for instance, during past extreme climate events. Yet, in its current version, OMEN-SED is not able to predict the transient response of benthic process rates and fluxes to short-term or seasonal variations of boundary conditions. Yet, future versions of OMEN-SED, could approximate non-steady state conditions by incorporating a time-step dependent relaxation between different steady states, similar to the schemes used in Ruardij and Van Raaphorst (1995) and Arndt and

Regnier (2007). Such a pseudo-transient approach would enable the application of OMEN-SED to systems characterised by high-frequency fluctuations in boundary conditions, such as the coastal ocean or estuaries.

Second, although the model explicitly simulates DIC and alkalinity production and, thus, has the potential to predict pH profiles within the sediment, a major limitation at this stage is the lack of an explicit description for $CaCO_3$ precipitation/dissolution coupled to OM decomposition, which also controls the inorganic carbon system (Krumins et al., 2013). In

addition, the current version of OMEN-SED does not yet explicitly resolve iron and manganese dynamics (although note, that





iron is implicitly accounted for in the PO4 equation). This lack currently limits the applicability of OMEN-SED to iron- and manganese rich environments, such as coastal marine environments, upwelling regions or ferruginous oceans. In addition, it also compromises the ability of OMEN-SED in predicting $H_2S$ fluxes in Fe-rich anoxic environments, where high iron pore water concentrations can deplete pore water $H_2S$ by iron-sulfide mineral precipitation (e.g. Meyers, 2007). Therefore, already planned future extensions of OMEN-SED include an explicit description of iron.

Finally, just as all global models, the global application of OMEN-SED is complicated by the lack of an objective, global framework for biogeochemical process parameterisation. The sensitivity study presented here shows that this lack is particularly critical for OM degradation rate parameters ($k_i, f_i$) and the $\gamma$-values describing the completeness of secondary redox reactions. A comparison between simulated OM contents and observations indicates that depth dependent k-f relationships provide the best fit (Section 4.2.2), confirming more theoretical considerations regarding the different time and reactivity scales that need to be considered (see Section 4.2). With respect to $\gamma$-values, model simulations along the global hypsometry (Section 3.3) have shown that high $\gamma$-values generally capture the main SWI-flux features, but have also highlighted that slightly lower $\gamma$-values would result in a better fit of SWI-fluxes to observations of the shallow ocean.

## 6   Conclusions

Here, we have described in detail and tested OMEN-SED, a new, analytical early diagenetic model resolving organic matter cycling and associated biogeochemical dynamics. OMEN-SED has been explicitly designed for the coupling to EMICs and combines biogeochemical complexity with computational efficiency. It is the first analytical diagenetic model to explicitly represent oxic degradation, denitrification, sulfate reduction and implicitly methanogenesis, as well as the reoxidation of reduced substances, adsorption/desorption, as well as mineral precipitation/dissolution. Explicitly resolved pore water species include $O_2$, $NO_3$, $NH_4$, $SO_4$, $H_2S$, DIC and ALK and the solid phase includes two degradable fractions of organic matter, Fe-bound P and authigenic Ca-P minerals.

An extensive sensitive analysis, based on the density-based PAWN method (Pianosi and Wagener, 2015) emphasizes the importance of OM degradation rate parameters ($k_i$, $f_i$) and thus highlights the need for the development of an objective, global framework to parameterize OM degradation rate parameters. We have shown that the performance of OMEN-SED is similar to that of a fully formulated, multi-component numerical model. The new analytical model is able to reproduce observed pore water profiles across a wide range of depositional environments and captures observed global patterns of SWI-fluxes, oxygen penetration depths, biogeochemical reaction rates, as well as surface sediment organic matter contents. Coupled to EMICs or higher resolution Earth system models, OMEN-SED is thus well suited to examine the role of sediments in global biogeochemical cycles in response to a wide range of past or future carbon cycle or climate perturbations over various timescales.



*Code availability.* The commented OMEN-SED source code (FORTRAN and MATLAB) is provided as a supplement to this article and is also available for download on the web (https://github.com/DomHu/OMEN-SED.git). A ReadMe file for the stand-alone MATLAB version of OMEN-SED describes the source code files and includes instructions for executing the model and plotting the results.

*Competing interests.* The authors declare that they have no conflict of interest.

5   *Acknowledgements.* We thank Claire Reimers, Filip Meysman, Martin Thullner, Jack Middelburg, Andy Dale, Katherina Seiter, Christof Meile, Ronnie Glud and the British Oceanographic Data Centre for supplying the datasets and model results used in Sections 3.1, 3.2, 3.3 and 4. We are also grateful to Francesca Pianosi for helpful insights into sensitivity analysis. DH was supported by a graduate teaching studentship by the University of Bristol and a Heising–Simons Foundation award. SA acknowledges funding from the UK Natural Environmental Research Council (NERC) grant no. NE/I021322/1 and SD from the grants NERC JET (NE/N018508/1) and NERC BETR (NE/P013651/1).

10  SA and PR were supported by funding from the European Unions Horizon 2020 research and innovation programme under the Marie Skłodowska-Curie grant agreement no. 643052 (C-CASCADES). AR was supported by a Heising–Simons Foundation award, and by EU grant ERC-2013-CoG-617313.



**Appendix A: Reaction Network**





**Table A1.** Primary pathways of organic matter degradation, secondary redox reactions and stoichiometries implemented in the reaction network.

| Pathway | Stoichiometry |
| --- | --- |
| | **Primary Redox reactions** |
| Aerobic degradation | $(CH_2O)_x(NH_3)_y(H_3PO_4)_z + (x+2y)O_2 + (y+2z)HCO_3^- \rightarrow (x+y+2z)CO_2 + yNO_3^- + zHPO_4^{2-} + (x+2y+2z)H_2O$ |
| Denitrification | $(CH_2O)_x(NH_3)_y(H_3PO_4)_z + \frac{(4x+3y)}{5}NO_3^- \rightarrow \frac{(2x+4y)}{5}N_2 + \frac{(x-3y+10z)}{5}CO_2 + \frac{(4x+3y-10z)}{5}HCO_3^- + zHPO_4^{2-} + \frac{(3x+6y+10z)}{5}H_2O$ |
| Sulfate reduction | $(CH_2O)_x(NH_3)_y(H_3PO_4)_z + \frac{x}{2}SO_4^{2-} + (y-2z)CO_2 + (y-2z)H_2O \rightarrow \frac{x}{2}H_2S + (x+y-2z)HCO_3^- + yNH_4^+ + zHPO_4^{2-}$ |
| Methanogenesis | $(CH_2O)_x(NH_3)_y(H_3PO_4)_z + (y-2z)H_2O \rightarrow \frac{x}{2}CH_4 + \frac{x-2y+4z}{2}CO_2 + (x-2z)HCO_3^- + yNH_4^+ + zHPO_4^{2-}$ |
| | **Secondary Redox reactions** |
| Nitrification | $NH_4^+ + 2O_2 + 2HCO_3^- \rightarrow NO_3^- + 2CO_2 + 3H_2O$ |
| Sulfide oxidation | $H_2S + 2O_2 + 2HCO_3^- \rightarrow SO_4^{2-} + 2CO_2 + 2H_2O$ |
| AOM | $CH_4 + CO_2 + SO_4^{2-} \rightarrow 2HCO_3^- + H_2S$ |
| | **Adsorption reactions and mineral precipitation** |
| $NH_4$ adsorption | $NH_4^+ \xrightarrow{K_{NH_4}} NH_4^+ (ads)$ |
| P ad-/desorption | $PO_4^{2-} \xrightarrow{K_{PO_4}^{I,II}} PO_4^{2-} (ads); \quad PO_4^{2-} \xrightarrow{k_s} Fe-bound\ P \xrightarrow{k_m} PO_4^{2-}$ |
| CFA precipitation | $PO_4^{2-} \xrightarrow{k_a} CFA$ |





## Appendix B: Sensitivity Analysis

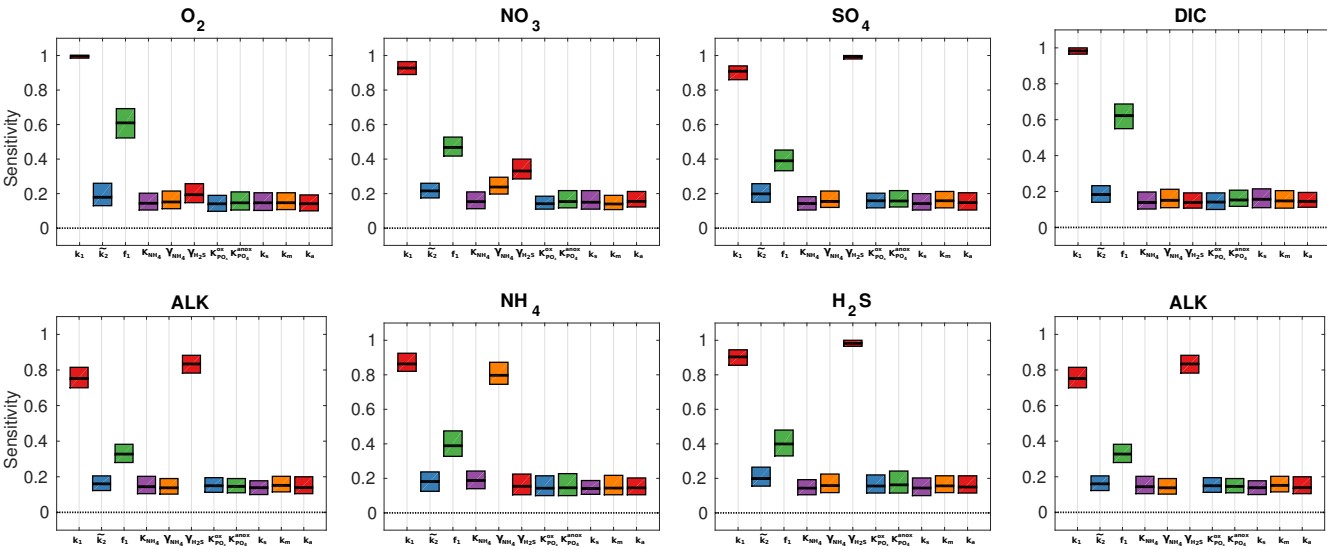

**Figure B1.** Box plot of parameter sensitivities for the calculated SWI-fluxes for the 4000m oxic condition. Average sensitivities (black lines) and 90% confidence intervals using $N = 11200$ model evaluations and $Nboot = 100$ bootstrap resamples.





## Appendix C: Prescribed burial flux fields

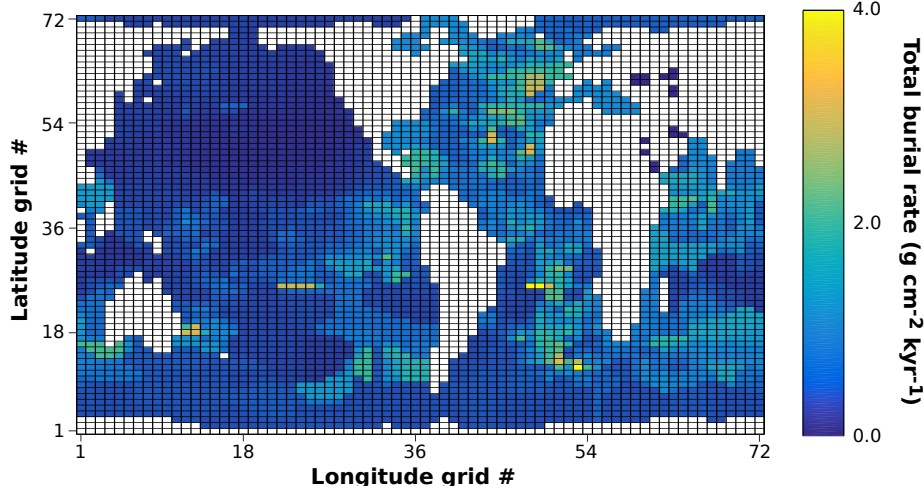

**Figure C1.** Distribution of prescribed total burial fluxes of detrital material, opal and $CaCO_3$ (in g cm$^{-2}$ kyr$^{-1}$), re-gridded from the data compilation of Archer (1996) using a method explained in the text. Note, latitude and longitude are shown in cGENIE grid-cells and not in degrees.





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
