# Peer review of "OMEN-SED 1.0: A novel, numerically efficient organic matter sediment diagenesis module for coupling to Earth system models"

_Geoscientific Model Development, 2017_

## Referee Comment (RC1) · Anonymous Referee #1 · 21 Feb 2018

This manuscript presents an early diagenetic model that can be coupled to an Earth System Model to study biogeochemical cycling over 'climate relevant' timescales. It starts with a nice introduction and overview of the role of early diagenetic processes. Then the model is described, an extensive sensitivity analysis is presented, the model is applied to a number of sites, and then compared to numerical models. Finally, in section 4, an application in which the model is coupled to an Earth System Model is presented.

Overall, I consider the approach which relies on analytical solutions of simplified set of early diagenetic processes promising and useful, and the description of the model is

clear. Below I outline my criticism. The model builds on numerous assumptions which likely limits its applicability in the coastal ocean, where most of the mineralisation and benthic-pelagic coupling takes place.

Model formulation - The model assumes no overlap of mineralization reactions with different terminal electron acceptors, and assumes that secondary redox reactions can be collapsed onto the interfaces between different mineralization zones (p.9). This is probably ok in environments typically encountered at greater water depth, but there is ample evidence of 'overlapping' mineralization pathways in surficial sediments, in particular in permeable or bioturbated settings. - In the denitrification layer, all N goes to N2. However, in the suite of processes involved in the breakdown of organic matter, ammonium is produced as well, even if nitrate serves as TEA. Ammonium produced in the denitrification zone would need to be accounted for at the transition to the oxic zone as well, which may further reduced the O2 penetration depth. Also, processes such as DNRA or anammox are not included, even though literature surveys (e.g. Dalsgaard et al. 2005) indicate that anammox is relevant at shallower water depth. - Methane oxidation: All methane is assumed to be oxidized anaerobically. Is this done for simplicity? Is there no leakage term (gamma_CH4; comparable to incomplete sulfide and ammonium oxidation) because methane escaping from the anoxic zone is assumed to be removed by aerobic methane oxidation? - Methanogenesis: the 1/2 methane to DIC ratio seems to imply acetoclastic methanogenesis. What evidence is there to ignore hydrogenotrophic methanogenesis? - For a globally applicable model, the lack of CaCO3 dissolution is an obvious issue. Thus, can you expand on what problems the modeling of CaCO3 dissolution would cause (page 18)? Is this linked to the calculation of pH? Why can pH (and then carbonate) not be estimated from DIC and alkalinity? - Is there no P sorption on iron oxides below the oxic zone? If so, why? - Why does some of the ammonium created below the oxic zone escape oxidation, but oxidation of ammonium to nitrate is complete in the oxic zone? - A fraction of the sulfide produced is assumed to escape complete oxidation. Does this mimicking the effect of precipitation with iron, rather than escape from the sediment? - Iron cycling is not

Interactive
comment

represented explicitly but some of the effect of iron cycling is parameterised. With its effect on sulphur cycling, P sorption and C mineralization (metal reduction can be the main mineralization pathway, see e.g. Canfield et al. 1993), I don't fully understand the reason for doing so (apart from the added complexity when dealing with another solid phase). - transport processes: A 1D diffusion/bioturbation model clearly faces major challenges in the coastal ocean, where sediments one predominantly are permeable. And setting fir=1, implying no bioirrigation, is also a very strong assumption. - The 2G model is a sensible choice. Model parameterisation is a general concern (page 3, line 30), but I suggest to cite some promising new approaches to address this issues such as presented by https://www.biogeosciences-discuss.net/bg-2017-397

site comparison case studies: - Table 13: fix the depths for the SB and IM sites (SB is the 585m site ...). - Why are the stoichiometric factors set to default values when Epping et al. provide the C/N ratio of the surface sediment? - If a non-local exchange mechanism resulting from bioirrigation is invoked for the Canyon site, what is the source of the high ammonium (and DIC) leading to the observed increase in concentration at depth?

- global transect case study: I don't see the value of the 5% reoxidation case. This is simply unrealistic for the conditions considered here. A possible explanation why lower gammas are giving better match to the data in the shallow sites is that under those conditions, the conceptual model of a vertical separation of reaction zones is more and more violated, so the gamma become a 'fudge-factor' to account for this (see also above comments about the coastal ocean). - on page 43, it says that if fPOC is computed to be > 1 (more than 100% is preserved), then this is discarded and all POC is remineralized. Imposing constraints is fine, but what is the rationale for jumping from >100 to 0 % preservation?

- link to cGENIE. Page 44 discusses the challenges in applying the model in such a setting. In addition, deposition fluxes may change over time. At what point is the steady state assumption on the POC profile still valid under such settings? This is addressed

summarily at of the bottom of page 54. However, I think it is important to lead with this, before interpreting the data-model comparison.

On p.54 it says "In theory, its scope of applicability thus ranges from the regional to the global and from the seasonal to the millennial time-scale. " - in the following paragraph they recognize that "This steady-state assumption is only valid if the variability in boundary conditions and fluxes is generally longer than the characteristic timescales of the reaction-transport processes. "

I recommend to be a little more cautious in the application of the model, since I am not convinced that violations of assumptions underlying the conceptual model and non-steady state effects can be ignored. The model clearly requires substantial tuning. It is clear that the authors are aware of the shortcomings, and they discuss that the model may not be adequate to assess seasonal patterns (and one can think of additional settings, where fluxes vary over timescales intrinsic to the POC profile in the top 50cm of sediment modeled here). My concern is that they largely ignore them in their application, before acknowledging them in the discussion. I also suggest to tone down the finding that "A comparison between simulated OM contents and observations indicates that depth dependent k-f relationships provide the best fit (Section 4.2.2), confirming more theoretical considerations regarding the different time and reactivity scales that need to be considered (see Section 4.2). " The age-reactivity relationship is pretty well established, without confirmation by this modeling effort.

The stated purpose of section 4 is "to showcase the feasibility of the model coupling, illustrate the range of results and thus information that can be generated with OMEN-SED and verify that model results capture the main observed global benthic biogeochemical features." This section indeed illustrates the feasibility of the coupling, but I (and I assume most readers) would not have doubted that in the first place. The manuscript then presents two 2G approaches, in which the rate constants for the mineralization of POC are either fixed or varied based on an empirical relationship with the burial velocity. It is shown that the magnitude of the degradation rate constants

[Figure]

matter. This is clearly expected, because the POC degradation rate is key when the input fluxes are reasonable. Yet, to what extent benthic solute fluxes (other than O2, which is tightly linked to OM mineralisation) are captured in this model is less clear to me. In fact, the sensitivity analysis and figure 8 show that the gamma values matter substantially, and they are in my opinion largely lumping unresolved processes. Hence, they will likely require tuning from site to site. The validation of the coupled model requires more work, and I wonder whether this was not better done in a separate paper, in which the coupling to cGENIE and the parameterization of POC mineralization was explored in more detail.

Figures 12 - 14: I gather the R2 values are for the bin averages. I don't see much value of that, as over- and underpredictions cancel each other out in the averaging. Why not compute statistics for the actual model results with the Seiter data directly?

Minor comments - page 8/ line 1: It is said that all parameters in Eq. 1 may vary with depth, but above it is stated that porosity and burial rates are constant with depth. - the fraction of POC buried is defined as the POC at z=0 relative to the POC at depth. Why is it not defined as the flux at z=0 vs. the flux at depth (it seems Eq. 5 ignores the diffusion flux)? Related to that, page 10, line 18 refers to a concentration/flux boundary conditions at the SWI. The following equations and Table 2 only show a known concentration, not a flux condition. However, the latter would be useful when connecting the sediment model to a model of the water column. On page 43, Eq. 51 this is addressed - make this clear earlier. - Can lines 21-23 on page 21 be deleted? - page 31, line 2: specify NH4, SO4 and H2S FLUXES - Figure 3 did not help me much. Does the green dashed vertical arrow indicate possible locations of zbio?
* * *

---

## Referee Comment (RC2) · Anonymous Referee #2 · 2 Mar 2018

This paper presents a one-dimensional analytical early diagenetic model resolving bio-geochemical processes associated with organic matter degradation in near-seafloor bioturbated marine sediment over short timescale assuming steady state solution. The model parameters are constrained by a database of oxygen and nitrate fluxes trough the sediment-water interface, global distribution pattern of POC content in surface sediments, porewater profiles of solid and dissolved species and the model developed before. The goal of the work is to provide a general model of marine sediments which can be used in ESM due to its very low compared to traditional numerical diagenetic models execution time (less than 0.1 sec). This analytical diagenetic model as well as parameterization procedure and sensitivity analysis are clearly explained, the comparison

of model predictions and measurements is discussed thoroughly and the manuscript presents a full procedure to model POC degradation in bioturbated marine sediments. I do find the work to be novel and important, however, I have some comments that the authors need to address.

Specific comments.

The model neglects the effect of sediment compaction "due to mathematical constraints". I understand the rational for this and accept a consistency of this assumption to near-seafloor (bioturbated) sediments; however, this might be a problem for deeper sediments discussed in the paper (down to 50 or 100cm). The authors should either define different porosity values for different depth-zones or to demonstrate that the results are not particularly sensitive to the value of this parameter.

Dividing the sediment column into functional zones in such a strict manner does not always represent reality well. Thus, "nitrogenous" zone may overlap with "oxic" zone. This assumption, as far as I understand, made it impossible to simulate nitrate SWI flux directed into the sediments in oxygenated environment, which is definitely not true. Validation of the model against measured benthic fluxes would probably demonstrate to some extent accordance of suggested method with real benthic system.

Nitrogen dynamics include "the metabolic production of ammonium, nitrification, denitrification as well as ammonium adsorption". Denitrification is considered as a single-step process ignoring $NO_2^-$ production/consumption and anaerobic ammonium oxidation (Anammox) which is undoubtedly a significant component of the biogeochemical nitrogen cycle (Devol, 2015). In other words, nitrogen dynamics is somewhat simplified. This simplification should be quantified/discussed in more details.

The efficiency of binning procedure discussed in section 4.2.1 is doubtful. First of all, such binning assumes presence of STD bars on the plots. Also, I think that it would be more logical to group POC content into POC rain rate (RRPOC) classes rather than WD classes as RRPOC may significantly vary at different regions of the ocean of the same

[Figure]

WD. Finally this binning gives a false impression of a good POC content fit. I realize that parameterization of multi-G model is beyond the scope of this sediment model development paper, therefore I suggest to use existing way to parameterize multi-G models and validate your model against the databases suggested in those studies (for example Stolpovsky et al., (submitted) https://www.biogeosciences-discuss.net/bg-2017-397/ ).

POC is not a very good constraint, since measured POC is in large part the less reactive stuff that is left over after mineralization of the more reactive fractions. This was shown in Stolpovsky et al., 2015 paper (see the discussion in section 4.3). Fluxes at the SWI are believed to be a better constraint.

Minor comments.

Eq. 1: As a time and depth independent parameter, porosity should be moved out of differential in order to emphasize that it is constant: Porosity*dC/dt instead of d(Porosity*C)/dt.

P. 8, L. 1: It is not immediately clear that the authors are talking about water (not sediment) depth.

Eq. 5: This representation sounds a bit odd. I think $z_\infty$ should be replaced with zmax, as POC content at infinite depth believed to be zero.

P. 9, L. 25: SWI is given without initial explanation.

P. 25, L. 6 – 13: I agree that bioirrigation may enhance SWI fluxes of dissolved species, therefore I do not understand why this way of transport is technically ignored for all water depths (fir=1)?

P. 27, L. 28: PAWN is given without explanation.

Fig. 7: Please add ticks and numbers to X-axis on H2S at 2213 and 4298m and NH4 at 108m. Some plates have very inconvenient ranges on horizontal axis, for example

H2S at 4298m.

Sec. 3.3.2: I do not understand the rational for comparing OMEN-SED results with another model (Thullner et al. 2009). I would suggest comparing it to existing SWI flux database mentioned before (Stolpovsky et al., 2015). Also, reporting global denitrification rate modeled with OMEN-SED and its comparison with previous studies would support the model.

P. 55, L. 24 – 25: Bold assumption, I suggest to avoid such formulations. The major advantage of OMEN-SED is its tremendously low computation time which is so important for ESMs. As always, only two options of the following three can be true the same time: "quickly", "cheaply (super-computer is not needed)" or "qualitatively".

––––––––––––––––––––––––––––

---

## Referee Comment (RC3) · K. Wallmann (Referee) · 6 Mar 2018

Dominik Hülse and co-authors present a new analytical model of early diagenesis that is much more advanced than any other analytical model that has been developed so far. It has some inherent limitations (steady state, no compaction, etc.) but it is clearly a very useful, flexible, and efficient tool. As explained and demonstrated by the authors, it can be coupled to any ocean or earth system model to define biogeochemical fluxes across the seabed. Ocean models have been previously coupled to diagenetic models that are solved numerically. However, the down-core spatial resolution of these models is very coarse and the number of sediment layers is limited in order to save

computation time. Due to these limitations, the numerical models have to assume that organic matter raining to the seabed has a low reactivity. Only under this assumption the coarse-resolution models can generate meaningful and numerically stable results. Analytical models have unlimited spatial resolution and can therefore be used to simulate the rapid degradation of very labile organic matter at the seabed. This aspect is the major advantage of the new model presented by Hülse et al.

There are three critical points that the authors should address:

1. The model ignores sulfide precipitation and pyrite formation. Consequently, dissolved sulfide produced by sulfate reduction and AOM at depth diffuses upward to be either oxidized by oxygen or released into ambient bottom waters. This is a very unrealistic set-up. In most sediments dissolved sulfide is removed from the pore water by pyrite precipitation while the remaining sulfide is oxidized with ferric iron, nitrate and nitrite before it can reach the oxic surface layer or the ocean. Aerobic sulfide oxidation is only important in highly reactive surface sediments where the diagenetic sequence is not maintained but several electron acceptors are used simultaneously. The model is based on the assumption that electron acceptors are used sequentially rather than simultaneously. Hence, it cannot simulate situations where aerobic sulfide oxidation is important but creates high rates of aerobic sulfide oxidation in geological settings where this process does in fact not occur. The authors should try to fix this problem. They could for example abandon the model parameter that defines the fraction of dissolved sulfide that escapes into bottom waters. In the modern ocean, sulfide leakage from sediments occurs only in very rare situations and it does not make sense to simulate these anoxic sediments with a model that ignores iron cycling, pyrite formation and sulfide precipitation. The authors could instead introduce a parameter that defines the fraction of sulfide that is precipitated as pyrite and update the alkalinity model accordingly.

2. The authors use an empirical equation by Middleburg et al. (1997) to define burial velocity (w) as function of water depth (Eq. 46). Unfortunately, w is seriously overes-

timates by this equation. As an example, w at 1000 m water depth results as 160 cm kyr-1 applying Eq. 46 whereas the available data indicate global mean rates in the order of 10 – 20 cm kyr-1 for this water depth (Burwicz et al., 2011). The extremely high burial velocities derived from Eq. 46 compromise the TOC concentration and other model results especially when the model is applied at global scale.

3. OMEN-SED is able to reproduce the strong down-core decrease in organic matter reactivity observed in marine sediments by using two or more organic matter fractions with widely different reactivity. This strength is nicely demonstrated in section 3.3 where the authors are able to show that typical pore water profiles are reproduced by the model applying kinetic constants (k1, k2) that span several orders of magnitude (Tab. 13). Subsequently, the authors try to reproduce the TOC distribution at the deep-sea floor by coupling OMEN-SED to an earth system model. I think that TOC in surface sediments is not a good parameter to validate the model because almost the entire organic matter raining to the deep-sea floor is degraded in the surface sediment rather than preserved as sedimentary TOC. TOC concentrations in surface sediments at the deep-sea floor are governed by TOC rain rates, mass accumulation rates (burial velocity), adsorption of organic matter on mineral surfaces, and the kinetic properties of the very small refractory fraction that survives degradation (about 1 % of the total rain rate). The strength of OMEN-SED to degrade the reactive fractions in a meaningful way does not play out in this application. Moreover, the model results are unrealistic. The best fit to the TOC data is apparently obtained assuming that the organic matter flux to the seabed is composed of two TOC fractions with very low reactivity in the order of 0.001 – 0.01 yr-1 (Fig. 12). This result is not consistent with the case study presented in section 3.3 that yields much higher k values (Tab. 13). Moreover, we have shown previously that this very low reactivity is not consistent with the benthic fluxes of oxygen and nitrate that have been measured at the seabed (Stolpovsky et al., 2015). The error may be caused by the too high burial velocities applied in OMEN-SED (Eq. 46) and/or may be related to the rain rate and reactivity of organic matter calculated in GENIE. I would encourage the authors to delete the entire section 4 of the paper because it

does not add useful information but presents rather misleading results. They should aim to present other more useful applications of their highly innovative analytical model in follow-up publications.
* * *

---

## Author Comment (AC1) · 8 May 2018

**Replies to Anonymous Referee #1 on "OMEN-SED 0.9: A novel, numerically efficient organic matter sediment diagenesis module for coupling to Earth system models"**

**1. Comment:**
Model formulation - The model assumes no overlap of mineralization reactions with different terminal electron acceptors, and assumes that secondary redox reactions can be collapsed onto the interfaces between different mineralization zones (p.9). This is probably ok in environments typically encountered at greater water depth, but there is ample evidence of 'overlapping' mineralization pathways in surficial sediments, in particular in permeable or bioturbated settings.

**Response:**
We agree with the reviewer that different biogeochemical zones can overlap. However, as stated in the text, OMEN-SED is designed for the coupling to ESMs and its formulation is thus first and foremost guided by achieving numerical efficiency while retaining biogeochemical reality. As summarized in the manuscript, there are essentially two approaches that can be used to describe biogeochemical processes in models. The first approach solves the general diagenetic equation numerically on a regular or irregular grid and biogeochemical zonation emerges in response to inhibition terms allowing a certain degree of overlap between biogeochemical zones. This approach is highly flexible and thus preferable. Yet, its excessive computational demand unfortunately renders its application within a three-dimensional Earth System Model framework impossible. On the other hand, analytical models that subdivide the sediment into distinct biogeochemical zones are computationally efficient and thus ideally suited to describe diagenetic dynamics in ESM.
By their very nature, analytical models do not allow for overlapping biogeochemical zones. As stated in the manuscript, this is a simplification. However, we disagree with the reviewer that this simplification would *per-se* prevent the application of such analytical approaches in shallower aquatic environments. In fact, OMEN-SED builds on a number of analytical models that were developed to investigate local, coupled nutrient and oxygen cycles in coastal sediments (e.g. Billen, 1982; Goloway and Bender, 1982; Jahnke et al., 1982; Slomp et al. 1996). Similar approaches were later successfully applied from oxic to anoxic sediments and at the regional coastal ocean scale (e.g. Ruardij and Van Raaphorst, 1995; Tromp et al., 1995; Gypens et al., 2008). In particular, Gypens et al., (2008) points out that accounting for secondary redox process in the boundary condition induces little error as: "Using a numerical model, Soetaert et al. (1996) showed that this re-oxidation mainly occurs at the oxic-anoxic transition interface."

Finally, the good agreement between OMEN-SED and the results obtained with a fully formulated numerical RTM (compare Section 3.3, allowing for overlapping TEA use) shows that this is not a critical limitation of OMEN-SED - even for shallow sediments.

We included a little paragraph on this in the limitation section (pg 54 lines 3-8):
*"Furthermore, by their very nature, analytical models do not allow for overlapping biogeochemical zones or depth dependent porosity, which introduces a certain error to simulation results. However, the energy yield dependent sequence of oxidants is generally valid (e.g. Hensen et al., 2006) and the good agreement between OMEN-SED and the results obtained with a fully formulated numerical RTM (allowing for overlapping TEA use and depth dependent porosity, Section 3.3) shows that these are not critical limitations of OMEN-SED - even for shallow sediments."*

**2. Comment:**
In the denitrification layer, all N goes to N2. However, in the suite of processes involved in the breakdown of organic matter, ammonium is produced as well, even if nitrate serves as TEA. Ammonium produced in the denitrification zone would need to be accounted for at the transition to the oxic zone as well, which may further reduced the O2 penetration depth.
Also, processes such as DNRA or anammox are not included, even though literature surveys (e.g. Dalsgaard et al. 2005) indicate that anammox is relevant at shallower water depth.

**Response:**
Anammox is implicitly included in the model. The organic nitrogen released during the denitrification process is assumed to be directly oxidized with nitrite to $N_2$ through a coupling between denitrification and anaerobic ammonium oxidation.
However, we would like to stress again that OMEN-SED is a benthic model designed for the coupling to ESMs. Most ESMs do not even explicitly resolve N-dynamics. In addition, OMEN-SED is a system/global scale model that aims to resolve the most pertinent biogeochemical dynamics on a global scale (including a paleoenvironmental context) and estimate the main SWI-fluxes and not a model that aims at resolving specific local scale dynamics. Even most local scale RTM applications do not resolve DNRA and anammox explicitly. However, OMEN-SED could be easily adapted to explicitly resolve these processes if the specific application requires their representation (e.g. coastal ocean).

We included a sentence on this in the Section 2.2.3 "Nitrate and Ammonium" (pg. 12, lines 22-24):
*"Anaerobic ammonium oxidation (anammox) is implicitly included in the model. The organic nitrogen released during denitrification is assumed to be directly oxidized with nitrite to N2 through a coupling between denitrification and anammox."*

**3. Comment:**
Methane oxidation: All methane is assumed to be oxidized anaerobically. Is this done for simplicity? Is there no leakage term (gamma_CH4; comparable to incomplete sulfide and ammonium oxidation) because methane escaping from the anoxic zone is assumed to be removed by aerobic methane oxidation?

**Response:**
First, there is a leakage term $\gamma_{CH4}$ comparable to incomplete sulfide and ammonium oxidation (see Table 10, $SO_4$ boundary conditions 5 and 8.2; H2S boundary conditions 5 and 9). Second, it can be safely assume that almost all CH4 is oxidized anaerobically in the sediments (e.g. Reeburgh (2007) and Hinrichs & Boetius (2002) estimated > 90%) - except for active (very localized) sites and slope failure, which can, in theory, be accounted for through the gamma term.

We included this sentence in the Section 2.2.4 "Sulfate and Sulfide" (pg. 14 line 10 – pg 15 line 1):
*"It can be safely assumed that almost all CH4 is oxidized anaerobically in the sediments (e.g. Reeburgh (2007) suggests up to 90%)- except for active (very localized) sites and slope failure, which can, in theory, be accounted for through the $\gamma_{CH4}$ term."*

**4. Comment:**
Methanogenesis: the 1/2 methane to DIC ratio seems to imply acetoclastic methanogenesis. What evidence is there to ignore hydrogenotrophic methanogenesis?

**Response:**
We thank the reviewer for this valid point. However, we are not aware of an approach that would allow splitting the two pathways in quantitative terms in a global context. Following the approach of Jourabchi et al. (2005), we assume a ½ methane to DIC ratio (i.e. the acetoclastic pathway). However, OMEN-SED could be easily adapted to represent hydrogenotrophic methanogenesis globally (i.e. by adapting the stoichiometric factor by a value depending on the specific pathway).

**5. Comment:**
For a globally applicable model, the lack of CaCO3 dissolution is an obvious issue. Thus, can you expand on what problems the modeling of CaCO3 dissolution would cause (page 18)? Is this linked to the calculation of pH? Why can pH (and then carbonate) not be estimated from DIC and alkalinity?

**Response:**
In general, the coupling between multiple reaction-transport equations (i.e. strongly coupled biogeochemical dynamics) make finding an analytical solution difficult or even impossible. Calcium carbonate dissolution is kinetically controlled by the amount of $CaCO_3$ and thermodynamically controlled by the porewater concentrations of calcium ion, $Ca^{2+}$, and carbonate ions, $CO_3^{2-}$. The reaction terms of the $CaCO_3$ , (the $Ca^{2+}$), the DIC and balance equations would thus depend on the concentrations of $CaCO_3$, $Ca^{2+}$ and $CO_3^{2-}$ (DIC, pH/Alk). This set of strongly coupled equations does not have a simple analytical solution. Finding a solution to the problem requires some creativity and a number of assumptions that need to be independently tested and validated. This task thus requires some additional work and model development that go beyond the scope of this first model description paper centered on organic carbon and nutrients. However as pointed out in Section 5 "already planned future extensions of OMEN-SED include an explicit description of carbonate dissolution and iron."

**6. Comment:**
Is there no P sorption on iron oxides below the oxic zone? If so, why?

**Response:**
We here follow the approaches of Slomp et al. (1996) and Gypens et al. (2008) who both assume that $HPO_4^{2-}$ is released due to the reduction of Fe oxides throughout the reduced sediment (i.e. starting at zox).

**7. Comment:**
Why does some of the ammonium created below the oxic zone escape oxidation, but oxidation of ammonium to nitrate is complete in the oxic zone?

**Response:**
This is a misunderstanding: There is also leakage term ($\gamma_{NH4}$) for ammonium to nitrate oxidation in the oxic zone (see Eqs. 12, 15 and 16).

**8. Comment:**
A fraction of the sulfide produced is assumed to escape complete oxidation. Does this mimicking the effect of precipitation with iron, rather than escape from the sediment?

**Response:**
In the reviewed version of OMEN-SED, this fraction mimicked the escape from the sediment. However, in response to the first critical comment of reviewer 3 (K. Wallmann) we made the following changes to OMEN-SED:
When coupled to an ESM the $\gamma_{H2S}$ value (fraction of H2S that is oxidised) becomes dependent on the bottom water oxygenation state. That is, $\gamma_{H2S}$ = 1.0 for oxic bottom waters and a user defined value $\gamma_{H2S}$ <1.0 for anoxic bottom waters.
In addition, we introduce another parameter ($\gamma_{FeS}$) representing the fraction of sulfide that is precipitated as pyrite (i.e. $0.0 <= \gamma_{FeS} < 1 - \gamma_{H2S}$) in the sulfate reduction zone. $\gamma_{FeS}$ is an auxiliary parameter used as a fix until iron is explicitly represented. We thus assume that a user-defined fraction of the produced H2S precipitates as FeS(FeS2) in the sulfate reduction zone. If a user does not want to make any assumptions about FeS precipitation – it can be set it to 0.

The text, tables and equations (for $O_2$, $SO_4$, $H_2S$ and alkalinity) are changed accordingly. The presented results have not been changed and we note that $\gamma_{FeS}$ = 0.0 for all simulations, as we do not want to make any assumptions. About FeS precipitation.

Changes made in:
2.2.2 Oxygen: Table 3 boundary condition 4.2 + Text pg. 12 lines 14-16:
*" It is assumed that respective fractions (γNH4 and γH2S) are directly reoxidised at the oxic/anoxic interface and the remaining fraction escapes reoxidation **(or is precipitated as pyrite, γFeS )**."*

2.2.4 Sulfate and Sulfide: Equations 23 + Table 5 boundary condition 5) + (pg. 14 lines 9-10):
*"In the sulfidic zone a defined fraction of sulfide, γFeS , can be precipitated as pyrite (in the presented simulations γFeS = 0.0 as we do not want to make any assumptions about pyrite precipitation)."*

2.2.7 Alkalinity: Table 8 boundary condition 5)
Text (pg. 20 lines 5-7):
*" In addition, the effect of secondary redox reactions, such as nitrification, sulfide and methane oxidation**, as well as pyrite precipitation,** are implicitly accounted for in the boundary conditions. "*

Text (pg. 20 lines 21-23):
*"The decrease of alkalinity due to oxidation of reduced species produced in the anoxic zones **and due to the precipitation of pyrite** (with stoichiometry ALK$^{NIT}$ , ALK$^{H2S}$ and **ALK$^{FeS}$** ) is implicitly taken into account through the flux boundary condition at zox (Table 8 Eq. 5). "*

2.4.2 Stoichiometries and reaction parameters: (pg. 27 lines 19-23):
*"However, when coupled to an ESM γH2S becomes dependent on the bottom water oxygenation state. That is, γH2 S = 1.0 for oxic bottom waters and a user defined value γH2S < 1.0 for anoxic bottom waters. The parameter γFeS represents the fraction of sulfide that is precipitated as pyrite in the sulfidic zone. The majority of H2S produced by sulfate reduction is reoxidised, but it is estimated that ∼ 10 − 25% is eventually buried as pyrite (Bottrell and Newton, 2006). However, this fraction can vary significantly over geological timescales (Berner, 1984). If a user does not want to make any assumptions about pyrite precipitation – it can be set to 0 (as in the results presented here)."*

We also added pyrite precipiation to Table 1 and A1 in the Appendix.

**9. Comment:**

Iron cycling is not represented explicitly but some of the effect of iron cycling is parameterised. With its effect on sulphur cycling, P sorption and C mineralization (metal reduction can be the main mineralization pathway, see e.g. Canfield et al. 1993), I don't fully understand the reason for doing so (apart from the added complexity when dealing with another solid phase).

**Response:**

OMEN-SED will be mainly applied on a system/global scale, coupled to an ESM, where iron reduction has been shown to play just a minor role (i.e. about 3% of the global carbon mineralization rate Thullner et al. (2009); or compare Archer et al. (2002) Fig. 9D). Also, Fe-dynamics are generally not explicitly resolved in ESMs. However, as stated in Section 5, "already planned future extensions of OMEN-SED include an explicit description of iron."

**10. Comment:**

transport processes: A 1D diffusion/bioturbation model clearly faces major challenges in the coastal ocean, where sediments one predominantly are permeable. And setting fir=1, implying no bioirrigation, is also a very strong assumption.

**Response:**

Most sediments on the globe are non-sandy, therefore we had decided to neglect sandy sediments. However, the bioirrigation coefficient has been changed and is now represented by the empirical relationship with seafloor depth derived by Soetaert et al. (1996): fir = Min{1; 15.9 · z−0.43 }. We also added to the limitations section that our approach might not be appropriate to simulate non-accumulating permeable sands.
The text has been changed to (pg. 25 lines 21-23):
"*Soetaert et al. (1996) derived an empirical relationship between fir and seafloor depth (fir = Min{1; 15.9 · z −0.43 }) based on observations from Archer and Devol (1992) and Devol and Christensen (1993) which is used in OMEN-SED*"
And in the limitations section (pg. 54 lines 17-18):
"*Also note that our 1-D diffusion/bioturbation model might not be appropriate to simulate non-accumulating permeable sands of the coastal ocean.*"

**11. Comment:**

The 2G model is a sensible choice. Model parameterisation is a general concern (page 3, line 30), but I suggest to cite some promising new approaches to address this issues such as presented by https://www.biogeosciences-discuss.net/bg-2017-397

**Response:**

We added the suggested reference to the manuscript (Section 4.2 "Parameterising the OM degradation rate constants in a global model"). However, the suggested approach is based on empirical relationships derived from modern ocean data, as well as strong assumptions. Its applicability to past and future oceans is thus questionable and the problem of parameterizing organic matter reactivity remains for these applications.

Added sentence (page 45 lines 19-22):
"*Stolpovsky et al. (2015, 2017) suggested empirically derived approaches to constrain degradation rate constants in a 2G model on a global scale. These approaches are derived from present-day observations and might help constrain parameters for present-day applications. However, the problem*

*of constraining 2G degradation model parameters remains for largely different environmental conditions encountered in the past that could also prevail in the future (Arndt et al., 2013)."*

**12. Comment:**
site comparison case studies: - Table 13: fix the depths for the SB and IM sites (SB is the 585m site ...).

**Response:**
Changed.

**13. Comment:**
site comparison case studies: - Why are the stoichiometric factors set to default values when Epping et al. provide the C/N ratio of the surface sediment?

**Response:**
We intended to do as little site tuning as possible in order to test how the default model performs for these sites and to be able to evaluate the performance of the model in data poor areas.

**14. Comment:**
site comparison case studies: - If a non-local exchange mechanism resulting from bioirrigation is invoked for the Canyon site, what is the source of the high ammonium (and DIC) leading to the observed increase in concentration at depth?

**Response:**
As suggested by the papers describing the study area (van Weering et al., 2002; Epping et al., 2002), we assume it is a result of degradation of organic matter which has been delivered from the shelf to the ocean interior.

**15. Comment:**
- global transect case study: I don't see the value of the 5% reoxidation case. This is simply unrealistic for the conditions considered here. A possible explanation why lower gammas are giving better match to the data in the shallow sites is that under those conditions, the conceptual model of a vertical separation of reaction zones is more and more violated, so the gamma become a 'fudge-factor' to account for this (see also above comments about the coastal ocean).

**Response:**
We agree with the reviewer, that gamma is a fudge factor. It accounts for all the processes that may enhance escape from re-oxidation but are not explicitly resolved. However, we would reject the comment that this is a problem with assuming strict zonation. We would argue that this reflects the more intense dynamics in shallow ocean regions and that the increased escape is due to enhanced macrofaunal activity.
We also want to stress again, that similar analytical approaches (with distinct redox zones) have given good results for coastal/estuarine sediments (e.g. Billen, 1982; Goloway and Bender, 1982; Jahnke et al., 1982; Ruardij and Van Raaphorst, 1995; Slomp et al., 1996; Gypens et al., 2008).

**16. Comment:**
- link to cGENIE: - on page 43, it says tha if fPOC is computed to be > 1 (more than 100% is preserved), then this is discarded and all POC is remineralized. Imposing constraints is fine, but what is the rationale for jumping from >100 to 0 % preservation?

**Response:**
The result fPOC > 1.0 does not imply that 100% preservation is a realistic result. It just means that the OMEN-SED solution does not provide sensitive values. Therefore, its results are discarded and a reflective boundary is assumed (a reflective boundary is a better choice than the conservative as in most cases the majority of OM is degraded during early diagenesis). However, this is just a safety measure and has not occurred in our experiments so far.

**17. Comment:**
- link to cGENIE. Page 44 discusses the challenges in applying the model in such a setting. In addition, deposition fluxes may change over time. At what point is the steady state assumption on the POC profile still valid under such settings? This is addressed summarily at of the bottom of page 54. However, I think it is important to lead with this, before interpreting the data-model comparison.

**Response:**
As suggested by the reviewer the steady-state assumption is addressed earlier in the text.
We added the following sentence to Section "4.2 Parameterising the OM degradation rate constants in a global model" (pg 45, lines 12-13):
*"Furthermore, by assuming steady-state in OMEN-SED we assume that deposition fluxes of OM are constant over the characteristic timescales of the reaction-transport processes."*

**18. Comment:**
On p.54 it says "In theory, its scope of applicability thus ranges from the regional to the global and from the seasonal to the millennial time-scale. " - in the following para- graph they recognize that "This steady-state assumption is only valid if the variability in boundary conditions and fluxes is generally longer than the characteristic timescales of the reaction-transport processes. "
I recommend to be a little more cautious in the application of the model, since I am not convinced that violations of assumptions underlying the conceptual model and non- steady state effects can be ignored. The model clearly requires substantial tuning. It is clear that the authors are aware of the shortcomings, and they discuss that the model may not be adequate to assess seasonal patterns (and one can think of additional set- tings, where fluxes vary over timescales intrinsic to the POC profile in the top 50cm of sediment modeled here). My concern is that they largely ignore them in their appli- cation, before acknowledging them in the discussion.

**Response:**
As stated in the manuscript, OMEN-SED is first and foremost designed for the coupling to ESMs.
More specific tuning/adaptation is needed if OMEN-SED is used for specific, regional environments, e.g coastal environments.
However, we would like to re-emphasize that the current version of OMEN-SED performs well across different depositional environments ranging from the coastal to the deep ocean as evidenced by the

model-data and model-model comparison. As outlined in the "Scope of applicability and model limitations" section additional developments, such as adapting pseudo-transient dynamics will further facilitate the application of OMEN-SED to more dynamic environments. A number of benthic models specifically designed for coastal/estuarine environments (e.g. ERSEM Ruardij and Rapphorst et al., 1997; Arndt and Regnier, 2007) have successfully applied such an approach. We therefore maintain our point of view that, in theory, the scope of applicability of OMEN-SED also includes coupling to system-scale estuarine and/or coastal ocean models..

**19. Comment:**
I also suggest to tone down the finding that "A comparison between simulated OM contents and observations indicates that depth dependent k-f relationships provide the best fit (Section 4.2.2), confirming more theoretical considerations regarding the different time and reactivity scales that need to be considered (see Section 4.2). " The age-reactivity relationship is pretty well established, without confirmation by this modeling effort.

**Response:**
This is a misunderstanding. We do not argue that model results "confirm" the reactivity-age link. We wanted to emphasize these results confirm that reducing the continuous distribution of organic matter reactivities into two distinct reactivity classes (2G Model) requires different k-f values for shallow vs deep ocean sediments because of the largely different reaction timescales involved (also see Fig. 10).

To clarify, we rephrased the sentence (pg 54 lines 22-25):
*"A comparison between simulated OM contents and observations indicates that a depth dependent k-f relationship provides the best fit (Section 4.2.2). These results confirm that reducing the continuous distribution of organic matter reactivities into two distinct reactivity classes (2G Model) requires different k-f values for shallow vs deep ocean sediments because of the largely different reaction timescales involved (also see Fig. 10)."*

**OMEN-SED – cGENIE coupling**
**20. Comment:**
The stated purpose of section 4 is …
… The validation of the coupled model requires more work, and I wonder whether this was not better done in a separate paper, in which the coupling to cGENIE and the parameterization of POC mineralization was explored in more detail.

**Response:**
As stated in the manuscript (page 45, lines 23-26):
*"Our objective is not to perform and discuss a detailed calibration of the coupled models as this is beyond the scope of this sediment model development paper. Rather we want to showcase the feasibility of the model coupling, illustrate the range of results and thus information that can be generated with OMEN-SED and verify that model results capture the main observed global benthic biogeochemical features."*

We think that demonstrating how OMEN-SED can be coupled to an ESM and illustrating the type of output/information generated by OMEN-SED within such a coupling is a central aspect of the model description paper.
However, we agree trimming down this section (as in the re-submitted version). We will discuss an

improved model-data analysis (also using observations of SWI-fluxes) in a follow-up publication (as also suggested by reviewer #3 K. Wallmann).

Specifically, the sensitivity analysis for the spatially uniform degradation rate constants (Figure 12) and it's discussion has been removed (compare pages 47-50).

**21. Comment:**
Figures 12 - 14: I gather the R2 values are for the bin averages. I don't see much value of that, as over- and underpredictions cancel each other out in the averaging. Why not compute statistics for the actual model results with the Seiter data directly?

**Response:**
Figure 12 has been deleted from the manuscript. As the statistics for the actual data are not helpful/misleading (see also comment 4 of reviewer #2) we decided to remove the R2 values in Figures 13 and 14 (as well as their discussion in the text). Compare changes on pages 47-50.

**Minor comments:**

**22. Comment:**
page 8/ line 1: It is said that all parameters in Eq. 1 may vary with depth, but above it is stated that porosity and burial rates are constant with depth.
**Response:**
We thank the reviewer for highlighting this. This has been changed in the revised manuscript to (pg 8 lines 1-2):
*"All parameters in Eq. (1)**, apart from porosity and burial rate,** may vary with **sediment** depth and many reaction rate expressions depend on the concentration of other species. "*

**23. Comment:**
- the fraction of POC buried is defined as the POC at $z=0$ relative to the POC at depth. Why is it not defined as the flux at $z=0$ vs. the flux at depth (it seems Eq. 5 ignores the diffusion flux)?
**Response:**
We decided to calculate the fraction of POC preserved dependent on the concentrations of POC at $z=0$ and $z=zinf$ mainly because this information is required by cGENIE. Also POC at $z=0$ is calculated on the basis of the flux provided by cGENIE, therefore does include advection-diffusion-reaction.

**24. Comment:**
Related to that, page 10, line 18 refers to a concentration/flux boundary conditions at the SWI. The following equations and Table 2 only show a known concentration, not a flux condition. However, the latter would be useful when connecting the sediment model to a model of the water column. On page 43, Eq. 51 this is addressed - make this clear earlier.
**Response:**
We thank the reviewer for highlighting this. The text has been changed and a reference to Eq. 51 has been added (pg 10 lines 17-20):
*"For organic matter, OMEN-SED applies a known **concentration** at the sediment-water interface and assumes continuity across the bottom of the bioturbated zone, zbio. **When OMEN-SED is coupled to an ESM, the POC depositional flux from the coupled ocean model is converted to a concentration by**

*solving the flux divergence equation (51)."*

**25. Comment:**
Can lines 21-23 on page 21 be deleted?
**Response:**
We refer to both equations later in the text (pg 22 line 14 and 22), therefore we would like to keep them in.

**26. Comment:**
page 31, line 2: specify NH4, SO4 and H2S FLUXES
**Response:**
Text has been corrected as suggested (pg 29 line 33).

**27. Comment:**
Figure 3 did not help me much. Does the green dashed vertical arrow indicate possible locations of zbio?
**Response:**
Yes, it does indicate locations of zbio. We added this to the caption of Figure 3. However, we strongly believe that Figure 3 illustrates the bioturbation boundary problem in an efficient way and also highlights the integration constants and ODE solutions for the different sediment layers.
New caption:
*"Schematic of the generic boundary condition matching (GBCM) problem. Showing the resulting integration constants ($A_i$ , $B_i$ ) and ODE solutions ($E_i$ , $F_i$, $G_i$ ) for the different sediment layers and the bioturbation boundary **(possible locations are indicated by the green vertical arrow)**."*

---

## Author Comment (AC2) · 8 May 2018

**Replies to Anonymous Referee #2 on "OMEN-SED 0.9: A novel, numerically efficient organic matter sediment diagenesis module for coupling to Earth system models"**

**1. Comment:**
The model neglects the effect of sediment compaction "due to mathematical constraints". I understand the rational for this and accept a consistency of this assumption to near-seafloor (bioturbated) sediments; however, this might be a problem for deeper sediments discussed in the paper (down to 50 or 100cm). The authors should either define different porosity values for different depth-zones or to demonstrate that the results are not particularly sensitive to the value of this parameter.

**Response:**
Assuming constant porosity is required to solve the diagenetic equation analytically. It is a mathematical limitation and it will induce a certain error. However, the error is not very large and we have already shown this by comparing the performance of OMEN-SED against observed data (Section 3.2) and against model results from a fully formulated RTM with depth-varying porosity (Section 3.3). The comparison of OMEN-SED with the results of the numerically solved RTM (Section 3.3) allows evaluating to which extend simplifying assumptions (e.g. constant porosity, non-overlapping redox zones etc) affect simulation results and, thus, quantitatively test the performance of the computationally efficient OMEN-SED approach against the computationally expensive numerical approach.
We also want to reiterate that OMEN-SED is designed for the coupling to ESMs and thus for global scale applications (see responses to reviewer #1). The novel model represents a big advance compared to the description of benthic-pelagic exchange processes currently incorporated into ESMs (Hülse et al., 2017; also see comment by the K. Wallmann). Conservative and reflective boundaries, as well as simple box models are characterized by much stronger, simplifying assumptions and far bigger limitations than constant porosity.

We included a little paragraph on this in the limitation section (pg 54 lines 3-8):
*"Furthermore, by their very nature, analytical models do not allow for overlapping biogeochemical zones or depth dependent porosity, which introduces a certain error to simulation results. However, the energy yield dependent sequence of oxidants is generally valid (e.g. Hensen et al., 2006) and the good agreement between OMEN-SED and the results obtained with a fully formulated numerical RTM (allowing for overlapping TEA use and depth dependent porosity, Section 3.3) shows that these are not critical limitations of OMEN-SED - even for shallow sediments."*

**2. Comment:**
Dividing the sediment column into functional zones in such a strict manner does not always represent reality well. Thus, "nitrogenous" zone may overlap with "oxic" zone. This assumption, as far as I understand, made it impossible to simulate nitrate SWI flux directed into the sediments in oxygenated environment, which is definitely not true. Validation of the model against measured benthic fluxes would probably demonstrate to some extent accordance of suggested method with real benthic system.

**Response:**
First, it is possible to simulate nitrate influx into the sediments in oxygenated environments with

OMEN-SED (see e.g. Fig. 6C).

In the following we repeat the response given to the 1[st] comment of Reviewer #1:

We agree with the reviewer that different biogeochemical zones can overlap. However, as stated in the text, OMEN-SED is designed for the coupling to ESMs and its formulation is thus first and foremost guided by achieving numerical efficiency while retaining biogeochemical reality. As summarized in the manuscript, there are essentially two approaches that can be used to describe biogeochemical processes in models. The first approach solves the general diagenetic equation numerically on a regular or irregular grid and biogeochemical zonation emerges in response to inhibition terms allowing a certain degree of overlap between biogeochemical zones. This approach is highly flexible and thus preferable. Yet, its excessive computational demand unfortunately renders its application within a three-dimensional Earth System Model framework impossible. On the other hand, analytical models that subdivide the sediment into distinct biogeochemical zones are computationally efficient and thus ideally suited to describe diagenetic dynamics in ESM.
By their very nature, analytical models do not allow for overlapping biogeochemical zones. As stated in the manuscript, this is a simplification. However, we disagree with the reviewer that this simplification would *per-se* prevent the application of such analytical approaches in shallower aquatic environments. In fact, OMEN-SED builds on a number of analytical models that were developed to investigate local, coupled nutrient and oxygen cycles in coastal sediments (e.g. Billen, 1982; Goloway and Bender, 1982; Jahnke et al., 1982; Slomp et al. 1996). Similar approaches were later successfully applied from oxic to anoxic sediments and at the regional coastal ocean scale (e.g. Ruardij and Van Raaphorst, 1995; Tromp et al., 1995; Gypens et al., 2008). In particular, Gypens et al., (2008) points out that accounting for secondary redox process in the boundary condition induces little error as: "Using a numerical model, Soetaert et al. (1996) showed that this re-oxidation mainly occurs at the oxic-anoxic transition interface."

Finally, the good agreement between OMEN-SED and the results obtained with a fully formulated numerical RTM (compare Section 3.3, allowing for overlapping TEA use) shows that this is not a critical limitation of OMEN-SED - even for shallow sediments.

We again want to refer to the paragraph we added to the limitations section (see last comment, pg 54 lines 3-8).

**3. Comment:**
Nitrogen dynamics include "the metabolic production of ammonium, nitrification, denitrification as well as ammonium adsorption". Denitrification is considered as a single-step process ignoring NO2- production/consumption and anaerobic ammonium oxidation (Anammox) which is undoubtedly a significant component of the biogeochemical nitrogen cycle (Devol, 2015). In other words, nitrogen dynamics is somewhat simplified. This simplification should be quantified/discussed in more details.

**Response:**
In the following we repeat the response given to the 2nd comment of Reviewer #1:

Anammox is implicitly included in the model. The organic nitrogen released during the denitrification process is assumed to be directly oxidized with nitrite to $N_2$ through a coupling between denitrification and anaerobic ammonium oxidation.
However, we would like to stress again that OMEN-SED is a benthic model designed for the coupling

to ESMs. Most ESMs do not even explicitly resolve N-dynamics. In addition, OMEN-SED is a system/global scale model that aims to resolve the most pertinent biogeochemical dynamics on a global scale (including a paleoenvironmental context) and estimate the main SWI-fluxes and not a model that aims at resolving specific local scale dynamics. Even most local scale RTM applications do not resolve DNRA and anammox explicitly. However, OMEN-SED could be easily adapted to explicitly resolve these processes if the specific application requires their representation (e.g. coastal ocean).

We included a sentence on this in the Section 2.2.3 "Nitrate and Ammonium" (pg. 12, lines 22-24)::
"*Anaerobic ammonium oxidation (anammox) is implicitly included in the model. The organic nitrogen released during denitrification is assumed to be directly oxidized with nitrite to N2 through a coupling between denitrification and anammox.*"

**4. Comment:**
The efficiency of binning procedure discussed in section 4.2.1 is doubtful. First of all, such binning assumes presence of STD bars on the plots. Also, I think that it would be more logical to group POC content into POC rain rate (RRPOC) classes rather than WD classes as RRPOC may significantly vary at different regions of the ocean of the same WD. Finally this binning gives a false impression of a good POC content fit. I realize that parameterization of multi-G model is beyond the scope of this sediment model development paper, therefore I suggest to use existing way to parameterize multi-G models and validate your model against the databases suggested in those studies (for example Stolpovsky et al., (submitted) https://www.biogeosciences-discuss.net/bg-2017-397/ ).

**Response:**
We thank the reviewer for the suggestion. We decided to follow suggestions from reviewer #1 and #3 (K. Wallmann) and shortened the cGENIE coupling section. Figure 12 (and its discussion) has been deleted from the manuscript. The R2 values in Figures 13 and 14 (as well as their discussion in the text) has been removed as well. Compare changes on pages 47-50.

The ranges of simulated SWI-fluxes from the stand-alone OMEN-SED model are already compared to the Stolpovsky et al., (2015) database in Figure 6.

If binned by RRPOC for uniform k-values, all grid-cells with same RRPOC have the same preservation in OMEN-SED. Therefore, this would not be very useful.

**5. Comment:**
POC is not a very good constraint, since measured POC is in large part the less reactive stuff that is left over after mineralization of the more reactive fractions. This was shown in Stolpovsky et al., 2015 paper (see the discussion in section 4.3). Fluxes at the SWI are believed to be a better constraint.

**Response:**
We shortened the coupling section of the manuscript (we removed the sensitivity analysis with the spatially uniform degradation rate constants, compare pages 47-50) and we will discuss an improved model-data analysis of the coupled model, using existing parameterizations and maps of SWI-fluxes, in a follow-up publication. Also compare response to comment 20 of reviewer #1:
 As stated in the manuscript (page 45, lines 23-26):
" *Our objective is not to perform and discuss a detailed calibration of the coupled models as this is*

*beyond the scope of this sediment model development paper. Rather we want to showcase the feasibility of the model coupling, illustrate the range of results and thus information that can be generated with OMEN-SED and verify that model results capture the main observed global benthic biogeochemical features."*

**Minor comments.**

**6. Comment:**
Eq. 1: As a time and depth independent parameter, porosity should be moved out of differential in order to emphasize that it is constant: Porosity*dC/dt instead of d(Porosity*C)/dt.
**Response:**
This has been changed as indicated.

**7. Comment:**
P. 8, L. 1: It is not immediately clear that the authors are talking about water (not sediment) depth.
**Response:**
We agree that this is a bit misleading. We are actually talking about sediment depth. This has been changed in the revised manuscript to (pg. 8 lines 1-2):
*"All parameters in Eq. (1)****, apart from porosity and burial rate,*** *may vary with* ***sediment*** *depth and many reaction rate expressions depend on the concentration of other species. "*

**8. Comment:**
Eq. 5: This representation sounds a bit odd. I think $z\infty$ should be replaced with zmax, as POC content at infinite depth believed to be zero.
**Response:**
The POC content of marine sediments does not tend to zero. A significant amount of POC is buried in marine sediments and enters the longterm C cycle (rock cycle). Without this imbalance between production and respiration, no O2 would have accumulated in the atmosphere.
But we agree that the use of $z\infty$ is not ideal, as the sediment column in OMEN-SED is not modeled until infinite depth. We have replaced $z\infty$ with zmax in the entire manuscript.

**9. Comment:**
P. 9, L. 25: SWI is given without initial explanation.
**Response:**
The explanation has been added at this part in the manuscript: *"...* ***sediment-water interface (SWI)*** *…"*

**10. Comment:**
P. 25, L. 6 – 13: I agree that bioirrigation may enhance SWI fluxes of dissolved species, therefore I do not understand why this way of transport is technically ignored for all water depths (fir=1)?
**Response:**
In the following we repeat the answer given to Comment 10 of Reviewer #1:

The bioirrigation coefficient has been changed and is now represented by the empirical relationship with seafloor depth derived by Soetaert et al. (1996): fir = Min{1; 15.9 · z−0.43 }.
The text has been changed to (pg. 25 lines 21-23):
*"Soetaert et al. (1996) derived an empirical relationship between fir and seafloor depth (fir = Min{1;*

*15.9 · z −0.43 }) based on observations from Archer and Devol (1992) and Devol and Christensen (1993) which is used in OMEN-SED*"
And in the limitations section (pg. 54 lines 17-18):
*"Also note that our 1-D diffusion/bioturbation model might not be appropriate to simulate non-accumulating permeable sands of the coastal ocean."*

**11. Comment:**
P. 27, L. 28: PAWN is given without explanation.
**Response:**
As the name PAWN is derived from the authors names and not an acronym we do not think this information is of any value here.

**12. Comment:**
Fig. 7: Please add ticks and numbers to X-axis on H2S at 2213 and 4298m and NH4 at 108m. Some plates have very inconvenient ranges on horizontal axis, for example H2S at 4298m.
**Response:**
This has been changed as indicated.

**13. Comment:**
Sec. 3.3.2: I do not understand the rational for comparing OMEN-SED results with another model (Thullner et al. 2009). I would suggest comparing it to existing SWI flux database mentioned before (Stolpovsky et al., 2015). Also, reporting global denitrification rate modeled with OMEN-SED and its comparison with previous studies would support the model.
**Response:**
We evaluate the performance of OMEN-SED by comparing model results with data (section 3.2), as well as the results of a fully-formulated, numerical RTM (section 3.3). The comparison of OMEN-SED with the results of the numerically solved RTM allows evaluating to which extend simplifying assumptions (e.g. constant porosity, non-overlapping redox zones etc) affect simulation results and, thus, quantitatively test the performance of the computationally efficient OMEN-SED approach against the computationally expensive numerical approach.
The ranges of simulated SWI-fluxes from the stand-alone OMEN-SED model are already compared to the Stolpovsky et al., (2015) database in Figure 6.

**14. Comment:**
P. 55, L. 24 – 25: Bold assumption, I suggest to avoid such formulations. The major advantage of OMEN-SED is its tremendously low computation time which is so important for ESMs. As always, only two options of the following three can be true the same time: "quickly", "cheaply (super-computer is not needed)" or "qualitatively".

**Response:**
This is not an assumption, but the conclusion from the model-data and model-model comparison at the system scale. To clarify this, the sentence as been changed to (pg. 55 lines 5-6):
*"We have shown that the performance of OMEN-SED **at the system scale** is similar to that of a fully formulated, multi-component numerical model."*

---

## Author Comment (AC3) · 8 May 2018

**Replies to Referee #3: K. Wallmann on "OMEN-SED 0.9: A novel, numerically efficient organic matter sediment diagenesis module for coupling to Earth system models"**

**1. Comment:**
The model ignores sulfide precipitation and pyrite formation. Consequently, dissolved sulfide produced by sulfate reduction and AOM at depth diffuses upward to be either oxidized by oxygen or released into ambient bottom waters. This is a very unrealistic set-up. In most sediments dissolved sulfide is removed from the pore water by pyrite precipitation while the remaining sulfide is oxidized with ferric iron, nitrate and nitrite before it can reach the oxic surface layer or the ocean. Aerobic sulfide oxidation is only important in highly reactive surface sediments where the diagenetic sequence is not maintained but several electron acceptors are used simultaneously. The model is based on the assumption that electron acceptors are used sequentially rather than simultaneously. Hence, it cannot simulate situations where aerobic sulfide oxidation is important but creates high rates of aerobic sulfide oxidation in geological settings where this process does in fact not occur. The authors should try to fix this problem. They could for example abandon the model parameter that defines the fraction of dissolved sulfide that escapes into bottom waters. In the modern ocean, sulfide leakage from sediments occurs only in very rare situations and it does not make sense to simulate these anoxic sediments with a model that ignores iron cycling, pyrite formation and sulfide precipitation. The authors could instead introduce a parameter that defines the fraction of sulfide that is precipitated as pyrite and update the alkalinity model accordingly.

**Response:**
We thank Prof. Wallmann for this very valid suggestion. We made the following changes to OMEN-SED:
When coupled to an ESM the $\gamma_{H2S}$ value (fraction of H2S that is oxidised) becomes dependent on the bottom water oxygenation state. That is, $\gamma_{H2S} = 1.0$ for oxic bottom waters and a user defined value $\gamma_{H2S}$ <1.0 for anoxic bottom waters. For simplicity this is still implemented as aerobic sulfide oxidation. In addition, we introduce another parameter ($\gamma_{FeS}$) representing the fraction of sulfide that is precipitated as pyrite (i.e. $0.0 <= \gamma_{FeS} < 1 - \gamma_{H2S}$) in the sulfate reduction zone. $\gamma_{FeS}$ is an auxiliary parameter used as a fix until iron is explicitly represented (see pg. 54 lines 10-11, "... already planned future extensions of OMEN-SED include an explicit description of carbonate dissolution iron."). We thus assume that a user-defined fraction of the produced H2S precipitates as FeS(FeS2) in the sulfate reduction zone. If a user does not want to make any assumptions about FeS precipitation – it can be set it to 0.

The text, tables and equations (for $O_2$, $SO_4$, $H_2S$ and alkalinity) are changed accordingly. The presented results have not been changed and we note that $\gamma_{FeS} = 0.0$ for all simulations, as we do not want to make any assumptions.

Changes made in:
2.2.2 Oxygen: Table 3 boundary condition 4.2 + Text pg. 12 lines 14-16:
*" It is assumed that respective fractions (γNH4 and γH2S) are directly reoxidised at the oxic/anoxic interface and the remaining fraction escapes reoxidation **(or is precipitated as pyrite, γFeS )**."*

2.2.4 Sulfate and Sulfide (pg. 14 lines 9-10): + Equations 23 + Table 5 boundary condition 5)
*"In the sulfidic zone a defined fraction of sulfide, $\gamma FeS$ , can be precipitated as pyrite (in the presented simulations $\gamma FeS = 0.0$ as we do not want to make any assumptions about pyrite precipitation)."*

2.2.7 Alkalinity: Table 8 boundary condition 5)
Text (pg. 20 lines 5-7):
*" In addition, the effect of secondary redox reactions, such as nitrification, sulfide and methane oxidation**, as well as pyrite precipitation,** are implicitly accounted for in the boundary conditions. "*

Text (pg. 20 lines 21-23):
*"The decrease of alkalinity due to oxidation of reduced species produced in the anoxic zones **and due to the precipitation of pyrite** (with stoichiometry $ALK^{NIT}$ , $ALK^{H2S}$ and $\mathbf{ALK^{FeS}}$ ) is implicitly taken into account through the flux boundary condition at zox (Table 8 Eq. 5). "*

2.4.2 Stoichiometries and reaction parameters: (pg. 27 lines 19-23):
*"However, when coupled to an ESM $\gamma H2S$ becomes dependent on the bottom water oxygenation state. That is, $\gamma H2 S = 1.0$ for oxic bottom waters and a user defined value $\gamma H2S < 1.0$ for anoxic bottom waters. The parameter $\gamma FeS$ represents the fraction of sulfide that is precipitated as pyrite in the sulfidic zone. The majority of H2S produced by sulfate reduction is reoxidised, but it is estimated that $\sim 10 - 25\%$ is eventually buried as pyrite (Bottrell and Newton, 2006). However, this fraction can vary significantly over geological timescales (Berner, 1984). If a user does not want to make any assumptions about pyrite precipitation – it can be set to 0 (as in the results presented here)."*

We also added pyrite precipiation to Table 1 and A1 in the Appendix.

**2. Comment:**
The authors use an empirical equation by Middleburg et al. (1997) to define burial velocity (w) as function of water depth (Eq. 46). Unfortunately, w is seriously overestimates by this equation. As an example, w at 1000 m water depth results as 160 cm kyr-1 applying Eq. 46 whereas the available data indicate global mean rates in the order of 10 – 20 cm kyr-1 for this water depth (Burwicz et al., 2011). The extremely high burial velocities derived from Eq. 46 compromise the TOC concentration and other model results especially when the model is applied at global scale.

**Response:**
The Middelburg et al. (1997) equation is just used in the stand-alone OMEN-SED version. When coupled to cGENIE we use the burial velocity of the ESM. In addition, the Burwicz et al. (2011) parameterisation is already added as an option in OMEN-SED (see pg. 24). We made it the default version for the stand-alone model. The sentence as been changed accordingly (compare pg. 24 line 28 – pg. 25 line 3).

**3. Comment:**
**Comment 3.1:** OMEN-SED is able to reproduce the strong down-core decrease in organic matter reactivity observed in marine sediments by using two or more organic matter fractions with widely different reactivity. This strength is nicely demonstrated in section 3.3 where the authors are able to show that typical pore water profiles are reproduced by

the model applying kinetic constants (k1, k2) that span several orders of magnitude (Tab. 13). Subsequently, the authors try to reproduce the TOC distribution at the deep-sea floor by coupling OMEN-SED to an earth system model. I think that TOC in surface sediments is not a good parameter to validate the model because almost the entire organic matter raining to the deep-sea floor is degraded in the surface sediment rather than preserved as sedimentary TOC. TOC concentrations in surface sediments at the deep-sea floor are governed by TOC rain rates, mass accumulation rates (burial velocity), adsorption of organic matter on mineral surfaces, and the kinetic properties of the very small refractory fraction that survives degradation (about 1 % of the total rain rate). The strength of OMEN-SED to degrade the reactive fractions in a meaningful way does not play out in this application.

**Response:**
We agree with the statement that TOC is not necessarily a good way to validate the coupled model and we would also favor fluxes or rates. However, we are not convinced that they give much better results if the database is limited. TOC in surface sediments was the data available on a global scale and also other ESM studies compare their results to it (e.g. HAMOCC, Palastanga et al. (2011)). As mentioned earlier, we will put in some more effort in a follow-up study where we compare calculated SWI-fluxes with observations.
In addition, as stated in the manuscript (page 45, lines 23-26):
" *Our objective is not to perform and discuss a detailed calibration of the coupled models as this is beyond the scope of this sediment model development paper. Rather we want to showcase the feasibility of the model coupling, illustrate the range of results and thus information that can be generated with OMEN-SED and verify that model results capture the main observed global benthic biogeochemical features*."

**Comment 3.2:** Moreover, the model results are unrealistic. The best fit to the TOC data is apparently obtained assuming that the organic matter flux to the seabed is composed of two TOC fractions with very low reactivity in the order of $0.001 - 0.01$ yr-1 (Fig. 12). This result is not consistent with the case study presented in section 3.3 that yields much higher k values (Tab. 13).

**Response:**
The low reactivities obtained for the global application (e.g. Fig. 12 – old manuscript) agree with published results (see Arndt et al., (2013)), as well as with the results obtained with HAMOCC using a 1G-model (they found kox=0.005 yr-1 & kanox=0.002 yr-1 for deep sea sediments, Palastanga et al. (2011)). In addition, our simulated oxygen penetration depths compare well with observations (see Fig. 16). Especially deep sea sites in the gyres are characterised by very low POC input and degradation rates which causes $O_2$ to diffuse down to the basement of the sediments (Fischer et al., 2009; D'Hondt et al., 2015).
The sites used for the stand-alone case study in section 3.3 where not really deep sea sites (complete data sets from deep sea sites within gyres are difficult to obtain).

However, we decided to shorten the cGENIE coupling section (compare also response to your comment 3.3). The sensitivity analysis for the spatially uniform degradation rate constants (Figure 12) and it's discussion has been removed (compare pages 47-50).

**Comment 3.2:** Moreover, we have shown previously that this very low reactivity is not consistent with the benthic fluxes of oxygen and nitrate that have been measured at the seabed (Stolpovsky et al., 2015). The error may be caused by the too high burial velocities applied in OMEN-SED (Eq. 46) and/or may be related to the rain rate and reactivity of organic matter calculated in GENIE.

**Response:**
The Stolpovsky et al. (2015) database is a very valuable source of information and we will compare our calculated fluxes using the coupled model with it in the follow-up study. The ranges of simulated SWI-fluxes from the stand-alone OMEN-SED model are already compared to the database in Figure 6. However, we would also argue that the Stolpovsky et al. (2015) database does not contain a representative amount of very deep ocean sites (e.g. within ocean gyres) characterised by very low SWI-fluxes (see e.g. Fischer et al., 2009; D'Hondt et al., 2015).  D'Hondt el al. (2009) for instance found that the net rate of diagenetic degradation in the South Pacific Gyre is 1 to 3 orders of magnitude lower than at previously explored sites and they suggest that almost 50% of the worlds ocean may be characterised by these rates. In a more recent study D'Hondt et al. (2015) suggest: "...that oxygen and aerobic communities may occur throughout the entire sediment sequence in 15–44% of the Pacific and 9–37% of the global sea floor."

**Comment 3.3:** I would encourage the authors to delete the entire section 4 of the paper because it does not add useful information but presents rather misleading results. They should aim to present other more useful applications of their highly innovative analytical model in follow-up publications.

**Response:**
Here, we repeat parts of the response to comment 20 of reviewer #1:
We think that demonstrating how OMEN-SED can be coupled to an ESM and illustrating the type of output/information generated by OMEN-SED within such a coupling is a central aspect of the model description paper.
However, we agree trimming down this section (as in the re-submitted version). We will discuss an improved model-data analysis (also using observations of SWI-fluxes) .
Specifically, the sensitivity analysis for the spatially uniform degradation rate constants (Figure 12) and it's discussion has been removed (compare pages 47-50).

---

## Referee Report (RR1)

The authors have addressed  my comments and from my perspective the revised draft can now be published. Klaus Wallmann